# Decentralized Optimistic Hyperpolicy Mirror Descent: Provably No-Regret Learning in Markov Games

## Abstract

We study decentralized policy learning in Markov games where we control a single agent to play with nonstationary and possibly adversarial opponents. Our goal is to develop a no-regret online learning algorithm that (i) takes actions based on the local information observed by the agent and (ii) is able to find the best policy in hindsight. For such a problem, the nonstationary state transitions due to the varying opponent pose a significant challenge. In light of a recent hardness result [33], we focus on the setting where the opponent's previous policies are revealed to the agent for decision making. With such an information structure, we propose a new algorithm, Decentralized Optimistic hypeRpolicy mIrror deScent (DORIS), which achieves $\sqrt{K}$-regret in the context of general function approximation, where $K$ is the number of episodes. Moreover, when all the agents adopt DORIS, we prove that their mixture policy constitutes an approximate coarse correlated equilibrium. In particular, DORIS maintains a *hyperpolicy* which is a distribution over the policy space. The hyperpolicy is updated via mirror descent, where the update direction is obtained by an optimistic variant of least-squares policy evaluation. Furthermore, to illustrate the power of our method, we apply DORIS to constrained and vector-valued MDPs, which can be formulated as zero-sum Markov games with a fictitious opponent.

## 1 Introduction

Multi-agent reinforcement learning (MARL) studies how each agent learns to maximize its cumulative rewards by interacting with the environment as well as other agents, where the state transitions and rewards are affected by the actions of all the agents. Equipped with powerful function approximators such as deep neural networks [31], MARL has achieved significant empirical success in various domains including the game of Go [47], StarCraft [50], DOTA2 [5], Atari [38], multi-agent robotics systems [8] and autonomous driving [45]. Compared with the centralized setting where a central controller collects the information of all agents and coordinates their behaviors, decentralized algorithms [19, 42] where each agent autonomously chooses its action based on its own local information are often more desirable in MARL applications. In specific, decentralized methods (1) are easier to implement and enjoy better scalability, (2) are more robust to possible adversaries, and (3) require less communication overhead [21, 22, 9, 59, 18].

In this work, we aim to design a provably efficient decentralized reinforcement learning (RL) algorithm in the online setting with function approximation. In the sequel, for the ease of presentation, we refer to the controllable agent as the *player* and regard the rest of the agents as a meta-agent, called the *opponent*, which specifies its policies arbitrarily. Our goal is to maximize the cumulative rewards of the player in the face of a possibly adversarial opponent, in the online setting where the policies of the player and opponent can be based on adaptively gathered local information.

From a theoretical perspective, arguably the most distinctive challenge of the decentralized setting is *nonstationarity*. That is, from the perspective of any agent, the states transitions are affected by the policies of other agents in an unpredictable and potentially adversarial way and are thus nonstationary. This is in stark contrast to the centralized setting which can be regarded as a standard RL problem for the central controller which decides the actions for all the players. Furthermore, in the online setting, as the environment is unknown, to achieve sample efficiency, the player needs to strike a balance

between *exploration* and *exploitation* in the context of function approximation and in the presence of an adversarial opponent. The dual challenges of nonstationarity and efficient exploration are thus intertwined, making it challenging to develop provably efficient decentralized MARL algorithms.

Consequently, there seem only limited theoretical understanding of the decentralized MARL setting with a possibly adversarial opponent. Most of the existing algorithms [7, 53, 49, 27, 23] can only compete against the Nash value of the Markov game when faced with an arbitrary opponent. This is a much weaker baseline compared with the results in classic matrix games [17, 1] where the player is required to compete against *the best fixed policy in hindsight*. Meanwhile, [33] seems the only work we know that can achieve no-regret learning in MARL against the best hindsight policy, which focuses on the *policy revealing* setting where the player observes the policies played by the opponent in previous episodes. However, the algorithm and theory in this work are limited to tabular cases and fail to deal with large or even continuous state and action space. To this end, we would like to answer the following question:

> *Can we design a decentralized MARL algorithm that provably achieves no-regret against the best fixed policy in hindsight in the context of function approximation?*

In this work, we provide a positive answer to the above question under the *policy revealing* setting with general function approximation. In specific, we propose an actor-critic-type algorithm [29] called DORIS, which maintains a distribution over the policy space, named *hyperpolicy*, for decision-making. To combat the nonstationarity, DORIS updates the hyperpolicy via mirror descent (or equivalently, Hedge [16]). Furthermore, to encourage exploration, the descent directions of mirror descent are obtained by solving optimistic variants of policy evaluation subproblems with general function approximation, which only involve the local information of the player. Under standard regularity assumptions on the underlying function classes, we prove that DORIS achieves a sublinear regret in the presence of an adversarial opponent. In addition, when the agents all adopt DORIS independently, we prove that their average policy constitutes an approximate coarse correlated equilibrium. At the core of our analysis is a new complexity measure of function classes that is tailored to the decentralized MARL setting. Furthermore, to demonstrate the power of DORIS, we adapt it for solving constrained Markov decision process (CMDP) and vector-valued Markov decision process (VMDP), which can both be formulated as a zero-sum Markov game with a fictitious opponent.

**Our Contributions.** Our contributions are four-fold. First, we propose a new decentralized policy optimization algorithm, DORIS, that provably achieves no-regret in the context of general function approximation. As a result, when all agents adopt DORIS, their average policy converges to a CCE of the Markov game. Secondly, we propose a new complexity measure named Bellman Evaluation Eluder dimension, which generalizes Bellman Eluder dimension [25] for single-agent MDP to decentralized learning in Markov games, which might be of independent interest. Third, we modify DORIS for solving CMDP with general function approximation, which is shown to achieve sublinear regret and constraint violation. Finally, we extend DORIS to solving the approchability task [36] in vector-valued Markov decision process (VMDP) and attain a near-optimal solution. To our best knowledge, DORIS seems the first provably efficient decentralized algorithm for achieving no-regret in MARL with general function approximation.

**Notations.** In this paper we let $[n] = \{1, \cdots, n\}$ for any integer $n$. We denote the set of probability distributions over any set $\mathcal{S}$ by $\Delta_{\mathcal{S}}$ or $\Delta(\mathcal{S})$. We also let $\|\cdot\|$ denote the $\ell_2$-norm by default.

**Related works.** Our work is related to the bodies of literature on decentralized learning with an adversarial opponent, finding equilibria in self-play Markov games, CMDPs and VMDPs. These works either consider centralized setting or do not have function approximation in decentralized online setting. Due to the page limit, we compare to these works in Appendix B.

## 2 PRELIMINARIES

### 2.1 MARKOV GAMES

Let us consider an $n$-agent general-sum Markov game (MG) $\mathcal{M}_{\mathrm{MG}} = (\mathcal{S}, \{\mathcal{A}_i\}_{i=1}^n, \{P_h\}_{h=1}^H, \{r_{h,i}\}_{h=1,i=1}^{H,n}, H)$, where $\mathcal{S}$ is the state space, $\mathcal{A}_i$ is the action space of the $i$-th agent, $P_h : \mathcal{S} \times \prod_{i=1}^n \mathcal{A}_i \to \Delta(\mathcal{S})$ is the transition function at the $h$-th step, $r_{h,i} : \mathcal{S} \times \prod_{i=1}^n \mathcal{A}_i \to \mathbb{R}_+$ is the reward function of the $i$-th agent at the $h$-th step and $H$ is the length of each episode. We assume

each episode starts at a fixed start state $s_1$ and terminates at $s_{H+1}$. At step $h \in [H]$, each agent $i$ observes the state $s_h$ and takes action $a_{h,i}$ simultaneously. After that, agent $i$ receives its own reward $r_{h,i}(s_h, \boldsymbol{a}_h)$ where $\boldsymbol{a}_h := (a_{h,1}, \cdots, a_{h,n})$ is the joint action and the environment transits to a new state $s_{h+1} \sim P_h(\cdot|s_h, \boldsymbol{a}_h)$.

**Policy.** A policy of the $i$-th agent $\mu_i = \{\mu_{h,i} : \mathcal{S} \to \Delta_{\mathcal{A}_i}\}_{h\in[H]}$ specifies the action selection probability of agent $i$ in each state at each step. In the following discussion we will drop the $h$ in $\mu_{h,i}$ when it is clear from the context. We use $\pi$ to represent the joint policy of all agents and $\mu_{-i}$ to denote the joint policy of all agents other than $i$. Further, we assume each agent $i$ chooses its policy from a policy class $\Pi_i$. Similarly, let $\Pi_{-i} := \prod_{j\neq i} \Pi_j$ denote the product of all agents' policy classes excluding the $i$-th agent.

**Value functions and Bellman operators.** Given any joint policy $\pi$, the $i$-th agent's value function $V_{h,i}^\pi : \mathcal{S} \to \mathbb{R}$ and action-value (or Q) function $Q_{h,i}^\pi : \mathcal{S} \times \prod_{i=1}^n \mathcal{A}_i \to \mathbb{R}$ characterize its expected cumulative rewards given a state or a state-action pair, which are defined as below:

$$V_{h,i}^\pi(s) := \mathbb{E}_\pi\left[\sum_{l=h}^H r_{l,i}(s_l, \boldsymbol{a}_l)\bigg|s_h = s\right], Q_{h,i}^\pi(s, \boldsymbol{a}) := \mathbb{E}_\pi\left[\sum_{l=h}^H r_{l,i}(s_l, \boldsymbol{a}_l)\bigg|s_h = s, \boldsymbol{a}_h = \boldsymbol{a}\right],$$

where the expectation is w.r.t. to the distribution of the trajectory induced by executing the joint policy $\pi$ in $\mathcal{M}_{\mathrm{MG}}$. Here we suppose the action-value function is bounded:

$$Q_{h,i}^\pi(s, \boldsymbol{a}) \leq V_{\max}, \forall s, \boldsymbol{a}, h, i, \pi.$$

Notice that when the reward function is bounded in $[0, 1]$, $V_{\max} = H$ naturally.

## 2.2 DECENTRALIZED POLICY LEARNING

In this paper we consider the decentralized learning setting [27, 23, 33] where only one agent is under our control, which we call *player*, and the other agents can be adversarial. Without loss of generality, assume that we can only control agent 1 and view the other agents as a meta *opponent*. To simplify writing, we use $a_h, \mathcal{A}, r_h, \mu, \Pi, V_h^\pi, Q_h^\pi$ to denote $a_{h,1}, \mathcal{A}_1, r_{h,1}, \mu_1, \Pi_1, V_{h,1}^\pi, Q_{h,1}^\pi$ respectively. We also use $b_h, \mathcal{B}, \nu, \Pi'$ to represent the action, the action space, the policy and the policy class of the meta opponent.

By *decentralized* we mean during the episode, the player can only observe its own rewards, actions and some information of the opponent specified by the protocol, i.e., $\{s_h^t, a_h^t, \mathcal{J}_h^t, r_h^t\}_{h=1}^H$ where $\{\mathcal{J}_h\}_{h=1}^H$ is the information revealed by the opponent in each episode and we will specify it later. At the beginning of the $t$-th episode, the player chooses a policy $\mu^t$ from its policy class $\Pi$ based only on its local information collected from previous episodes, without any coordination from a centralized controller. Meanwhile, the opponent selects $\nu^t$ from $\Pi'$ secretly and probably adversely.

The learning objective is to minimize the regret of the player by comparing its performance against the best fixed policy in hindsight as standard in online learning literature [1, 20]:

**Definition 1** (Regret). *Suppose $(\mu^t, \nu^t)$ are the policies played by the player and the opponent in the $t$-th episode. Then the regret for $K$ episodes is defined as*

$$Regret(K) = \max_{\mu \in \Pi} \sum_{t=1}^K V_1^{\mu \times \nu^t}(s_1) - \sum_{t=1}^K V_1^{\mu^t \times \nu^t}(s_1), \tag{1}$$

*where $\mu \times \nu$ denotes the joint policy where the player and the opponent play $\mu$ and $\nu$ independently. We also use $\pi^t$ to denote $\mu^t \times \nu^t$.*

Achieving low regrets defined in (1) indicates that if we sample a policy $\overline{\mu}$ uniformly from $\{\mu^t\}_{t=1}^K$ at random, the resulting mixture policy will be close to the best fixed policy in hindsight.

**Relation between Definition 1 and equilibria.** An inspiration for our definition of regrets comes from the tight connection between low regrets and equilibria in the matrix game [17, 6, 10]. By viewing each policy in the policy class as a pure strategy in the matrix game, we can generalize the notion of equilibria in matrix games to Markov games naturally. In particular, a correlated mixed strategy profile $\overline{\pi}$ can be defined as a mixture of the joint policy of all agents, i.e., $\overline{\pi} \in \Delta(\prod_{i\in[n]} \Pi_i)$. Suppose the marginal distribution of $\overline{\pi}$ over the policy of agent $i$ is $\overline{\mu}_i$, then we can see that $\overline{\mu}_i$ is a

mixture of the policies in $\Pi_i$. For a correlated profile, the agents might not play their mixed policies $\overline{\mu}_i$ independently, which means that $\overline{\pi}$ might not be the product of $\overline{\mu}_i$. A coarse correlated equilibrium (CCE) is a correlated profile that all the agents have no incentive to deviate from by playing a different independent policy:

**Definition 2** ($\epsilon$-approximate coarse correlated equilibrium (CCE) for $n$-player MG). *A correlated strategy profile $\overline{\pi}$ is an $\epsilon$-approximate coarse correlated equilibrium if we have for all $i \in [n]$*

$$V_{1,i}^{\overline{\pi}}(s_1) \geq \max_{\mu' \in \Pi_i} V_{1,i}^{\mu' \times \overline{\mu}_{-i}}(s_1) - \epsilon, \tag{2}$$

*where $\overline{\mu}_{-i}$ is the marginal distribution of $\overline{\pi}$ over the joint policy of all agents other than $i$.*

**Remark 1.** *Our definition of correlated strategy profile and CCEs is slightly different from [35]. This is because we are considering with policy classes while [35] does not. In fact, our definition is more strict in the sense that a correlated profile satisfying our definition must also satisfy theirs.*

Specially, if a CCE $\overline{\pi}$ satisfies $\overline{\pi} = \prod_{i \in [n]} \overline{\mu}_i$, it is also called a Nash Equilibrium (NE). We will use our algorithm as an example to show that if a decentralized algorithm can achieve low regrets under Definition 1, we will be able to find a CCE by running the algorithm independently for each agent just like in classic matrix games.

## 2.3 FUNCTION APPROXIMATION

To deal with the potentially large or even infinite state and action space, we consider learning with general value function approximation in this paper [24, 25]. We assume the player is given a function class $\mathcal{F} = \mathcal{F}_1 \times \cdots \times \mathcal{F}_{H+1}$ ($\mathcal{F}_h \subseteq (\mathcal{S} \times \mathcal{A} \times \mathcal{B} \to [0, V_{\max}])$) to approximate action-value functions. Since there is no reward in state $s_{H+1}$, we let $f_{H+1}(s, a, b) = 0$ for all $s \in \mathcal{S}, a \in \mathcal{A}, b \in \mathcal{B}, f \in \mathcal{F}$.

To measure the size of $\mathcal{F}$, we use $|\mathcal{F}|$ to denote its cardinality when $\mathcal{F}$ is finite. For infinite function classes, we use $\epsilon$-covering number to measure its size, which is defined as follows:

**Definition 3** ($\epsilon$-covering number). *The $\epsilon$-covering number of $\mathcal{F}$, denoted by $\mathcal{N}_{\mathcal{F}}(\epsilon)$, is the minimum integer $n$ such that there exists a subset $\mathcal{F}' \subset \mathcal{F}$ with $|\mathcal{F}'| = n$ and for any $f \in \mathcal{F}$ there exists $f' \in \mathcal{F}'$ such that $\max_{h \in [H]} \|f_h - f'_h\|_\infty \leq \epsilon$.*

In addition to the size, we also need to impose some complexity assumption on the structure of the function class to achieve small generalization error. Here we introduce one of such structure complexity measures called Distributional Eluder (DE) dimension [25], which we will utilize in our subsequent analysis. First let us define independence between distributions:

**Definition 4** ($\epsilon$-independence between distributions). *Let $\mathcal{W}$ be a function class defined on $\mathcal{X}$, and $\rho, \rho_1, \cdots, \rho_n$ be probability measures over $\mathcal{X}$. We say $\rho$ is $\epsilon$-independent of $\{\rho_1, \cdots, \rho_n\}$ with respect to $\mathcal{W}$ if there exists $w \in \mathcal{W}$ such that $\sqrt{\sum_{i=1}^n (\mathbb{E}_{\rho_i}[w])^2} \leq \epsilon$ but $|\mathbb{E}_\rho[w]| > \epsilon$.*

From the definition we can see that a probability distribution $\rho$ is independent from $\{\rho_1, \cdots, \rho_n\}$ if there exists a discriminator function in $\mathcal{W}$ such that the function values are small at $\{\rho_1, \cdots, \rho_n\}$ while large at $\rho$. DE dimension is simply the length of the longest sequence of independent probability measures that the function class can discriminate:

**Definition 5** (Distributional Eluder (DE) dimension). *Let $\mathcal{W}$ be a function class defined on $\mathcal{X}$, and $\mathcal{Q}$ be a family of probability measures over $\mathcal{X}$. The distributional Eluder dimension $\dim_{\mathrm{DE}}(\mathcal{W}, \mathcal{Q}, \epsilon)$ is the length of the longest sequence $\{\rho_1, \cdots, \rho_n\} \subset \mathcal{Q}$ such that there exists $\epsilon' \geq \epsilon$ where $\rho_i$ is $\epsilon'$-independent of $\{\rho_1, \cdots, \rho_{i-1}\}$ for all $i \in [n]$.*

Eluder dimension, another commonly-used complexity measure proposed by [43], is a special case of DE dimension. If we choose $\mathcal{Q} = \{\delta_x(\cdot) | x \in \mathcal{X}\}$ where $\delta_x(\cdot)$ is the dirac measure centered at $x$, then the Eluder dimension can be formulated as

$$\dim_{\mathrm{E}}(\mathcal{W}, \epsilon) = \dim_{\mathrm{DE}}(\mathcal{W} - \mathcal{W}, \mathcal{Q}, \epsilon),$$

where $\mathcal{W} - \mathcal{W} = \{w_1 - w_2 : w_1, w_2 \in \mathcal{W}\}$. Many function classes in MDPs are known to have low Eluder dimension, including linear MDPs [28], generalized linear complete models [52] and kernel MDPs [25].

We also assume the existence of an auxiliary function class $\mathcal{G} = \mathcal{G}_1 \times \cdots \times \mathcal{G}_H$ ($\mathcal{G}_h \subseteq (\mathcal{S} \times \mathcal{A} \times \mathcal{B} \to [0, V_{\max}])$) to capture the results of applying Bellman operators on $\mathcal{F}$ as in [25, 27]. When $\mathcal{F}$ satisfies completeness (Assumption 3), we can simply choose $\mathcal{G} = \mathcal{F}$.

## 3  ALGORITHM: DORIS

**Policy revealing setting.** Recall that in decentralized policy learning setting, the player is also able to observe some information of the opponent, denoted by $\mathcal{J}_h$, aside from its own actions and rewards. There have been works studying the case where $\mathcal{J}_h = \emptyset$ [49] and $\mathcal{J}_h = b_h$ [27, 23] in two-player zero-sum games. However, their benchmark is the Nash value of the Markov game, i.e., $V_1^{\mu^* \times \nu^*}(s_1)$ where $\mu^* \times \nu^*$ is an NE, which is strictly weaker than our benchmark $\max_{\mu \in \Pi} \sum_{t=1}^K V_1^{\mu \times \nu^t}(s_1)$ in two-player zero-sum games. In fact, [33] has showed achieving low regrets under Definition 1 is exponentially hard in tabular cases when the opponent's policies are not revealed (see Appendix E.2 for details). Therefore in this paper we let $\mathcal{J}_h = \{b_h, \nu_h\}$ just like [33] and call this information structure *policy revealing* setting.

That said, even in policy revealing setting, the challenge of nonstationarity still exists because the opponent's policy can be adversarial and only gets revealed after the player plays a policy. Thus from the perspective of the player, the transition kernel $P_h^\nu(\cdot|s,a) := \mathbb{E}_{b \sim \nu_h(s)} P_h(\cdot|s,a,b)$ still changes in an unpredictable way across episodes. In addition, the problem of how to balance exploration and exploitation with general function approximation also remains due to the unknown transition probability. In this section we propose DORIS, an algorithm that is capable of handling both these challenges and achieving a $\sqrt{K}$ regret upper bound in policy revealing setting.

**Remark 2.** *When the opponent's policy is not revealed but changes slowly, we indeed can infer the opponent's policy approximately via the procedures in [39, 46] and this can be viewed as an \*\*approximate policy revealing condition\*\* in practice.*

**DORIS.** Intuitively, our algorithm is an actor-critic / mirror descent (Hedge) algorithm where each policy $\mu$ in $\Pi$ is regarded as an expert and the performance of each expert at episode $t$ is given by the value function of $V_1^{\mu \times \nu^t}(s_1)$. We call it Decentralized Optimistic hypeRpolicy mIrror deScent (DORIS). DORIS possesses three important features, whose details are shown in Algorithm 1:

- **Hyperpolicy and Hedge:** Motivated from the adversarial bandit literature [1, 20, 30] and no-regret learning works [33], DORIS maintains a distribution $p$ over the policies in $\Pi$, which we call *hyperpolicy*, to combat the nonstaionarity. The hyperpolicy is updated using Hedge, with the reward of each policy $\mu$ being an estimation of the value function $V_1^{\mu \times \nu^t}(s_1)$. This is equivalent to running mirror ascent algorithm over the policy space $\Pi$ with the gradient being $V_1^{\mu \times \nu^t}(s_1)$.

- **Optimism:** However, we do not have access to the exact value function since the transition probability is unknown, which forces us to deal with the exploration-exploitation tradeoff. Here we utilize the *Optimism in the Face of Uncertainty* principle [2, 28, 25, 27, 23] and choose our estimation $\overline{V}^t(\mu)$ to be optimistic with respect to the true value $V_1^{\mu \times \nu^t}(s_1)$. In this way DORIS will prefer policies with more uncertainty and thus encourage exploration in the Markov game.

- **Optimistic policy evaluation with general function approximation:** Finally we need to design an efficient method to obtain such optimistic estimation $\overline{V}^t(\mu)$ with general function approximation. Here we propose OptLSPE to accomplish this task. In short, OptLSPE constructs a confidence set for the target action-value function $Q^{\mu \times \nu}$ based on the player's local information and chooses an optimistic estimation from the confidence set, as shown in Algorithm 2. The construction of the confidence set utilizes the fact that $Q_h^{\mu \times \nu}$ satisfies the Bellman equation [41]:

$$Q_h^{\mu \times \nu}(s,a,b) = (\mathcal{T}_h^{\mu,\nu} Q_{h+1}^{\mu \times \nu})(s,a,b) := r_h(s,a,b) + \mathbb{E}_{s' \sim P_h(\cdot|s,a,b)}[Q_{h+1}^{\mu \times \nu}(s',\mu,\nu)],$$

where $Q_{h+1}^{\mu \times \nu}(s',\mu,\nu) = \mathbb{E}_{a' \sim \mu(\cdot|s'),b' \sim \nu(\cdot|s')}[Q_{h+1}^{\mu \times \nu}(s',a',b')]$. We call $\mathcal{T}_h^{\mu,\nu}$ the Bellman operator induced by $\mu \times \nu$ at the $h$-th step. Then the construction rule of $\mathcal{B}_\mathcal{D}(\mu,\nu)$ is based on least-squares policy evaluation with slackness $\beta$ as below:

$$\mathcal{B}_\mathcal{D}(\mu,\nu) \leftarrow \left\{ f \in \mathcal{F} : \mathcal{L}_\mathcal{D}(f_h, f_{h+1}, \mu, \nu) \leq \inf_{g \in \mathcal{G}} \mathcal{L}_\mathcal{D}(g_h, f_{h+1}, \mu, \nu) + \beta, \forall h \in [H] \right\}, \quad (3)$$

where $\mathcal{L}_\mathcal{D}$ is the empirical Bellman residuals on $\mathcal{D}$:

$$\mathcal{L}_\mathcal{D}(\xi_h, \zeta_{h+1}, \mu, \nu) = \sum_{(s_h, a_h, b_h, r_h, s_{h+1}) \in \mathcal{D}} [\xi_h(s_h, a_h, b_h) - r_h - \zeta_{h+1}(s_{h+1}, \mu, \nu)]^2.$$

---

**Algorithm 1 `DORIS`**

---

**Input**: learning rate $\eta$, confidence parameter $\beta$.
Initialize $p^1 \in \Delta_\Pi$ to be uniform over $\Pi$.
**for** $t = 1, \cdots, K$ **do**
    **Collect samples:**
    The player samples $\mu^t$ from $p^t$.
    The player runs $\mu^t$ against the opponent and collects $\mathcal{D}_t = \{s_1^t, a_1^t, b_1^t, r_1^t, \cdots, s_{H+1}^t\}$.
    **Update policy distribution:**
    The opponent reveals its policy $\nu^t$ to the player.
    $\overline{V}^t(\mu) \leftarrow \texttt{OptLSPE}(\mu, \nu^t, \mathcal{D}_{1:t-1}, \mathcal{F}, \mathcal{G}, \beta), \quad \forall \mu \in \Pi.$
    $p^{t+1}(\mu) \propto p^t(\mu) \cdot \exp(\eta \cdot \overline{V}^t(\mu)), \quad \forall \mu \in \Pi.$
**end for**

---

**Algorithm 2 `OptLSPE`$(\mu, \nu, \mathcal{D}, \mathcal{F}, \mathcal{G}, \beta)$**

---

**Construct** $\mathcal{B}_\mathcal{D}(\mu, \nu)$ based on $\mathcal{D}$ via (3).
**Select** $\bar{V} \leftarrow \max_{f \in \mathcal{B}_\mathcal{D}(\mu,\nu)} f(s_1, \mu, \nu).$
**return** $\bar{V}$.

---

**Decentralized Algorithm.** Here we want to highlight that `DORIS` is a decentralized algorithm because the player can run `DORIS` based only on its local information, i.e., $\{s_h, a_h, \mathcal{J}_h, r_h\}$, and we do not make any assumptions on the policies of the opponent. We also discuss the computational complexity of `DORIS` in Appendix E.3.

**Comparison with OPMD [33].** The hyperpolicy and Hedge part of `DORIS` indeed follows OPMD proposed by [33], which is an algorithm for no-regret learning in tabular MG. The novelty of `DORIS` lies in the new policy evaluation algorithm specially designed for the policy revealing setting that can tackle general function approximation. We also need to propose new techniques to analyze the performance of `DORIS`, which is more complicated than tabular cases. See Appendix E.1 for more detailed comparison.

### 3.1 `DORIS` IN SELF-PLAY SETTING

Apart from decentralized learning setting with a possibly adversarial opponent, we are also interested in the self-play setting where we can control all the agents and need to find an equilibrium for the $n$-agent general-sum Markov game. Inspired by the existing relationships between no-regret learning and CCEs in matrix games [17, 6, 10], a natural idea is to simply let all agents run `DORIS` independently. To achieve this, we assume each agent $i$ is given a value function class $\mathcal{F}_i = \mathcal{F}_{1,i} \times \cdots \times \mathcal{F}_{H,i}$ and an auxiliary function class $\mathcal{G}_i = \mathcal{G}_{1,i} \times \cdots \times \mathcal{G}_{H,i}$ as in `DORIS`, and run `DORIS` with learning rate $\eta_i$ and confidence parameter $\beta_i$ by viewing the other agents as its opponent. Suppose the policies played by agent $i$ during $K$ episodes are $\{\mu_i^t\}_{t=1}^K$, then we output the final joint policy as a uniform mixture:

$$\widehat{\pi} \sim \text{Unif}\Big(\Big\{\prod_{i \in [n]} \mu_i^1, \cdots, \prod_{i \in [n]} \mu_i^K\Big\}\Big).$$

See Algorithm 3 for more details.

**Remark 3.** *Algorithm 3 is also a decentralized algorithm since every agent runs their local algorithm independently without coordination. The only step that requires centralized control is the output process where all the agents need to share the same iteration index, which is also required in the existing decentralized algorithms [35, 26].*

## 4 THEORETICAL GUARANTEES

In this section we analyze the theoretical performance of `DORIS` in decentralized policy learning and self-play setting. We first introduce a new complexity measure for function classes, called Bellman Evaluation Eluder (BEE) dimension, and then illustrate the regret and sample complexity bounds based on this new measure.

### 4.1 BELLMAN EVALUATION ELUDER DIMENSION

Motivated from Bellman Eluder (BE) dimension in classic MDPs and its variants in MGs [25, 27, 23], we propose a new measure specifically tailored to the decentralized policy learning setting, called

Bellman Evaluation Eluder (BEE) dimension. First, for any function class $\mathcal{F}$, we define $(\mathcal{I} - \mathcal{T}_h^{\Pi,\Pi'})\mathcal{F}$ to be the Bellman residuals induced by the policies in $\Pi$ and $\Pi'$:

$$(\mathcal{I} - \mathcal{T}_h^{\Pi,\Pi'})\mathcal{F} := \{f_h - \mathcal{T}_h^{\mu,\nu} f_{h+1} : f \in \mathcal{F}, \mu \in \Pi, \nu \in \Pi'\}.$$

Then Bellman Evaluation Eluder (BEE) dimension is the DE dimension of the Bellman residuals induced by the policy class $\Pi$ and $\Pi'$ on function class $\mathcal{F}$:

**Definition 6.** *The $\epsilon$-Bellman Evaluation Eluder dimension of function class $\mathcal{F}$ on distribution family $\mathcal{Q}$ with respect to the policy class $\Pi \times \Pi'$ is defined as follows:*

$$\dim_{\mathrm{BEE}}(\mathcal{F}, \epsilon, \Pi, \Pi', \mathcal{Q}) := \max_{h \in [H]} \dim_{\mathrm{DE}}((\mathcal{I} - \mathcal{T}_h^{\Pi,\Pi'})\mathcal{F}, \mathcal{Q}_h, \epsilon).$$

BEE dimension is able to capture the generalization error of evaluating value function $V^{\mu \times \nu}$ where $\mu \in \Pi, \nu \in \Pi'$, which is one of the most essential tasks in decentralized policy space optimization as shown in DORIS. Similar to [25, 27, 23], we mainly consider two distribution families for $\mathcal{Q}$ here:

- $\mathcal{Q}^1 = \{\mathcal{Q}_h^1\}_{h \in [H]}$: the collection of all probability measures over $\mathcal{S} \times \mathcal{A} \times \mathcal{B}$ at each step when executing $(\mu, \nu) \in \Pi \times \Pi'$.

- $\mathcal{Q}^2 = \{\mathcal{Q}_h^2\}_{h \in [H]}$: the collection of all probability measures that put measure 1 on a single state-action pair $(s, a, b)$ at each step.

We also use $\dim_{\mathrm{BEE}}(\mathcal{F}, \epsilon, \Pi, \Pi')$ to denote $\min\{\dim_{\mathrm{BEE}}(\mathcal{F}, \epsilon, \Pi, \Pi', \mathcal{Q}^1), \dim_{\mathrm{BEE}}(\mathcal{F}, \epsilon, \Pi, \Pi', \mathcal{Q}^2)\}$ for simplicity in the following discussion.

**Relation with Eluder dimension.** To illustrate the generality of BEE dimension, we show that all function classes with low Eluder dimension also have low BEE dimension, as long as completeness (Assumption 3) is satisfied. More specifically, we have the following proposition and its proof is deferred to Appendix F:

**Proposition 1.** *Assume $\mathcal{F}$ satisfies completeness, i.e., $\mathcal{T}_h^{\mu,\nu} f_{h+1} \in \mathcal{F}_h, \forall f \in \mathcal{F}, \mu \in \Pi, \nu \in \Pi', h \in [H]$. Then for all $\epsilon > 0$, we have*

$$\dim_{\mathrm{BEE}}(\mathcal{F}, \epsilon, \Pi, \Pi') \leq \max_{h \in [H]} \dim_{\mathrm{E}}(\mathcal{F}_h, \epsilon). \tag{4}$$

Inequality (4) shows that BEE dimension is always upper bounded by Eluder dimension when completeness is satisfied. With Proposition 1, Appendix H validates that kernel Markov games (including tabular Markov games and linear Markov games) and generalized linear complete models all have small Bellman Evaluation Eluder Dimension. Furthermore, in this case the upper bound of BEE dimension does not depend on $\Pi$ and $\Pi'$, which is a desirable property when $\Pi$ and $\Pi'$ is large.

**Comparison with multi-agent BE dimension.** Jin et al. [27], Huang et al. [23] also propose a variant of BE dimension for Markov games called multi-agent BE dimension. However, this complexity measure and its analysis techniques are not applicable in our setting because DORIS is different from their algorithms in terms of confidence set construction and optimism. See Appendix E.1 for details.

## 4.2 DECENTRALIZED POLICY LEARNING REGRET

Next we present the regret analysis for DORIS in decentralized policy learning setting. For simplicity, we focus on finite $\Pi$ here:

**Assumption 1** (Finite player's policy class). *We assume $\Pi$ is finite.*

We consider two cases, the oblivious opponent (i.e., the opponent determines $\{\nu^t\}_{t=1}^K$ secretly before the game starts) and the adaptive opponent (i.e., the opponent determines its policy adaptively as the game goes on) separately. The difference between these two cases lies in the policy evaluation step in DORIS. The policy $\nu^t$ of an oblivious opponent does not depend on the collected dataset $\mathcal{D}_{1:t-1}$ and thus $V^{\mu,\nu^t}$ is easier to evaluate. However, for an adaptive opponent, $\nu^t$ will be chosen adaptively based on $\mathcal{D}_{1:t-1}$ and we need to introduce an additional union bound over $\Pi'$ when analyzing the evaluation error of $V^{\mu,\nu^t}$.

**Oblivious opponent.** To attain accurate value function estimation, we first need to introduce two standard assumptions, realizability and generalized completeness, on $\mathcal{F}$ and $\mathcal{G}$ [25, 27]. Here realizability refers to that all the true action value functions belong to $\mathcal{F}$ and generalized completeness means that $\mathcal{G}$ contains all the results of applying Bellman operator to the functions in $\mathcal{F}$.

**Assumption 2** (Realizability and generalized completeness). *Assume that for any $h \in [H], \mu \in \Pi, \nu \in \{\nu^1, \cdots, \nu^K\}, f_{h+1} \in \mathcal{F}_{h+1}$, we have $Q_h^{\mu \times \nu} \in \mathcal{F}_h, \mathcal{T}_h^{\mu,\nu} f_{h+1} \in \mathcal{G}_h$.*

**Remark 4.** *Some existing works [57, 23] assume the completeness assumption, which can also be generalized to our setting:*

**Assumption 3.** *Assume for any $h \in [H], \mu \in \Pi, \nu \in \Pi', f_{h+1} \in \mathcal{F}_{h+1}$, we have $\mathcal{T}_h^{\mu,\nu} f_{h+1} \in \mathcal{F}_h$.*

*We want to clarify that Assumption 3 is stronger than generalized completeness in Assumption 2 since if Assumption 3 holds, we can simply let $\mathcal{G} = \mathcal{F}$ to satisfy generalized completeness.*

Appendix G shows that realizability and generalized completeness are satisfied in many examples including tabular MGs, linear MGs and kernel MGs. With the above assumptions, we have Theorem 1 to characterize the regret of DORIS when the opponent is oblivious, whose proof sketch is deferred to Appendix I. To simplify writing, we use the following notations in Theorem 1:

$$d_{\text{BEE}} := \dim_{\text{BEE}}\big(\mathcal{F}, \sqrt{1/K}, \Pi, \Pi'\big), \quad \mathcal{N}_{\text{cov}} := \mathcal{N}_{\mathcal{F} \cup \mathcal{G}}(V_{\max}/K)KH.$$

**Theorem 1** (Regret of Oblivious Adversary). *Under Assumption 1,2, there exists an absolute constant $c$ such that for any $\delta \in (0,1], K \in \mathbb{N}$, if we choose $\beta = cV_{\max}^2 \log(\mathcal{N}_{\text{cov}}|\Pi|/\delta)$ and $\eta = \sqrt{\log|\Pi|/(KV_{\max}^2)}$ in DORIS, then with probability at least $1 - \delta$, we have:*

$$Regret(K) \leq \mathcal{O}\big(HV_{\max}\sqrt{Kd_{\text{BEE}}\log(\mathcal{N}_{\text{cov}}|\Pi|/\delta)}\big). \tag{5}$$

The $\sqrt{K}$ regret bound in Theorem 1 is consistent with the rate in tabular case [33] and suggests that the uniform mixture of the output policies $\{\mu^t\}_{t=1}^K$ is an $\epsilon$-approximate best policy in hindsight when $K = \widetilde{\mathcal{O}}(1/\epsilon^2)$. The complexity of the problem affects the regret through the covering number and the BEE dimension, implying that BEE dimension indeed captures the essence of this problem. Further, in oblivious setting, the regret bound in (5) does not depend on $\Pi'$ directly (the upper bound of the BEE dimension is also independent of $\Pi'$ in some special cases as shown in Proposition 1) and thus Theorem 1 can still hold when $\Pi'$ is infinite, as long as Assumptions 2 is satisfied.

**Adaptive Opponent.** In the adaptive setting, we first need to modify Assumption 2 to hold for all $\nu \in \Pi'$ since $\nu^t$ depends on the collected data (recall that $\Pi$ is the player's policy class and $\Pi'$ is the opponent's policy class):

**Assumption 4** (Uniform realizability and generalized completeness). *Assume that for any $h \in [H], \mu \in \Pi, \nu \in \Pi', f_{h+1} \in \mathcal{F}_{h+1}$, we have $Q_h^{\mu \times \nu} \in \mathcal{F}_h, \mathcal{T}_h^{\mu,\nu} f_{h+1} \in \mathcal{G}_h$.*

Further, as we have mentioned before, we need to introduce a union bound over the policies in $\Pi'$ in our analysis and thus we also assume $\Pi'$ to be finite for simplicity.

**Assumption 5** (Finite opponent's policy class). *We assume $\Pi'$ is finite.*

**Remark 5.** *It is straightforward to generalize our analysis to infinite $\Pi'$ by replacing $|\Pi'|$ with the covering number of $\Pi'$. However, the regret still depends on the size of $\Pi'$, which is not the case in tabular setting [33]. This dependency originates from our model-free type of policy evaluation algorithm (Algorithm 2) and is inevitable for DORIS in general. That said, when the Markov game has special structures (e.g., the Markov games in Appendix C and D), we can avoid this dependency.*

With the above assumptions, we have Theorem 2 to show that DORIS can still achieve sublinear regret in adaptive setting, whose proof is deferred to Section I:

**Theorem 2** (Regret of Adaptive Adversary). *Under Assumption 1,4,5, there exists an absolute constant $c$ such that for any $\delta \in (0,1], K \in \mathbb{N}$, choosing $\beta = cV_{\max}^2 \log(\mathcal{N}_{\text{cov}}|\Pi||\Pi'|/\delta)$ and $\eta = \sqrt{\log|\Pi|/(KV_{\max}^2)}$ in DORIS, then with probability at least $1 - \delta$ we have:*

$$Regret(K) \leq \mathcal{O}\big(HV_{\max}\sqrt{Kd_{\text{BEE}}\log(\mathcal{N}_{\text{cov}}|\Pi||\Pi'|/\delta)}\big). \tag{6}$$

We can see that in adaptive setting the regret also scales with $\sqrt{K}$, implying that DORIS can still find an $\epsilon$-approximate best policy in hindsight with $\widetilde{\mathcal{O}}(1/\epsilon^2)$ episodes even when the opponent is adaptive. Compared to Theorem 1, Theorem 2 has an additional $\log|\Pi'|$ in the upper bound (6), which comes from the union bound over $\Pi'$ in the analysis.

**Intuitions on the regret bounds.** The regrets in Theorem 1 and Theorem 2 can be decomposed to two parts, the online learning error incurred by Hedge and the cumulative value function estimation error incurred by OptLSPE. From the online learning literature [20], the online learning error is

$\mathcal{O}(V_{\max}\sqrt{K \log |\Pi|})$ by viewing the policies in $\Pi$ as experts and $\overline{V}^t(\mu)$ as the reward function of expert $\mu$. For the estimation error, we utilize BEE dimensions to bridge $\overline{V}^t(\mu^t) - V_1^{\pi^t}(s_1)$ with the function's empirical Bellman residuals on $\mathcal{D}_{1:t-1}$. This further incurs $\widetilde{\mathcal{O}}(V_{\max}\sqrt{Kd_{\mathrm{BEE}}})$ in the results. Our technical contribution mainly lies in bounding the cumulative value function estimation error with the newly proposed BEE dimensions, which is different from [25] where they focus on bounding the cumulative distance from the optimal value function.

**Comparison with existing works.** There have been works studying decentralized policy learning. However, most of them [7, 53, 49, 27, 23] only competes against the Nash value in a two-player zero-sum games, which is a much weaker baseline than ours. [33] can achieve $\sqrt{K}$ regret under Definition 1, but they are restricted in tabular cases and the bound becomes vacuous with more complicated cases like linear MGs and kernel MGs in Appendix H. DORIS is the first algorithm that can achieve $\sqrt{K}$ regret under Definition 1 with general function approximation and is capable of tackling all models with low BEE dimension, including linear MGs, kernel MGs and generalized linear complete models. More details are deferred to Appendix E.1.

### 4.3 SELF-PLAY SAMPLE COMPLEXITY

Our previous discussion assumes the opponent is arbitrary or even adversary. A natural question is to ask whether there are any additional guarantees if the player and opponent run DORIS simultaneously, which is exactly the self-play setting. The following corollary answers this question affirmatively and shows that Algorithm 3 can find an approximate CCE $\widehat{\pi}$ efficiently:

**Corollary 1.** *Suppose Assumption 1,4 hold for all the agents $i$ and its corresponding $\mathcal{F}_i, \mathcal{G}_i, \Pi_i, \Pi_{-i}$. Then for any $\delta \in (0, 1], \epsilon > 0$, if we choose*

$$K \geq \mathcal{O}\left( H^2 V_{\max}^2 \cdot \max_{i \in [n]}\left\{ d_{\mathrm{BEE},i} \cdot \left( \log \mathcal{N}_{\mathrm{cov},i} + \sum_{j=1}^{n} \log |\Pi_j| + \log(n/\delta) \right) \right\} \middle/ \epsilon^2 \right), \quad (7)$$

*where $d_{\mathrm{BEE},i}$ and $\mathcal{N}_{\mathrm{cov},i}$ are defined respectively as*

$$d_{\mathrm{BEE},i} := \dim_{\mathrm{BEE}}\left(\mathcal{F}_i, \sqrt{1/K}, \Pi_i, \Pi_{-i}\right), \quad \mathcal{N}_{\mathrm{cov},i} := \mathcal{N}_{\mathcal{F}_i \cup \mathcal{G}_i}(V_{\max}/K)KH,$$

*and set $\beta_i = cV_{\max}^2 \log(\mathcal{N}_{\mathrm{cov},i}|\Pi_i||\Pi_{-i}|n/\delta), \eta_i = \sqrt{\log |\Pi_i|/(KV_{\max}^2)}$, then with probability at least $1 - \delta$, $\widehat{\pi}$ is $\epsilon$-approximate CCE.*

The proof is deferred to Appendix K. Corollary 1 shows that if we run DORIS independently for each agent, we are able to find an $\epsilon$-approximate CCE with $\widetilde{\mathcal{O}}(1/\epsilon^2)$ samples. This can be regarded as a counterpart in Markov games to the classic connection between no-regret learning algorithms and equilibria in matrix games. However, this guarantee does not hold if an algorithm can only achieve low regrets with respect to the Nash values.

**Avoiding curse of multiagents.** The sample complexity in (7) avoids exponential scaling with the number of agents $n$ and only scales with $\max_{i \in [n]} d_{\mathrm{BEE},i}$, $\max_{i \in [n]} \mathcal{N}_{\mathrm{cov},i}$ and $\sum_{j=1}^{n} \log |\Pi_j|$, suggesting that statistically Algorithm 3 is able to escape the *curse-of-multiagents* problem in the literature [26]. Nevertheless, the input dimension of functions in $\mathcal{F}_i$ and $\mathcal{G}_i$ may scale with the number of agents, leading to the computational inefficiency of OptLSPE. We comment that finding computational efficient algorithms is beyond the scope of this paper and we leave it to future works.

**Comparison with existing algorithms.** There have been many works studying how to find equilibria in Markov games. However, most of them are focused on centralized two-player zero-sum games [3, 56, 27, 23] rather than decentralized algorithms. For decentralized algorithms, existing literature mainly handle with potential Markov games [60, 32, 12] and two-player zero-sum games [11, 44, 54]. [35, 26] are able to tackle decentralized multi-agent general-sum Markov games while their algorithms are restricted to tabular cases. Algorithm 3 can deal with more general cases with function approximation and policy classes in multi-agent general-sum games. Furthermore, compared to the above works, DORIS has an additional advantage of robustness to adversaries since all the benign agents can exploit the opponents and achieve no-regret learning.

**Extensions.** Although Theorem 1, Theorem 2 and Corollary 1 are aimed at Markov games, DORIS can be applied to a much larger scope of problems. Two such problems are finding the optimal policy in constrained MDPs and vector-valued MDPs. We will investigate these two problems in Appendix C and D, where we demonstrate how to convert such problems into Markov games with a fictitious opponent by duality so that DORIS is ready to use.

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

## A  DORIS IN SELF-PLAY SETTING

Here we present the pseudocode of Algorithm 3.

---

**Algorithm 3 DORIS** in self-play setting

---

**Input**: learning rate $\{\eta_i\}_{i=1}^n$, confidence parameter $\{\beta_i\}_{i=1}^n$.
Initialize $p_i^1 \in \Delta_{|\Pi_i|}$ to be uniform over $\Pi_i$ for all $i \in [n]$.
**for** $t = 1, \cdots, K$ **do**
   **Collect samples:**
   Agent $i$ samples $\mu_i^t$ from $p_i^t$.
   Run $\mu^t = \prod_{i=1}^n \mu_i^t$ and collect $\mathcal{D}_{t,i} = \{s_1^t, \boldsymbol{a}_1^t, r_{1,i}^t, \cdots, s_{H+1}^t\}$ for each agent $i$.
   **Update policy distribution:**
   All agents reveal their policies $\mu_i^t$.
   $\overline{V}_i^t(\mu_i) \leftarrow \texttt{OptLSPE}(\mu_i, \mu_{-i}^t, \mathcal{D}_{1:t-1,i}, \mathcal{F}_i, \mathcal{G}_i, \beta_i), \quad \forall \mu_i \in \Pi_i, i \in [n].$
   $p_i^{t+1}(\mu_i) \propto p_i^t(\mu_i) \cdot \exp(\eta_i \cdot \overline{V}_i^t(\mu_i)), \quad \forall \mu_i \in \Pi_i, i \in [n].$
**end for**
**Output**: $\widehat{\pi} \sim \mathrm{Unif}(\{\prod_{i \in [n]} \mu_i^1, \cdots, \prod_{i \in [n]} \mu_i^K\}).$

---

## B  RELATED WORKS

In this section we supplement the related literature.

**Decentralized learning with an adversarial opponent.**   There have been a few works studying decentralized policy learning in the presence of a possibly adversarial opponent. [7] proposes R-max and is able to attain an average game value close to the Nash value in tabular MGs. More recently, [53, 49] improve the regret bounds in tabular cases and [27, 23] extend the results to general function approximation setting. However, these works only compete against the Nash value of the game and are unable to exploit the opponent. A more related paper is [33], which develops a provably efficient algorithm that achieves a sublinear regret against the best fixed policy in hindsight. But there results are only limited to the tabular case. Our work extends the results in [33] to the setting with general function approximation, which requires novel technical analysis.

**Finding equilibria in self-play Markov games.**   Our work is closely related to the recent literature on finding equilibria in Markov games via reinforcement learning. Most of the existing works focus on two-player zero-sum games and consider centralized algorithms with unknown model dynamics. For example, [53, 3] utilize optimism to tackle the exploration-expoitation tradeoff and find Nash equilibria in tabular cases, and [56, 27, 23] extend the results to linear and general function approximation setting. Furthermore, under the decentralized setting with well-explored data, [11, 60, 44, 54, 32, 12] utilize independent policy gradient algorithms to deal with potential Markov games and two-player zero-sum games. Meanwhile, under the online setting, [4, 35, 26] have designed algorithms named V-learning, which are able to find CCE in multi-agent general-sum games. However, there results are only limited to the tabular case.

**Constrained Markov decision process.**   [15, 13] propose a series of primal-dual algorithms for CMDPs which achieve $\sqrt{K}$ bound on regrets and constraint violations in tabular and linear approximation cases. [34] reduces the constraint violation to $\widetilde{\mathcal{O}}(1)$ by adding slackness to the algorithm and achieves zero violation when a strictly safe policy is known; [55] further avoids such requirement with the price of worsened regrets. Nevertheless, these improvements are only discussed in the tabular case.

**Approchability for vector-valued Markov decision process.**   [36] first introduces the approchability task for VMDPs but does not provide an algorithm with polynomial sample complexity. Then [58] proposes a couple of primal-dual algorithms to solve this task and achieves a $\widetilde{\mathcal{O}}(\epsilon^{-2})$ sample complexity in the tabular case. More recently, [37] utilizes reward-free reinforcement learning to

tackle the problem and studies both the tabular and linear approximation cases, achieving roughly the same sample complexity as [58].

## C  EXTENSION: CONSTRAINED MARKOV DECISION PROCESS

Although `DORIS` is designed to solve Markov games, there are quite a lot of other problems which `DORIS` can tackle with small adaptation. In this section we investigate an important scenario in practice called constrained Markov decision process (CMDP). By converting CMDPs into a maximin problem via Lagrangian multiplier, we will be able to view it as a zero-sum Markov game and apply `DORIS` readily.

**Constrained Markov decision process.** Consider the Constrained Markov Decision Process (CMDP) [13] $\mathcal{M}_{\text{CMDP}} = (\mathcal{S}, \mathcal{A}, \{P_h\}_{h=1}^H, \{r_h\}_{h=1}^H, \{g_h\}_{h=1}^H, H)$ where $\mathcal{S}$ is the state space, $\mathcal{A}$ is the action space, $H$ is the length of each episode, $P_h : \mathcal{S} \times \mathcal{A} \to \Delta(\mathcal{S})$ is the transition function at the $h$-th step, $r_h : \mathcal{S} \times \mathcal{A} \to \mathbb{R}_+$ is the reward function and $g_h : \mathcal{S} \times \mathcal{A} \to [0,1]$ is the utility function at the $h$-th step. We assume the reward $r_h$ is also bounded in $[0,1]$ for simplicity and thus $V_{\max} = H$. Then given a policy $\mu = \{\mu_h : \mathcal{S} \to \Delta_{\mathcal{A}}\}_{h \in [H]}$, we can define the value function $V_{r,h}^\mu$ and action-value function $Q_{r,h}^\mu$ with respect to the reward function $r$ as follows:

$$V_{r,h}^\mu(s) = \mathbb{E}_\mu\left[\sum_{l=h}^H r_l(s_l, a_l)\middle| s_h = s\right], Q_{r,h}^\mu(s,a) = \mathbb{E}_\mu\left[\sum_{l=h}^H r_l(s_l, a_l)\middle| s_h = s, a_h = a\right].$$

The value function $V_{g,h}^\mu$ and action-value function $Q_{g,h}^\mu$ with respect to the utility function $g$ can be defined similarly. Another related concept is the state-action visitation distribution, which can be defined as

$$d_h^\mu(s,a) = \Pr_\mu[(s_h, a_h) = (s,a)],$$

where $\Pr_\mu$ denotes the distribution of the trajectory induced by executing policy $\mu$ in $\mathcal{M}_{\text{CMDP}}$.

**Learning objective.** In CMDPs, the player aims to solve a constrained problem where the objective function is the expected total rewards and the constraint is on the expected total utilities:

---

**Problem 1: Optimization problem of CMDP**

$$\max_{\mu \in \Pi} V_{r,1}^\mu(s_1) \quad \text{subject to} \quad V_{g,1}^\mu(s_1) \geq b, \tag{8}$$

where $b \in (0, H]$ to avoid triviality.

---

Denote the optimal policy for (8) by $\mu_{\text{CMDP}}^*$, then the regret can be defined as the performance gap with respect to $\mu_{\text{CMDP}}^*$:

$$\text{Regret}(K) = \sum_{t=1}^K \left(V_{r,1}^{\mu_{\text{CMDP}}^*}(s_1) - V_{r,1}^{\mu^t}(s_1)\right). \tag{9}$$

However, since utility information is only revealed after a policy is decided, it is impossible for each policy to satisfy the constraints. Therefore, like [13], we allow each policy to violate the constraint in each episode and focus on minimizing total constraint violations over $K$ episodes:

$$\text{Violation}(K) = \left[\sum_{t=1}^K \left(b - V_{g,1}^{\mu^t}(s_1)\right)\right]_+. \tag{10}$$

Achieving sublinear violations in (10) implies that if we sample a policy uniformly from $\{\mu^t\}_{t=1}^K$, its constraint violation can be arbitrarily small given large enough $K$. Therefore, if an algorithm can achieve sublinear regret in (9) and sublinear violations in (10) at the same time, this algorithm will be able to find a good approximate policy to $\mu_{\text{CMDP}}^*$.

## C.1 ALGORITHM: DORIS-C

To solve Problem 1 with DORIS, we first need to convert it into a Markov game. A natural idea is to apply the Lagrangian multiplier $Y \in \mathbb{R}_+$ to Problem 1, which brings about the equivalent maximin problem below:

$$\max_{\mu \in \Pi} \min_{Y \geq 0} \mathcal{L}_{\mathrm{CMDP}}(\mu, Y) := V_{r,1}^{\mu}(s_1) + Y(V_{g,1}^{\mu}(s_1) - b). \tag{11}$$

Although Problem 1 is non-concave in $\mu$, there have been works indicating that strong duality still holds for Problem 1 when the policy class is described by a good parametrization [40]. Therefore, here we assume strong duality holds and it is straightforward to generalize our analysis to the case where there exists a duality gap:

**Assumption 6** (Strong duality). *Assume strong duality holds for Problem 1, i.e.,*

$$\max_{\mu \in \Pi} \min_{Y \geq 0} \mathcal{L}_{\mathrm{CMDP}}(\mu, Y) = \min_{Y \geq 0} \max_{\mu \in \Pi} \mathcal{L}_{\mathrm{CMDP}}(\mu, Y) \tag{12}$$

**Remark 6.** *One example case where strong duality (12) holds is when policy class $\Pi$ satisfies global realizability. Let $\mu_{glo}^*$ denote the solution to $\max_{\mu_h(\cdot|s) \in \Delta_{\mathcal{A}}} \min_{Y \geq 0} \mathcal{L}_{\mathrm{CMDP}}(\mu, Y)$. [13] shows that $\max_{\mu \in (\Delta_{\mathcal{A}})^{|\mathcal{S}|H}} \min_{Y \geq 0} \mathcal{L}_{\mathrm{CMDP}}(\mu, Y)$ satisfies strong duality, and thus as long as $\mu_{glo}^* \in \Pi$, Problem 1 also has strong duality.*

Further, let $D(Y) := \max_{\mu \in \Pi} \mathcal{L}_{\mathrm{CMDP}}(\mu, Y)$ denote the dual function and suppose the optimal dual variable is $Y^* = \arg\min_{Y \geq 0} D(Y)$. To ensure $Y^*$ is bounded, we need to assume the standard Slater's Condition holds:

**Assumption 7.** *There exists $\lambda_{\mathrm{sla}} > 0$ and $\widetilde{\mu} \in \Pi$ such that $V_{g,1}^{\widetilde{\mu}}(s_1) \geq b + \lambda_{\mathrm{sla}}$.*

Then the following lemma shows that Assumption 7 implies bounded optimal dual variable, whose proof is deferred to Appendix L.1:

**Lemma 1.** *Suppose Assumption 6,7 hold, then we have $0 \leq Y^* \leq H/\lambda_{\mathrm{sla}}$.*

Now we are ready to adapt DORIS into a primal-dual algorithm to solve Problem 1. Notice that the maximin problem (11) can be viewed as a zero-sum Markov game where the player's policy is $\mu$ and the reward function for the player is $r_h(s, a) + Y g_h(s, a)$. The opponent's action is $Y \in \mathbb{R}_+$ which remains the same throughout a single episode. With this formulation, we can simply run DORIS on the player, assuming the player is given function classes $\{\mathcal{F}^r, \mathcal{G}^r\}$ and $\{\mathcal{F}^g, \mathcal{G}^g\}$ to approximate $Q_{r,h}^{\mu}$ and $Q_{g,h}^{\mu}$ respectively. In the meanwhile, we run online projected gradient descent on the opponent so that its action $Y$ can capture the total violation so far.

This new algorithm is called DORIS-C and shown in Algorithm 4. It consists of the three steps below in each iteration. For the policy evaluation task in the second step, DORIS-C runs a single-agent version of OptLSPE to estimate $V_{r,1}^{\mu}(s_1)$ and $V_{g,1}^{\mu}(s_1)$ separately, which is essential for DORIS-C to deal with the infinity of the opponent's policy class, i.e., $\mathbb{R}_+$.

- The player plays a policy $\mu^t$ sampled from its hyperpolicy $p^t$ and collects a trajectory.

- The player runs OptLSPE-C to obtain optimistic value function estimations $\overline{V}_r^t(\mu), \overline{V}_g^t(\mu)$ for all $\mu \in \Pi$ and updates the hyperpolicy using Hedge with the rewards being $\overline{V}_r^t(\mu) + Y_t \overline{V}_g^t(\mu)$. The construction rule for $\mathcal{B}_{\mathcal{D}}(\mu)$ is still based on relaxed least-squares policy evaluation:

$$\mathcal{B}_{\mathcal{D}}(\mu) \leftarrow \{f \in \mathcal{F} : \mathcal{L}_{\mathcal{D}}(f_h, f_{h+1}, \mu) \leq \inf_{g \in \mathcal{G}} \mathcal{L}_{\mathcal{D}}(g_h, f_{h+1}, \mu) + \beta, \forall h \in [H]\}, \tag{13}$$

where $\mathcal{L}_{\mathcal{D}}$ is the empirical Bellman residuals on $\mathcal{D}$:

$$\mathcal{L}_{\mathcal{D}}(\xi_h, \zeta_{h+1}, \mu) = \sum_{(s_h, a_h, x_h, s_{h+1}) \in \mathcal{D}} [\xi_h(s_h, a_h) - x_h - \zeta_{h+1}(s_{h+1}, \mu)]^2.$$

- The dual variable is updated using online projected gradient descent.

---

**Algorithm 4 `DORIS-C`**

---

**Input**: learning rate $\eta, \alpha$, confidence parameter $\beta_r, \beta_g$, projection length $\chi$.

Initialize $p^1 \in \mathbb{R}^{|\Pi|}$ to be uniform over $\Pi$, $Y_1 \leftarrow 0$.

**for** $t = 1, \cdots, K$ **do**

  **Collect samples:**

  The player samples $\mu^t$ from $p^t$.

  Run $\mu^t$ and collect $\mathcal{D}_t^r = \{s_1^t, a_1^t, r_1^t, \cdots, s_{H+1}^t\}, \mathcal{D}_t^g = \{s_1^t, a_1^t, g_1^t, \cdots, s_{H+1}^t\}$.

  **Update policy distribution:**

  $\overline{V}_r^t(\mu) \leftarrow \texttt{OptLSPE-C}(\mu, \mathcal{D}_{1:t-1}^r, \mathcal{F}^r, \mathcal{G}^r, \beta_r), \quad \forall \mu \in \Pi.$

  $\overline{V}_g^t(\mu) \leftarrow \texttt{OptLSPE-C}(\mu, \mathcal{D}_{1:t-1}^g, \mathcal{F}^g, \mathcal{G}^g, \beta_g), \quad \forall \mu \in \Pi.$

  $p^{t+1}(\mu) \propto p^t(\mu) \cdot \exp(\eta \cdot (\overline{V}_r^t(\mu) + Y_t \overline{V}_g^t(\mu))), \quad \forall \mu \in \Pi.$

  **Update dual variable:**

  $Y_{t+1} \leftarrow \text{Proj}_{[0,\chi]}(Y_t + \alpha(b - \overline{V}_g^t(\mu^t))).$

**end for**

---

**Algorithm 5 `OptLSPE-C`$(\mu, \mathcal{D}, \mathcal{F}, \mathcal{G}, \beta)$**

---

**Construct** $\mathcal{B}_{\mathcal{D}}(\mu)$ based on $\mathcal{D}$ via (13).

**Select** $\bar{V} \leftarrow \max_{f \in \mathcal{B}_{\mathcal{D}}(\mu)} f(s_1, \mu).$

**return** $\bar{V}$.

---

## C.2 THEORETICAL GUARANTEES

Next we provide the regret and constraint violation bounds for `DORIS-C`. Here we also consider the case where $\Pi$ is finite, i.e., Assumption 1 is true. However, we can see that here the opponent is adaptive and its policy class is infinite, suggesting that Assumption 5 is violated. Fortunately, since the opponent only affects the reward function, the player can simply first estimate $V_{r,1}^\mu(s_1)$ and $V_{g,1}^\mu(s_1)$ respectively and then use their weighted sum to approximate the target value function $V_{r,1}^\mu(s_1) + Y \cdot V_{g,1}^\mu(s_1)$. In this way, `DORIS-C` circumvents introducing a union bound on $Y$ and thus can work even when the number of possible values for $Y$ is infinite.

We also need to introduce the realizability and general completeness assumptions on the function classes as before:

**Assumption 8** (Realizability and generalized completeness in CMDP). *Assume that for any $h \in [H], \mu \in \Pi, f_{h+1}^r \in \mathcal{F}_{h+1}^r, f_{h+1}^g \in \mathcal{F}^g$, we have*

$$Q_{r,h}^\mu \in \mathcal{F}_h^r, Q_{g,h}^\mu \in \mathcal{F}_h^g, \mathcal{T}_h^{\mu,r} f_{h+1}^r \in \mathcal{G}_h^r, \mathcal{T}_h^{\mu,g} f_{h+1}^g \in \mathcal{G}_h^g. \tag{14}$$

Here $\mathcal{T}_h^{\mu,r}$ is the Bellman operator at step $h$ with respect to $r$:

$$(\mathcal{T}_h^{\mu,r} f_{h+1})(s,a) = r_h(s,a) + \mathbb{E}_{s' \sim P(\cdot|s,a)} f_{h+1}(s', \mu),$$

where $f_{h+1}(s', \mu) = \mathbb{E}_{a' \sim \mu(\cdot|s)}[f_{h+1}(s', a')]$. $\mathcal{T}_h^{\mu,g}$ is defined similarly. We can see that (14) simply says that all the action value functions with respect to $r$ ($g$) belong to $\mathcal{F}^r$ ($\mathcal{F}^g$) and $\mathcal{G}^r$ ($\mathcal{G}^g$) contains all the results of applying Bellman operator with respect to $r$ ($g$) to the functions in $\mathcal{F}^r$ ($\mathcal{F}^g$).

In addition, as a simplified case of Definition 6, BEE dimension for single-agent setting can be defined as follows:

**Definition 7.** *The single-agent $\epsilon$-Bellman Evaluation Eluder dimension of function class $\mathcal{F}$ on distribution family $\mathcal{Q}$ with respect to the policy class $\Pi$ and the reward function $r$ is defined as follows:*

$$\dim_{\text{BEE}}(\mathcal{F}, \epsilon, \Pi, r, \mathcal{Q}) := \max_{h \in [H]} \dim_{\text{DE}}((\mathcal{I} - \mathcal{T}_h^{\Pi,r})\mathcal{F}, \mathcal{Q}_h, \epsilon),$$

*where $(\mathcal{I} - \mathcal{T}_h^{\Pi,r})\mathcal{F} := \{f_h - \mathcal{T}_h^{\mu,r} f_{h+1} : f \in \mathcal{F}, \mu \in \Pi\}$.*

We also let $\dim_{\text{BEE}}(\mathcal{F}, \epsilon, \Pi, r)$ denote $\min\{\dim_{\text{BEE}}(\mathcal{F}, \epsilon, \Pi, r, \mathcal{Q}^1), \dim_{\text{BEE}}(\mathcal{F}, \epsilon, \Pi, r, \mathcal{Q}^2)\}$ as before. $\dim_{\text{BEE}}(\mathcal{F}, \epsilon, \Pi, g, \mathcal{Q})$ and $\dim_{\text{BEE}}(\mathcal{F}, \epsilon, \Pi, g)$ are defined similarly but with respect to the utility function $g$.

Now we can present Theorem 3 which shows that `DORIS-C` is capable of achieving sublinear regret and constraint violation for Problem 1. We also use the following notations to simplify writing:

$$d_{\text{BEE},r} := \dim_{\text{BEE}}\big(\mathcal{F}^r, \sqrt{1/K}, \Pi, r\big), \quad \mathcal{N}_{\text{cov},r} := \mathcal{N}_{\mathcal{F}^r \cup \mathcal{G}^r}(H/K)KH,$$

$$d_{\text{BEE},g} := \dim_{\text{BEE}}\big(\mathcal{F}^g, \sqrt{1/K}, \Pi, g\big), \quad \mathcal{N}_{\text{cov},r} := \mathcal{N}_{\mathcal{F}^g \cup \mathcal{G}^g}(H/K)KH.$$

**Theorem 3.** *Under Assumption 6,7,1,8, there exists an absolute constant $c$ such that for any $\delta \in (0, 1]$, $K \in \mathbb{N}$, if we choose $\beta_r = cH^2 \log(\mathcal{N}_{\text{cov},r}|\Pi|/\delta)$, $\beta_g = cH^2 \log(\mathcal{N}_{\text{cov},g}|\Pi|/\delta)$, $\alpha = 1/\sqrt{K}$, $\chi = 2H/\lambda_{\text{sla}}$ and $\eta = \sqrt{\log|\Pi|/(K(\chi+1)^2H^2)}$ in `DORIS-C`, then with probability at least $1-\delta$, we have:*

$$Regret(K) \leq \mathcal{O}\bigg(\bigg(H^2 + \frac{H^2}{\lambda_{\text{sla}}}\bigg)\sqrt{Kd_{\text{BEE},r}\log\big(\mathcal{N}_{\text{cov},r}|\Pi|/\delta\big)}\bigg), \tag{15}$$

$$Violation(K) \leq \mathcal{O}\bigg(\bigg(H^2 + \frac{H}{\lambda_{\text{sla}}}\bigg)\sqrt{K\epsilon_{BEE}}\bigg), \tag{16}$$

*where*

$$\epsilon_{BEE} = \max\Big\{d_{\text{BEE},r}\log\big(\mathcal{N}_{\text{cov},r}|\Pi|/\delta\big), d_{\text{BEE},g}\log\big(\mathcal{N}_{\text{cov},g}|\Pi|/\delta\big)\Big\}.$$

The bounds in (15) and (16) show that both the regret and constraint violation of `DORIS-C` scale with $\sqrt{K}$. This implies that for any $\epsilon > 0$, if $\widehat{\mu}$ is sampled uniformly from $\{\mu^t\}_{t=1}^K$ and $K \geq \widetilde{\mathcal{O}}(1/\epsilon^2)$, $\widehat{\mu}$ will be an $\epsilon$ near-optimal policy with high probability in the sense that

$$V_{r,1}^{\widehat{\mu}}(s_1) \geq V_{r,1}^{\mu_{\text{CMDP}}^*}(s_1) - \epsilon, \qquad V_{g,1}^{\widehat{\mu}}(s_1) \geq b - \epsilon.$$

In addition, compared to the results in Theorem 1 and Theorem 2, (15) and (16) have an extra term scaling with $1/\lambda_{\text{sla}}$. This is because `DORIS-C` is a primal-dual algorithm and $\lambda_{\text{sla}}$ characterizes the regularity of this constrained optimization problem.

The proof of the regret bound is similar to Theorem 1 and Theorem 2 by viewing $V_{r,1}^{\mu}(s_1) + YV_{g,1}^{\mu}(s_1)$ as the target value function and decomposing the regret into cumulative estimation error and online learning error. To bound the constraint violation, we need to utilize the strong duality and the property of online projected gradient descent. See Appendix L for more details.

**Comparison with existing algorithms.** There has been a line of works studying the exploration and exploitation in CMDPs. [15, 13] propose a series of algorithms which can achieve $\sqrt{K}$ bound on regrets and constraint violations. However, they focus on tabular cases or linear function approximation and do not consider policy classes while `DORIS-C` can deal with nonlinear function approximation and policy classes. As an interesting follow-up, [34] reduces the constraint violation to $\widetilde{\mathcal{O}}(1)$ by adding slackness to the algorithm and achieves zero violation when a strictly safe policy is known; [55] further avoids such requirement with the price of worsened regrets. However, these improvements are all limited in tabular cases and we leave the consideration of their general function approximation counterpart to future works.

## D  EXTENSION: VECTOR-VALUED MARKOV DECISION PROCESS

Another setting where `DORIS` can play a role is the approachability task for vector-valued Markov decision process (VMDP) [36, 58, 37]. Similar to CMDP, we convert it into a zero-sum Markov game by Fenchel's duality and then adapt `DORIS` properly to solve it.

**Vector-valued Markov decision process.** Consider the Vector-valued Markov decision process (VMDP) [58] $\mathcal{M}_{\text{VMDP}} = (\mathcal{S}, \mathcal{A}, \{P_h\}_{h=1}^H, \boldsymbol{r}, H)$ where $\boldsymbol{r} = \{\boldsymbol{r}_h : \mathcal{S} \times \mathcal{A} \to [0, 1]^d\}_{h=1}^H$ is a collection of $d$-dimensional reward functions and the rest of the components are defined the same as in Section C. Then given a policy $\mu \in \Pi$, we can define the corresponding $d$-dimensional value function $\boldsymbol{V}_h^{\mu} : \mathcal{S} \to [0, H]^d$ and action-value function $\boldsymbol{Q}_h^{\mu} : \mathcal{S} \times \mathcal{A} \to [0, H]^d$ as follows:

$$\boldsymbol{V}_h^{\mu}(s) = \mathbb{E}_{\mu}\bigg[\sum_{l=h}^H \boldsymbol{r}_l(s_l, a_l)\bigg|s_h = s\bigg], \quad \boldsymbol{Q}_h^{\mu}(s, a) = \mathbb{E}_{\mu}\bigg[\sum_{l=h}^H \boldsymbol{r}_l(s_l, a_l)\bigg|s_h = s, a_h = a\bigg].$$

**Learning objective.** In this paper we study the approachability task [36] in VMDP where the player needs to learn a policy whose expected cumulative reward vector lies in a convex target set $\mathcal{C}$. We consider a more general agnostic version [58, 37] where we do not assume the existence of such policies and the player learns to minimize the Euclidean distance between expected reward and the target set $\mathcal{C}$:

---

**Problem 2: Approachability for VMDP**

$$\min_{\mu \in \Pi} \text{dist}(\boldsymbol{V}_1^\mu(s_1), \mathcal{C}),$$

where $\text{dist}(\boldsymbol{x}, \mathcal{C})$ is the Euclidean distance between point $\boldsymbol{x}$ and set $\mathcal{C}$.

---

The approachability for VMDP is a natural objective in multi-task reinforcement learning where each dimension of the reward can be regarded as a task. It is important in many practical domains such as robotics, autonomous vehicles and recommendation systems [58]. Therefore, finding the optimal policy for Problem 2 efficiently is of great significance in modern reinforcement learning.

## D.1    ALGORITHM: DORIS-V

To deal with Probelm 2, we first convert Problem 2 into a Markov game as we have done in Appendix C. By Fenchel's duality of the distance function, we know Problem 2 is equivalent to the following minimax problem:

$$\min_{\mu \in \Pi} \max_{\boldsymbol{\theta} \in \mathbb{B}(1)} \mathcal{L}_{\text{VMDP}}(\mu, \boldsymbol{\theta}) := \langle \boldsymbol{\theta}, \boldsymbol{V}_1^\mu(s_1) \rangle - \max_{\boldsymbol{x} \in \mathcal{C}} \langle \boldsymbol{\theta}, \boldsymbol{x} \rangle,$$

where $\mathbb{B}(r)$ is the $d$-dimensional Euclidean ball of radius $r$ centered at the origin. Regarding $\mu$ as the player's policy and $\boldsymbol{\theta}$ as the opponent, we can again view this minimax problem as a Markov game where the reward function for the player is $\langle \boldsymbol{\theta}, \boldsymbol{r}_h(s, a) \rangle$. Consider the general function approximation case where the player is given function classes $\mathcal{F} := \{\mathcal{F}_h^j\}_{h,j=1}^{H,d}, \mathcal{G} := \{\mathcal{G}_h^j\}_{h,j=1}^{H,d}$ to approximate $\boldsymbol{Q}_h^\mu$ ($\mathcal{F}_h^j$ and $\mathcal{G}_h^j$ are the $j$-th dimension of $\mathcal{F}_h$ and $\mathcal{G}_h$), we can run DORIS for the player while the opponent will update $\boldsymbol{\theta}$ via online projected gradient ascent just like DORIS-C.

We call this new algorithm DORIS-V, which is shown in Algorithm 6 and also consists of three steps in each iteration. For the policy evaluation task here, we apply OptLSPE-V and construct a confidence set for each dimension of the function class separately, and let the final confidence set be their intersection. Therefore the construction rule for $\mathcal{B}_\mathcal{D}(\mu)$ is given as:

$$\mathcal{B}_\mathcal{D}(\mu) \leftarrow \{f \in \mathcal{F} : \mathcal{L}_{\mathcal{D}^j}(f_h^j, f_{h+1}^j, \mu) \leq \inf_{g \in \mathcal{G}} \mathcal{L}_{\mathcal{D}^j}(g_h^j, f_{h+1}^j, \mu) + \beta, \forall h \in [H], j \in [d]\}, \quad (17)$$

where for any $j \in [d]$ and $h \in [H]$,

$$\mathcal{L}_{\mathcal{D}^j}(\xi_h^j, \zeta_{h+1}^j, \mu) = \sum_{(s_h, a_h, r_h^j, s_{h+1}) \in \mathcal{D}} [\xi_h^j(s_h, a_h) - r_h^j - \zeta_{h+1}^j(s_{h+1}, \mu)]^2,$$

and $r_h^j$ is the $j$-the dimension of $\boldsymbol{r}_h$. In addition, since here we want to minimize the distance, OptLSPE-V will output a pessimistic estimation of the target value function instead of an optimistic one.

- The player plays a policy $\mu^t$ sampled from its hyperpolicy $p^t$ and collects a trajectory.
- The player runs OptLSPE-V to obtain pessimistic value function estimations $\langle \boldsymbol{\theta}^t, \underline{\boldsymbol{V}}^t(\mu) \rangle$ for all $\mu \in \Pi$ and updates the hyperpolicy using Hedge.
- The dual variable is updated using online projected gradient ascent.

## D.2    THEORETICAL GUARANTEES

We still consider finite policy class $\Pi$ here. Notice that in the fictitious MG of VMDP, the opponent's policy class is also infinite, i.e., $\mathbb{B}(1)$. However, since the player only needs to estimate $\boldsymbol{V}_1^\mu(s_1)$, which is independent of $\boldsymbol{\theta}$, DORIS-V can also circumvent the union bound on $\boldsymbol{\theta}$ just like DORIS-C.

---

**Algorithm 6 `DORIS-V`**

---

**Input**: learning rate $\eta, \alpha_t$, confidence parameter $\beta$.
Initialize $p^1 \in \mathbb{R}^{|\Pi|}$ to be uniform over $\Pi$, $\boldsymbol{\theta}_1 \leftarrow 0$.
**for** $t = 1, \cdots, K$ **do**
    **Collect samples:**
    The learner samples $\mu^t$ from $p^t$.
    Run $\mu^t$ and collect $\mathcal{D}_t = \{s_1^t, a_1^t, \boldsymbol{r}_1^t, \cdots, s_{H+1}^t\}$.
    **Update policy distribution:**
    $\underline{\boldsymbol{V}}^t(\mu) \leftarrow \texttt{OptLSPE-V}(\mu, \mathcal{D}_{1:t-1}, \mathcal{F}, \mathcal{G}, \beta, \boldsymbol{\theta}^t), \quad \forall \mu \in \Pi.$
    $p^{t+1}(\mu) \propto p^t(\mu) \cdot \exp(-\eta \langle \underline{\boldsymbol{V}}^t(\mu), \boldsymbol{\theta}^t \rangle), \quad \forall \mu \in \Pi.$
    **Update dual variable:**
    $\boldsymbol{\theta}_{t+1} \leftarrow \text{Proj}_{\mathbb{B}(1)}(\boldsymbol{\theta}_t + \alpha_t(\underline{\boldsymbol{V}}^t(\mu_t) - \arg\max_{\boldsymbol{x} \in \mathcal{C}} \langle \boldsymbol{\theta}_t, \boldsymbol{x} \rangle)).$
**end for**
**Output**: $\widehat{\mu}$ uniformly sampled from $\mu^1, \cdots, \mu^K$.

---

**Algorithm 7 `OptLSPE-V`$(\mu, \mathcal{D}, \mathcal{F}, \mathcal{G}, \beta, \boldsymbol{\theta})$**

---

**Construct** $\mathcal{B}_{\mathcal{D}}(\mu)$ based on $\mathcal{D}$ via (17).
**Select** $\underline{\boldsymbol{V}} \leftarrow f_1(s_1, \mu)$, where $f = \arg\min_{f' \in \mathcal{B}_{\mathcal{D}}(\mu)} \langle f_1'(s_1, \mu), \boldsymbol{\theta} \rangle$.
**return** $\underline{\boldsymbol{V}}$.

---

In addition, we need to introduce the realizability and generalized completeness assumptions in this specific setting, which is simply a vectorized version as before:

**Assumption 9** (Realizability and generalized completeness in VMDP). *Assume that for any $h \in [H], j \in [d], \mu \in \Pi, f_{h+1} \in \mathcal{F}_{h+1}$, we have $Q_h^{\mu,j} \in \mathcal{F}_{h,j}, \mathcal{T}_h^{\mu,j} f_{h+1}^j \in \mathcal{G}_h^j$, where $Q_h^{\mu,j}$ is the $j$-the dimension of $\boldsymbol{Q}_h^\mu$ and $\mathcal{T}_h^{\mu,j}$ is the $j$-th dimensional Bellman operator at step $h$ defined in (18).*

Here $\mathcal{T}_h^{\mu,j}$ is defined as:

$$(\mathcal{T}_h^{\mu,j} f_{h+1}^j)(s, a) := r_h^j(s, a) + \mathbb{E}_{s' \sim P(\cdot|s,a)} f_{h+1}^j(s', \mu). \tag{18}$$

In addition, the BEE dimension for VMDP can be defined as the maximum BEE dimension among all $d$ dimensions:

**Definition 8.** *The $d$-dimensional $\epsilon$-Bellman Evaluation Eluder dimension of function class $\mathcal{F}$ on distribution family $\mathcal{Q}$ with respect to the policy class $\Pi$ is defined as follows:*

$$\dim_{\text{BEE}}(\mathcal{F}, \epsilon, \Pi, \mathcal{Q}) := \max_{j \in [d], h \in [H]} \dim_{\text{DE}}((\mathcal{I} - \mathcal{T}_h^{\Pi,j})\mathcal{F}^j, \mathcal{Q}_h, \epsilon),$$

*where $(\mathcal{I} - \mathcal{T}_h^{\Pi,j})\mathcal{F}^j := \{f_h^j - \mathcal{T}_h^{\mu,j} f_{h+1}^j : f \in \mathcal{F}, \mu \in \Pi\}$.*

We also use $\dim_{\text{BEE}}(\mathcal{F}, \epsilon, \Pi)$ to denote $\min\{\dim_{\text{BEE}}(\mathcal{F}, \epsilon, \Pi, \mathcal{Q}^1), \dim_{\text{BEE}}(\mathcal{F}, \epsilon, \Pi, \mathcal{Q}^2)\}$ as before.

The next theorem shows that `DORIS-V` is able to find a near optimal policy for Problem 2 with polynomial samples, where we use the following notations to simplify writing:

$$d_{\text{BEE,V}} := \dim_{\text{BEE}}\big(\mathcal{F}, \sqrt{1/K}, \Pi\big), \quad \mathcal{N}_{\text{cov,V}} := \max_{j \in [d]} \mathcal{N}_{\mathcal{F}^j \cup \mathcal{G}^j}(H/K)KH.$$

**Theorem 4.** *Under Assumption 1,9, there exists an absolute constant $c$ such that for any $\delta \in (0, 1]$, $K \in \mathbb{N}$, if we choose $\beta = cH^2 \log(\mathcal{N}_{\text{cov,V}} |\Pi| d/\delta)$, $\alpha_t = 2/(H\sqrt{dt})$, and $\eta = \sqrt{\log |\Pi|/(KH^2 d)}$ in `DORIS-V`, then with probability at least $1 - \delta$, we have:*

$$\text{dist}(\boldsymbol{V}_1^{\widehat{\mu}}(s_1), \mathcal{C}) \leq \min_{\mu \in \Pi} \text{dist}(\boldsymbol{V}_1^\mu(s_1), \mathcal{C}) + \mathcal{O}\Big(H^2\sqrt{d} \cdot \sqrt{d_{\text{BEE,V}} \log(\mathcal{N}_{\text{cov,V}} |\Pi| d/\delta)/K}\Big). \tag{19}$$

The bound in (19) shows that for any $\epsilon > 0$, if $K \geq \widetilde{\mathcal{O}}(d/\epsilon^2)$, $\widehat{\mu}$ will be an $\epsilon$ near-optimal policy with high probability. Compared to the results in Theorem 1 and Theorem 2, there is an additional term $d$. This is because the reward is $d$-dimensional and we are indeed evaluating $d$ scalar value functions in `OptLSPE-V`.

The proof is similar to that of Theorem 3 and utilizes the fact that both $\mu$ and $\boldsymbol{\theta}$ are updated via no-regret online learning algorithms (Hedge for $\mu$ and online projected gradient ascent for $\boldsymbol{\theta}$). See Appendix M for more details.

| Paper | Setting | Decentralized? | Baseline | Optimism |
|---|---|---|---|---|
| Tian et al. [49] | Tabular | **Yes** | Nash Value | Local, w.r.t. $V_h^*(s), \forall h, s$ |
| Liu et al. [33] | Tabular | **Yes** | **Best policy in hindsight** | Local, w.r.t. $\forall h, s, a, b$, $Q_h^{\mu,\nu^k}(s,a,b)$ |
| Jin et al. [27] Huang et al. [23] | **General FA** | **Yes** | Nash value | Global, w.r.t. $V_1^*(s_1)$ |
| This work | **General FA** | **Yes** | **Best policy in hindsight** | Global, w.r.t. $V_1^{\mu,\nu^k}(s_1)$ |

Table 1: Comparison with related works on decentralized learning with an adversarial opponent. Here General FA means general function approximation setting. From the table we can see that Tian et al. [49] focuses on tabular cases and can only compete agasint the Nash value of the game. Liu et al. [33] also works on tabular cases but the baseline is much stronger, i.e., the best policy in hindsight. Jin et al. [27], Huang et al. [23] are able to deal with general function approximation, but they can only compete against Nash value. In contrast, our work considers general function approximation and the baseline is the strongest (the same as Liu et al. [33]).

| Paper | Setting | Decentralized? | Number of players | Optimism |
|---|---|---|---|---|
| Jin et al. [26] Mao et al. [35] | Tabular | **Yes** | $\geq 2$ | Local, w.r.t. $\forall h, s$ $\max_\mu V_h^{\mu,\hat\nu}(s)$ |
| Jin et al. [27] Huang et al. [23] | **General FA** | No | 2 | Global, w.r.t. $V_1^*(s_1)$ |
| This work | **General FA** | **Yes** | $\geq 2$ | Global, w.r.t. $V_1^{\mu,\nu^k}(s_1)$ |

Table 2: Comparison with related works on finding equilibria in self-play Markov games. Here General FA means general function approximation setting. We can see that Jin et al. [26], Mao et al. [35] studies tabular cases. Their algorithms are decentralized and can still work when the number of players is larger than 2. Jin et al. [27], Huang et al. [23] works on general function approximation setting but their algorithms are centralized and limited to two-player zero-sum games. In comparison, our work can handle multi-agent ($\geq 2$ players) general-sum games with general function approximation and the algorithm is decentralized.

**Comparison with existing algorithms.** [58] has also proposed algorithms for approachability tasks in tabular cases and achieve the same sub-optimality gap with respect to $d$ and $K$ as Theorem 4. [37] studies the tabular and linear approximation cases, achieving $\sqrt{K}$ regret as well. Their sample complexity does not scale with $d$ because they have normalized the reward vector to lie in $\mathbb{B}(1)$ in tabular cases and $\mathbb{B}(\sqrt{d_{\text{lin}}})$ in $d_{\text{lin}}$-dimensional linear VMDPs. Compared to the above works, `DORIS-V` is able to tackle the more general cases with nonlinear function approximation and policy classes while retaining the sample efficiency.

# E  DISCUSSION

## E.1  COMPARISON WITH CLOSELY-RELATED WORKS

In this section we provide a more detailed comparison between `DORIS` and some closely-related works in this section. First we would like to summarize the comparison with decentralized learning literature and self-play literature in Table 1 and 2. Next we will clarify the novelty of our work given some relarted works.

**Novelty given Liu et al. [33].**  The idea of maintaining a hyperpolicy and utilizing Hedge to update it in `DORIS` is inspired from OPMD proposed in Liu et al. [33]. However, the policy evaluation algorithm in Liu et al. [33] can only work in tabular cases and our novelty lies in the new optimism and policy evaluation step (Algorithm 2) specially designed for the policy revealing setting. Note that the extension to the setting of general function approximation is not trivial. Combining existing techniques on reinforcement learning with general function approxiamtion (for example, Bellman Eluder dimension in Jin et al. [25]) with Liu et al. [33] does not lead to our work because the optimism in `DORIS` is (i) **global** (in the sense that optimism is only true for $s_1$, which differs from

Liu et al. [33]) and (ii) **policy-pair specific** (that is, our confidence set is only optimistic with respect to $V_1^{\mu,\nu^k}(s_1)$, which differs from Jin et al. [25]).

More specifically, Liu et al. [33] attains optimism via adding **a bonus term** $\beta$ for **each** step $h$ and state-action pair $(s,a,b)$ in value iteration as follows:

$$\bar{Q}_h^{\mu,\nu^k}(s,a,b) = \mathbb{E}_{s'\sim\hat{P}(\cdot|s,a)}[\bar{V}_{h+1}^{\mu,\nu^k}(s')] + r(s,a,b) + \beta,$$
$$\bar{V}_h^{\mu,\nu^k}(s) = \mathbb{E}_{a\sim\mu_h(s),b\sim\nu_h^k(s)}[\bar{Q}_h^{\mu,\nu^k}(s,a,b)].$$

This guarantees that $\bar{V}_h^{\mu,\nu^k}(s,a,b)$ is optimistic with respect to the true value function $V_h^{\mu,\nu^k}(s,a,b)$ for **each** $h,s,a,b$. However, DORIS picks the **most optimistic estimation** from the **constructed confidence set** directly:

$$\bar{V}^{\mu,\nu^k} = \max_{f\in\mathcal{B}(\mu,\nu^k)} f(s_1,\mu,\nu^k).$$

This can only guarantee that $\bar{V}^{\mu,\nu^k}$ is optimistic with respect to $V_1^{\mu,\nu^k}(s_1)$. Thus bounding the regret will be harder in our case with only global optimism.

In addition, although Jin et al. [25] also uses global optimism, it is optimistic with respect to the **fixed optimal value function** $V_1^*(s_1)$ and only needs to construct **one** confidence set of it. Nevertheless, to tackle the non-stationary optimal policy in the decentralized setting which keeps changing across episodes due to the adversarial opponent, DORIS needs to construct a confidence set of $V_1^{\mu,\nu^k}(s_1)$ for **each policy** $\mu$ in the policy class and the estimation $\bar{V}^{\mu,\nu^k}$ is only optimistic with respect to $V_1^{\mu,\nu^k}(s_1)$ for each $\mu$ respectively. Therefore, the analysis techniques in Jin et al. [25] cannot be applied directly here. We need to decompose the regret in a different way and propose a new complexity measure (i.e., BEE dimension) to bound the regret.

**Novelty given Jin et al. [27], Huang et al. [23].** Jin et al. [23], Huang et al. [23] also design algorithms and utilize analysis techniques based on Bellman-eluder-type complexity (i.e., multi-agent BE dimension [27, 23]) in Markov games. Here we want to clarify that although the BEE dimension is also inspired from Bellman Eluder dimension proposed in Jin et al. [25], DORIS is very different from Jin et al. [27], Huang et al. [23] in terms of **confidence set construction** and **optimism**, which makes their analysis techniques not applicable here either.

Jin et al. [27], Huang et al. [23] consider zero-sum Markov games in the **centralized** setting with general function approximation. The algorithms in these works are based on (i) constructing confidence regions of **the optimal value function (i.e., Nash value function in zero-sum games) or model** and (ii) solving the Nash equilibrium with respect to the optimistic function/model. As a result, their algorithms can be regarded as running **optimistic greedy policies** in games and the estimated value functions are always optimistic estimates of the optimal value function $V_1^*(s_1)$ of the underlying game.

In contrast, in the **decentralized** setting, one unique challenge faced by DORIS is that the optimal policy is indeed changing across the episodes because we cannot control the opponent and the opponent can be adversarially adjusting its own policy. Therefore, there does not exist such a fixed optimal value function that we can run optimism with respect to. More importantly, from the view of the single agent, the environment is adversarially changing due to the opponent. Such nonstationarity does not appear in these works. To deal with this challenge, DORIS is based on (i) constructing the confidence region for **policy evaluation** problems, (ii) running **mirror descent** over the space of policies. As a result, DORIS is more like a **decentralized policy optimization** algorithm and the value functions maintained by the DORIS are only optimistic with respect to the value functions associated with **the current policy pair** $(\mu,\nu^k)$, which changes at each iteration.

More importantly, such a different version of optimism leads to a different regret decomposition. Specifically, in (22), we show that the regret is upper bounded by the policy evaluation error $\sum_{t=1}^{K}(\bar{V}^t(\mu^t) - V_1^{\pi^t}(s_1))$ and online learning error induced $\sum_{t=1}^{K}(\bar{V}^t(\mu^*) - \bar{V}^t(\mu^t))$ by mirror descent. Bounding the evaluation error $\sum_{t=1}^{K}(\bar{V}^t(\mu^t) - V_1^{\pi^t}(s_1))$ incurred by achieving optimism in policy evaluation has not been considered in Jin et al. [27], Huang et al. [23]. Multi-agent BE dimension [27, 23] cannot be applied here either because it measures

the Bellman residuals $f_h(s,a,b) - r_{h+1}(s,a,b) - \min_\nu f_{h+1}(s,\mu,\nu)$ and can only help bound $\sum_{t=1}^K (\bar{V}^t(\mu^t) - \min_\nu V_1^{\mu^t,\nu}(s_1))$ when the policy $\nu^t$ played by the opponent is a pessimistic best response of $\mu^t$. In our case $\nu^t$ is arbitrary (typically not the best response of $\mu^t$) and the value we want to bound is also different, therefore multi-agent BE dimension is not applicable and we have to propose BEE dimension to evaluate the complexity of policy evaluation tasks with general function approximation. Along with the new measure, we have also identified common function classes with low BEE dimension in the paper to illustrate the capacity of BEE dimension.

### E.2 LOWER BOUND IN LIU ET AL. [33]

Here we present a lower bound from Liu et al. [33]:

**Theorem 5** (Liu et al. [33, Theorem 4]). *There exists a Markov game with $S, A = \mathcal{O}(H)$ and an opponent who chooses policy uniformly at random from an unknown set of $H$ Markov policies in each episode, such that when the opponent's policy is not revealed, the regret for competing with the best fixed Markov policy in hindsight is $\Omega(\min\{K, 2^H\}/H)$.*

The above lower bound shows that if the opponent's policy is not revealed, even when the opponent only plays a finite number of Markov policies, the exponential regret lower bound for competing with the best Markov policy in hindsight is inevitable, which validates the necessity of policy revealing condition.

### E.3 COMPUTATIONAL COMPLEXITY OF `DORIS`

There are mainly two steps in `DORIS` that require computation, optimistic policy evaluation via `OptLSPE` and hyperpolicy update via Hedge. Assuming the policy class is finite facilitates the second step, but even with finite policy class, `OptLSPE` is still computationally inefficient. This is due to the global optimism step in `OptLSPE`, i.e., constructing the confidence set (Equation (3)) and finding the most optimistic estimation. This is a common issue of algorithms with general function approximation even in single-agent MDPs. For example, the global optimism step of the algorithms in [24, 25, 14, 23, 27] are all computationally inefficient and hard to implement. However, if we only consider linear MGs, computationally efficient algorithms are possible since we can use local optimism and implement `OptLSPE` by an analog of LSVI-UCB [28], which is computationally efficient. In addition, if there is a computationally efficient solver for optimistic policy evaluation with general function approximation in single-agent MDPs, we believe that we can also utilize it here since the confidence set update rule (Equation (3)) is similar to single-agent MDPs. That said, in this work we mainly focus on the statistical complexity of learning the Markov game and thus computationally efficient algorithms are left as future works.

## F   PROOFS OF PROPOSITION 1

From the completeness assumption, we know that there exists $g_h \in \mathcal{F}_h$ such that $g_h = \mathcal{T}_h^{\mu,\nu} f_{h+1}$, which implies that

$$f_h - \mathcal{T}_h^{\mu,\nu} f_{h+1} \in \mathcal{F}_h - \mathcal{F}_h, \forall f \in \mathcal{F}, \mu \in \Pi, \nu \in \Pi'.$$

In other words, $(I - \mathcal{T}_h^{\Pi,\Pi'})\mathcal{F} \subseteq \mathcal{F}_h - \mathcal{F}_h$. Therefore, from the definition of $\dim_{\text{BEE}}(\mathcal{F}, \epsilon, \Pi, \Pi')$ we have

$$\dim_{\text{BEE}}(\mathcal{F}, \epsilon, \Pi, \Pi') \leq \dim_{\text{BEE}}(\mathcal{F}, \epsilon, \Pi, \Pi', \mathcal{Q}^2) = \max_{h \in [H]} \dim_{\text{DE}}((I - \mathcal{T}_h^{\Pi,\Pi'})\mathcal{F}, \mathcal{Q}_h^2, \epsilon)$$

$$\leq \max_{h \in [H]} \dim_{\text{DE}}((\mathcal{F}_h - \mathcal{F}_h), \mathcal{Q}_h^2, \epsilon) = \max_{h \in [H]} \dim_{\text{E}}(\mathcal{F}_h, \epsilon),$$

where the last step comes from the definition of $\dim_{\text{E}}$ and $\mathcal{Q}_h^2$ is the dirac distribution family. This concludes our proof.

# G EXAMPLES FOR REALIZABILITY, GENERALIZED COMPLETENESS AND COVERING NUMBER

In this section we illustrate practical examples where realizability and generalized completeness hold and the covering number is upper bounded at the same time. In specific, we will consider tabular MGs, linear MGs and kernel MGs.

## G.1 TABULAR MGS

For tabular MGs, we let $\mathcal{F}_h = \{f | f : \mathcal{S} \times \mathcal{A} \times \mathcal{B} \mapsto [0, V_{\max}]\}$ and $\mathcal{G}_h = \mathcal{F}_h$ for all $h \in [H]$. Then it is obvious that $Q_h^{\mu \times \nu} \in \mathcal{F}_h$ and $\mathcal{T}^{\mu,\nu} f_{h+1} \in \mathcal{G}_h$ for any $f \in \mathcal{F}, h \in [H], \mu, \nu$, which implies that realizability and generalized completeness are satisfied. In addition, notice that in this case we have

$$\log \mathcal{N}_{\mathcal{F}_h}(\epsilon) = \log \mathcal{N}_{\mathcal{G}_h}(\epsilon) \le |\mathcal{S}||\mathcal{A}||\mathcal{B}| \log(V_{\max}/\epsilon).$$

This suggests that the size of $\mathcal{F}$ and $\mathcal{G}$ is also not too large.

## G.2 LINEAR MGS

In this subsection we consider linear MGs. Here we generalize the definition of linear MDPs in classic MDPs [28] to Markov games:

**Definition 9** (Linear MGs). *We say an MG is linear of dimension $d$ if for each $h \in [H]$, there exists a feature mapping $\phi_h : \mathcal{S} \times \mathcal{A} \times \mathcal{B} \mapsto \mathbb{R}^d$ and $d$ unknown signed measures $\psi_h = (\psi_h^{(1)}, \cdots, \psi_h^{(d)})$ over $\mathcal{S}$ and an unknown vector $\theta_h \in \mathbb{R}^d$ such that $P_h(\cdot|s,a,b) = \phi_h(s,a,b)^\top \psi_h(\cdot)$ and $r_h(s,a,b) = \phi_h(s,a,b)^\top \theta_h$ for all $(s,a,b) \in \mathcal{S} \times \mathcal{A} \times \mathcal{B}$.*

Without loss of generality, we assume $\|\phi_h(s,a,b)\| \le 1$ for all $s \in \mathcal{S}, a \in \mathcal{A}, b \in \mathcal{B}$ and $\|\psi_h(\mathcal{S})\| \le \sqrt{d}, \theta_h \le \sqrt{d}$ for all $h$. Let $\mathcal{F}_h = \mathcal{G}_h = \{\phi_h(\cdot)^\top w | w \in \mathbb{R}^d, \|w\| \le (H-h+1)\sqrt{d}, 0 \le \phi_h(\cdot)^\top w \le H - h + 1\}$.

**Realizability.** We have for any $\mu, \nu$,

$$Q_h^{\mu \times \nu}(s,a,b) = r_h(s,a,b) + \mathbb{E}_{s' \sim P(\cdot|s,a,b)}[V_{h+1}^{\mu \times \nu}(s')]$$
$$= \langle \phi_h(s,a,b), \theta_h \rangle + \left\langle \phi_h(s,a,b), \int_{\mathcal{S}} V_{h+1}^{\mu \times \nu}(s') d\psi_h(s') \right\rangle$$
$$= \left\langle \phi_h(s,a,b), \theta_h + \int_{\mathcal{S}} V_{h+1}^{\mu \times \nu}(s') d\psi_h(s') \right\rangle$$
$$= \langle \phi_h(s,a,b), w_h^{\mu \times \nu} \rangle,$$

where $w_h^{\mu \times \nu} = \theta_h + \int_{\mathcal{S}} V_{h+1}^{\mu \times \nu}(s') d\psi_h(s')$ and thus $\|w_h^{\mu \times \nu}\| \le (H-h+1)\sqrt{d}$. Therefore, $Q_h^{\mu \times \nu} \in \mathcal{F}_h$, which means that realizability holds.

**Generalized completeness.** For any $f_{h+1} \in \mathcal{F}_{h+1}$, we have

$$\mathcal{T}^{\mu,\nu} f_{h+1}(s,a,b) = r_h(s,a,b) + \mathbb{E}_{s' \sim P(\cdot|s,a,b)}[f_{h+1}(s',\mu,\nu)]$$
$$= \left\langle \phi_h(s,a,b), \theta_h + \int_{\mathcal{S}} f_{h+1}(s',\mu,\nu) d\psi_h(s') \right\rangle.$$

Since $\|f_{h+1}\|_\infty \le H - h$, we have $\|\theta_h + \int_{\mathcal{S}} f_{h+1}(s',\mu,\nu) d\psi_h(s')\| \le (H - h + 1)\sqrt{d}$, which indicates $\mathcal{T}^{\mu,\nu} f_{h+1} \in \mathcal{G}_h$ and thus generalized completeness is satisfied.

**Covering number.** First notice that from the literature [51], the covering number of a $l_2$-norm ball can be bounded as $\log \mathcal{N}_{\mathbb{B}((H-h+1)\sqrt{d})}(\epsilon) \le d \log(3H\sqrt{d}/\epsilon)$. Therefore, there exists $\mathcal{W} \subset \mathbb{B}((H - h + 1)\sqrt{d})$ where $\log |\mathcal{W}| \le d \log(3H\sqrt{d}/\epsilon)$ such that for any $w \in \mathbb{B}((H - h + 1)\sqrt{d})$, there exists $w' \in \mathcal{W}$ satisfying $\|w' - w\| \le \epsilon$. Now let $\mathcal{F}'_h = \{\phi_h(\cdot)^\top w | w \in \mathcal{W}\}$. For any

$f_h \in \mathcal{F}_h$, suppose $f_h(\cdot) = \phi_h(\cdot)^\top w_{f_h}$. Then we know there exists $f'_h(\cdot) = \phi_h(\cdot)^\top w'_{f_h} \in \mathcal{F}'_h$ where $\|w'_{f_h} - w_{f_h}\| \le \epsilon$, which implies

$$|f_h(s, a, b) - f'_h(s, a, b)| \le \|\phi_h(s, a, b)\| \|w'_{f_h} - w_{f_h}\| \le \epsilon.$$

Therefore $\log \mathcal{N}_{\mathcal{F}_h}(\epsilon) \le \log |\mathcal{F}'_h| = \log |\mathcal{W}| \le d \log(3H\sqrt{d}/\epsilon)$.

## G.3 Kernel MGs

In this subsection we show that kernel MGs also satisfy realizability and generalized completeness naturally. In addition, when a kernel MG has a bounded effective dimension, its covering number will also be bounded. First we generalize the definition of kernel MDPs [25] to MGs as follows.

**Definition 10** (Kernel MGs). *In a kernel MDP, for each step $h \in [H]$, there exist feature mapping $\phi_h : \mathcal{S} \times \mathcal{A} \times \mathcal{B} \mapsto \mathcal{H}$ and $\psi_h : \mathcal{S} \mapsto \mathcal{H}$ where $\mathcal{H}$ is a separable Hilbert space such that $P_h(s'|s, a, b) = \langle \phi_h(s, a, b), \psi_h(s') \rangle_{\mathcal{H}}$ for all $s \in \mathcal{S}, a \in \mathcal{A}, b \in \mathcal{B}, s' \in \mathcal{S}$. Besides, the reward function os linear in $\phi$, i.e., $r_h(s, a, b) = \langle \phi_h(s, a, b), \theta_h \rangle_{\mathcal{H}}$ for some $\theta_h \in \mathcal{H}$. Moreover, a kernel MG satisfies the following regularization conditions:*

- $\|\theta_h\|_{\mathcal{H}} \le 1, \|\phi_h(s, a, b)\|_{\mathcal{H}} \le 1$, *for all $s \in \mathcal{S}, a \in \mathcal{A}, b \in \mathcal{B}, h \in [H]$.*

- $\|\sum_{s \in \mathcal{S}} V(s)\psi_h(s)\|_{\mathcal{H}} \le 1$, *for all function $V : \mathcal{S} \mapsto [0, 1], h \in [H]$.*

**Remark 7.** *It can be observed that tabular and linear MGs are special cases of kernel MGs. Therefore, the following discussion applies to tabular and linear MGs as well.*

Then we let $\mathcal{F}_h = \mathcal{G}_h = \{\phi_h(\cdot)^\top w | w \in \mathcal{B}_{\mathcal{H}}(H - h + 1)\}$ where $\mathcal{B}_{\mathcal{H}}(r)$ is a ball with radius $r$ in $\mathcal{H}$. Following the same arguments in linear MGs, we can validate that realizability and generalized completeness are satisfied in kernel MGs.

**Covering number.** Before bounding the covering number of $\mathcal{F}_h$, we need introduce a new measure to evaluate the complexity of a Hilbert space since $\mathcal{H}$ might be infinite dimensional. Here we use the effective dimension [14, 25], which is defined as follows:

**Definition 11** ($\epsilon$-effective dimension of a set). *The $\epsilon$-effective dimension of a set $\mathcal{X}$ is the minimum integer $d_{\text{eff}}(\mathcal{X}, \epsilon) = n$ such that*

$$\sup_{x_1, \cdots, x_n \in \mathcal{X}} \frac{1}{n} \log \det \left( I + \frac{1}{\epsilon^2} \sum_{i=1}^n x_i x_i^\top \right) \le e^{-1}.$$

**Remark 8.** *When $\mathcal{X}$ is finite dimensional, suppose its dimension is $d$. Then its effective dimension can be upper bounded by $\mathcal{O}(d \log(1 + R^2/\epsilon))$ where $R$ is the norm bound of $\mathcal{X}$ [14]. In addition, even when $\mathcal{X}$ is infinite dimensional, if the eigenspectrum of the covariance matrices concentrates in a low-dimension subspace, the effective dimension of $\mathcal{X}$ can still be small [48].*

We call a kernel MG is of effective dimension $d(\epsilon)$ if $d_{\text{eff}}(\mathcal{X}_h, \epsilon) \le d(\epsilon)$ for all $h$ and $\epsilon$ where $\mathcal{X}_h = \{\phi_h(s, a, b) : (s, a, b) \in \mathcal{S} \times \mathcal{A} \times \mathcal{B}\}$. Then the following proposition shows that the covering number of $\mathcal{F}_h$ is upper bounded by the effective dimension of the kernel MG:

**Proposition 2.** *If the kernel MG has effective dimension $d(\epsilon)$, then*

$$\log \mathcal{N}_{\mathcal{F}_h}(\epsilon) \le \mathcal{O}\big(d(\epsilon/2H) \log(1 + Hd(\epsilon/2H)/\epsilon)\big).$$

*Proof.* Suppose $\dim_{\text{E}}(\mathcal{F}_h, \epsilon) = n$. Then by the definition of Eluder dimension, there exists a sequence $\{\phi_i\}_{i=1}^n$ such that for any $w_1, w_2 \in \mathbb{B}_{\mathcal{H}}(H - h + 1), \phi \in \mathcal{X}_h$, if $\sum_{i=1}^n (\langle \phi_i, w_1 - w_2 \rangle)^2 \le \epsilon^2$, then $|\langle \phi, w_1 - w_2 \rangle| \le \epsilon$. Therefore, the covering number of kernel MGs can be reduced to covering the projection of $\mathbb{B}_{\mathcal{H}}(H - h + 1)$ onto the space spanned by $\{\phi_i\}_{i=1}^n$, whose dimension is at most $n$. From the literature [51], the covering number of such space is $\mathcal{O}(n \log(1 + nH/\epsilon))$, which implies

$$\log \mathcal{N}_{\mathcal{F}_h}(\epsilon) \le \mathcal{O}\big(n \log(1 + nH/\epsilon)\big).$$

Finally, by the proof of Proposition 3, we know $n \le d(\epsilon/2H)$, which concludes the proof.

$\square$

# H    EXAMPLES FOR BEE DIMENSION

In this section we will show that kernel MGs (including tabular MGs and linear MGs) and generalized linear complete models have low BEE dimensions.

## H.1    KERNEL MGS

Consider the kernel MG defined in Definition 10 and $\mathcal{F}_h = \{\phi_h(\cdot)^\top w | w \in \mathcal{B}_{\mathcal{H}}(H - h + 1)\}$, then we have the following proposition showing that the BEE dimension of a kernel MG is upper bounded by its effective dimension (Definition 11):

**Proposition 3.** *If the kernel MG has effective dimension $d(\epsilon)$, then for any policy classes $\Pi$ and $\Pi'$, we have $d_{\mathrm{BEE}}(\mathcal{F}, \epsilon, \Pi, \Pi') \leq d(\epsilon/2H)$.*

*Proof.* First in Appendix G we have showed that $\mathcal{F}$ satisfies completeness. By Proposition 1, we have $d_{\mathrm{BEE}}(\mathcal{F}, \epsilon, \Pi, \Pi') \leq \max_{h \in [H]} \dim_{\mathrm{E}}(\mathcal{F}_h, \epsilon)$. Therefore we only need to bound $\dim_{\mathrm{E}}(\mathcal{F}_h, \epsilon)$ for each $h \in [H]$. Suppose $\dim_{\mathrm{E}}(\mathcal{F}_h, \epsilon) = k > d(\epsilon/2H)$. Then by the definition of Eluder dimension, there exists a sequence $\phi_1, \cdots, \phi_k$ and $\{w_{1,i}\}_{i=1}^k, \{w_{2,i}\}_{i=1}^k$ where $\phi_i \in \mathcal{X}_h = \{\phi_h(s,a,b) : (s,a,b) \in \mathcal{S} \times \mathcal{A} \times \mathcal{B}\}, w_{1,i}, w_{2,i} \in \mathcal{B}_{\mathcal{H}}(H - h + 1)$ for all $i$ such that for any $t \in [k]$:

$$\sum_{i=1}^{t-1} (\langle \phi_i, w_{1,t} - w_{2,t} \rangle)^2 \leq (\epsilon')^2, \tag{20}$$

$$|\langle \phi_t, w_{1,t} - w_{2,t} \rangle| \geq \epsilon', \tag{21}$$

where $\epsilon' \geq \epsilon$. Let $\Sigma_t$ denote $\sum_{i=1}^{t-1} \phi_i \phi_i^\top + \frac{\epsilon^2}{4H^2} \cdot I$. Then we have for any $t \in [k]$

$$\|w_{1,t} - w_{2,t}\|_{\Sigma_t}^2 \leq (\epsilon')^2 + \epsilon^2.$$

On the other hand, by Cauchy-Schwartz inequality we know

$$\|\phi_t\|_{\Sigma_t^{-1}} \|w_{1,t} - w_{2,t}\|_{\Sigma_t} \geq |\langle \phi_t, w_{1,t} - w_{2,t} \rangle| \geq \epsilon'.$$

This implies for all $t \in [k]$

$$\|\phi_t\|_{\Sigma_t^{-1}} \geq \frac{\epsilon'}{\sqrt{\epsilon^2 + (\epsilon')^2}} \geq \frac{1}{\sqrt{2}}.$$

Therefore, applying elliptical potential lemma (e.g., Lemma 5.6 and Lemma F.3 in [14]), we have for any $t \in [k]$

$$\log\det\left(I + \frac{4H^2}{\epsilon^2} \sum_{i=1}^t \phi_i \phi_i^\top\right) = \sum_{i=1}^t \log(1 + \|\phi_i\|_{\Sigma_i^{-1}}^2) \geq t \cdot \log\frac{3}{2}.$$

However, by the definition of effective dimension, we know when $n = d_{\mathrm{eff}}(\mathcal{X}_h, \frac{\epsilon}{2H})$,

$$\sup_{\phi_1, \cdots, \phi_n} \log\det\left(I + \frac{4H^2}{\epsilon^2} \sum_{i=1}^n \phi_i \phi_i^\top\right) \leq n e^{-1}.$$

This is a contradiction since $n \leq d(\epsilon/2H) < k$ and $\log\frac{3}{2} > e^{-1}$. Therefore we have $\dim_{\mathrm{E}}(\mathcal{F}_h, \epsilon) \leq d(\epsilon/2H)$ for all $h \in [H]$, which implies

$$d_{\mathrm{BEE}}(\mathcal{F}, \epsilon, \Pi, \Pi') \leq d(\epsilon/2H).$$

This concludes our proof.                                                                             $\square$

**Tabular MGs.**    Tabular MGs are a special case of kernel MGs where the feature vectors are $|\mathcal{S}||\mathcal{A}||\mathcal{B}|$-dimensional one-hot vectors. From the standard elliptical potential lemma, we know $d(\epsilon) = \widetilde{\mathcal{O}}(|\mathcal{S}||\mathcal{A}||\mathcal{B}|)$ for tabular MDPs, suggesting their BEE dimension is also upper bounded $\widetilde{\mathcal{O}}(|\mathcal{S}||\mathcal{A}||\mathcal{B}|)$.

**Linear MGs.** When the feature vectors are $d$-dimensional, we can recover linear MGs. Similarly, by the standard elliptical potential lemma, we have the BEE dimension of linear MGs is upper bounded $\widetilde{\mathcal{O}}(d)$.

### H.2 GENERALIZED LINEAR COMPLETE MODELS

An important variant of linear MDPs is the generalized linear complete models proposed by [52]. Here we also generalize it into Markov games:

**Definition 12** (Generalized linear complete models). *In $d$-dimensional generalized linear complete models, for each step $h \in [H]$, there exists a feature mapping $\phi_h : \mathcal{S} \times \mathcal{A} \times \mathcal{B} \mapsto \mathbb{R}^d$ and a link function $\sigma$ such that:*

- *for the generalized linear function class $\mathcal{F}_h = \{\sigma(\phi_h(\cdot)^\top w) | w \in \mathcal{W}\}$ where $\mathcal{W} \subset \mathbb{R}^d$, realizability and completeness are both satisfied;*

- *the link function is strictly monotone, i.e., there exist $0 < c_1 < c_2 < \infty$ such that $\sigma' \in [c_1, c_2]$.*

- *$\phi_h, w$ satisfy the regularization conditions: $\|\phi_h(s, a, b)\| \leq R, \|w\| \leq R$ for all $s, a, b, h$ where $R > 0$ is a constant.*

When the link function is $\sigma(x) = x$, the generalized linear complete models reduce to the linear complete models, which contain instances such as linear MGs and LQRs. The following proposition shows that generalized linear complete models also have low BEE dimensions:

**Proposition 4.** *If a generalized linear complete model has dimension $d$, then for any policy classes $\Pi$ and $\Pi'$, its BEE dimension can be bounded as follows:*

$$d_{\text{BEE}}(\mathcal{F}, \epsilon, \Pi, \Pi') \leq \widetilde{\mathcal{O}}(dc_2^2/c_1^2).$$

*Proof.* The proof is similar to Proposition 3, except (20) and (21) become

$$\sum_{i=1}^{t-1} c_1^2 (\langle \phi_i, w_{1,t} - w_{2,t} \rangle)^2 \leq \sum_{i=1}^{t-1} (\sigma(\phi_i^\top w_{1,t}) - \sigma(\phi_i^\top w_{2,t}))^2 \leq (\epsilon')^2,$$

$$c_2 |\langle \phi_t, w_{1,t} - w_{2,t} \rangle| \geq |\sigma(\phi_t^\top w_{1,t}) - \sigma(\phi_t^\top w_{2,t})| \geq \epsilon'.$$

Then repeat the arguments in the proof of Proposition 3, we have $\dim_E(\mathcal{F}_h, \epsilon) \leq \widetilde{\mathcal{O}}(dc_2^2/c_1^2)$ for all $h \in [H]$. Since $\mathcal{F}$ satisfies completeness, we can use Proposition 1 and obtain

$$d_{\text{BEE}}(\mathcal{F}, \epsilon, \Pi, \Pi') \leq \widetilde{\mathcal{O}}(dc_2^2/c_1^2).$$

$\square$

## I PROOF OF THEOREM 1 AND THEOREM 2

In this section we present the proof for Theorem 1 and Theorem 2. We first consider the oblivious setting. Let $\mu^* = \arg\max_{\mu \in \Pi} \sum_{t=1}^K V_1^{\mu \times \nu^t}(s_1)$ and we can decompose the regret into the following terms:

$$\max_{\mu \in \Pi} \sum_{t=1}^K V_1^{\mu \times \nu^t}(s_1) - \sum_{t=1}^K V_1^{\pi^t}(s_1) = \underbrace{\left( \sum_{t=1}^K V_1^{\mu^* \times \nu^t}(s_1) - \sum_{t=1}^K \overline{V}^t(\mu^*) \right)}_{(1)}$$

$$+ \underbrace{\left( \sum_{t=1}^K \overline{V}^t(\mu^*) - \sum_{t=1}^K \langle \overline{V}^t, p^t \rangle \right)}_{(2)} + \underbrace{\left( \sum_{t=1}^K \langle \overline{V}^t, p^t \rangle - \sum_{t=1}^K \overline{V}^t(\mu^t) \right)}_{(3)}$$

$$+ \underbrace{\left( \sum_{t=1}^K \overline{V}^t(\mu^t) - \sum_{t=1}^K V_1^{\pi^t}(s_1) \right)}_{(4)}. \tag{22}$$

Our proof bounds these terms separately and mainly consists of three steps:

- Prove $\overline{V}^t(\mu)$ is an optimistic estimation of $V_1^{\mu \times \nu^t}(s_1)$ for all $t \in [K]$ and $\mu \in \Pi$, which implies that term $(1) \leq 0$.

- Bound term $(4)$, the cumulative estimation error $\sum_{t=1}^{K} \overline{V}^t(\mu^t) - V_1^{\pi^t}(s_1)$. In this step we utilize the newly proposed complexity measure BEE dimension to bridge the cumulative estimation error and the empirical Bellman residuals occurred in $\texttt{OptLSPE}$.

- Bound term $(2)$ using the existing results of online learning error induced by Hedge and bound $(3)$ by noticing that it is a martingale difference sequence.

## I.1   STEP 1: PROVE OPTIMISM

First we can show that the constructed set $\mathcal{B}_{\mathcal{D}_{1:t-1}}(\mu, \nu^t)$ is not vacuous in the sense that the true action-value function $Q^{\mu, \nu^t}$ belongs to it with high probability

**Lemma 2.** *With probability at least $1 - \delta/4$, we have for all $t \in [K], \mu \in \Pi, Q^{\mu, \nu^t} \in \mathcal{B}_{\mathcal{D}_{1:t-1}}(\mu, \nu^t)$.*

*Proof.* See Appendix J.1. □

Then since $\overline{V}^t(\mu) = \max_{f \in \mathcal{B}_{\mathcal{D}_{1:t-1}}(\mu, \nu^t)} f(s_1, \mu, \nu^t)$, we know for all $t \in [K]$ and $\mu \in \Pi$,

$$\overline{V}^t(\mu) \geq Q^{\mu, \nu^t}(s_1, \mu, \nu^t) = V_1^{\mu \times \nu^t}(s_1).$$

In particular, we have for all $t \in [K]$,

$$\overline{V}^t(\mu^*) \geq V_1^{\mu^* \times \nu^t}(s_1). \tag{23}$$

## I.2   STEP 2: BOUND ESTIMATION ERROR

Next we need to show the estimation error $\sum_{t=1}^{K} \overline{V}^t(\mu^t) - V_1^{\pi^t}(s_1)$ is small. Let $f^{t,\mu} = \arg\max_{f \in \mathcal{B}_{\mathcal{D}_{1:t-1}}(\mu, \nu^t)} f(s_1, \mu, \nu^t)$. Then using standard concentration inequalities, we can have the following lemma which says that empirical Bellman residuals are indeed close to true residuals with high probability. Recall that here $\pi^k = \mu^k \times \nu^k$.

**Lemma 3.** *With probability at least $1 - \delta/4$, we have for all $t \in [K]$, $h \in [H]$ and $\mu \in \Pi$,*

$$(a) \quad \sum_{k=1}^{t-1} \mathbb{E}_{\pi^k} \left[ \left( f_h^{t,\mu}(s_h, a_h, b_h) - (\mathcal{T}_h^{\mu, \nu^t} f_{h+1}^{t,\mu})(s_h, a_h, b_h) \right)^2 \right] \leq \mathcal{O}(\beta), \tag{24}$$

$$(b) \quad \sum_{k=1}^{t-1} \left( f_h^{t,\mu}(s_h^k, a_h^k, b_h^k) - (\mathcal{T}_h^{\mu, \nu^t} f_{h+1}^{t,\mu})(s_h^k, a_h^k, b_h^k) \right)^2 \leq \mathcal{O}(\beta). \tag{25}$$

*Proof.* See Appendix J.2. □

Besides, using performance difference lemma we can easily bridge $\overline{V}^t(\mu^t) - V_1^{\pi^t}(s_1)$ with Bellman residuals, whose proof is deferred to Appendix J.3:

**Lemma 4.** *For any $t \in [K]$, we have*

$$\overline{V}^t(\mu^t) - V_1^{\pi^t}(s_1) = \sum_{h=1}^{H} \mathbb{E}_{\pi^t} \left[ (f_h^{t,\mu^t} - \mathcal{T}_h^{\mu^t, \nu^t} f_{h+1}^{t,\mu^t})(s_h, a_h.b_h) \right].$$

Therefore, from Lemma 4 we can obtain

$$\sum_{t=1}^{K} \overline{V}^t(\mu^t) - V_1^{\pi^t}(s_1) = \sum_{h=1}^{H} \sum_{t=1}^{K} \mathbb{E}_{\pi^t} \left[ (f_h^{t,\mu^t} - \mathcal{T}_h^{\mu^t, \nu^t} f_{h+1}^{t,\mu^t})(s_h, a_h, b_h) \right]. \tag{26}$$

Notice that in (26) we need to bound the Bellman residuals of $f_h^{t,\mu^t}$ weighted by policy $\pi^t$. However, in Lemma 3, we can only bound the Bellman residuals weighted by $\pi^{1:t-1}$. Fortunately, we can utilize the inherent low BEE dimension to bridge these two values with the help of the following technical lemma:

**Lemma 5** ([25]). *Given a function class $\Phi$ defined on $\mathcal{X}$ with $\phi(x) \leq C$ for all $(\phi, x) \in \Phi \times \mathcal{X}$, and a family of probability measures $\mathcal{Q}$ over $X$. Suppose sequence $\{\phi_t\}_{t=1}^{K} \subset \Phi$ and $\{\rho_t\}_{t=1}^{K} \subset \mathcal{Q}$ satisfy that for all $t \in [K]$, $\sum_{k=1}^{t-1}(\mathbb{E}_{\rho_k}[\phi_t])^2 \leq \beta$. Then for all $t \in [K]$ and $w > 0$,*

$$\sum_{k=1}^{t} |\mathbb{E}_{\rho_k}[\phi_k]| \leq \mathcal{O}\Big(\sqrt{\dim_{\mathrm{DE}}(\Phi, \mathcal{Q}, w)\beta t} + \min\{t, \dim_{\mathrm{DE}}(\Phi, \mathcal{Q}, w)\}C + tw\Big).$$

Invoking Lemma 5 with $\mathcal{Q} = \mathcal{Q}_h^1$, $\Phi = (I - \mathcal{T}_h^{\Pi, \Pi'})\mathcal{F}$ and $w = \sqrt{1/K}$, conditioning on the event (24) in Lemma 3 holds true, we have

$$\sum_{t=1}^{K} \mathbb{E}_{\pi^t}\big[(f_h^{t,\mu^t} - \mathcal{T}_h^{\mu^t,\nu^t} f_{h+1}^{t,\mu^t})(s_h, a_h.b_h)\big]$$

$$\leq \mathcal{O}\bigg(\sqrt{V_{\max}^2 K \dim_{\mathrm{BEE}}\Big(\mathcal{F}, \sqrt{1/K}, \Pi, \Pi', \mathcal{Q}^1\Big) \log\big(\mathcal{N}_{\mathcal{F} \cup \mathcal{G}}(V_{\max}/K) K H |\Pi|/\delta\big)}\bigg). \quad (27)$$

Similarly, invoking Lemma 5 with $\mathcal{Q} = \mathcal{Q}_h^2$, $\Phi = (I - \mathcal{T}_h^{\Pi, \Pi'})\mathcal{F}$ and $w = \sqrt{1/K}$, conditioning on the event (25) in Lemma 3 holds true, we have with probability at least $1 - \delta/4$,

$$\sum_{t=1}^{K} \mathbb{E}_{\pi^t}\big[(f_h^{t,\mu^t} - \mathcal{T}_h^{\mu^t,\nu^t} f_{h+1}^{t,\mu^t})(s_h, a_h.b_h)\big]$$

$$\leq \sum_{t=1}^{K} \Big(f_h^{t,\mu^t}(s_h, a_h, b_h) - (\mathcal{T}_h^{\mu^t,\nu^t} f_{h+1}^{t,\mu^t})(s_h^t, a_h^t, b_h^t)\Big) + \mathcal{O}(\sqrt{K \log(K/\delta)})$$

$$\leq \mathcal{O}\bigg(\sqrt{V_{\max}^2 K \dim_{\mathrm{BEE}}\Big(\mathcal{F}, \sqrt{1/K}, \Pi, \Pi', \mathcal{Q}^2\Big) \log(\mathcal{N}_{\mathcal{F} \cup \mathcal{G}}(V_{\max}/K) K H |\Pi|/\delta)}\bigg), \quad (28)$$

where the first inequality comes from standard martingale difference concentration. Therefore, combining (27) and (28), we have:

$$\sum_{t=1}^{K} \mathbb{E}_{\pi^t}\big[(f_h^{t,\mu^t} - \mathcal{T}_h^{\mu^t,\nu^t} f_{h+1}^{t,\mu^t})(s_h, a_h.b_h)\big]$$

$$\leq \mathcal{O}\bigg(\sqrt{V_{\max}^2 K \dim_{\mathrm{BEE}}\Big(\mathcal{F}, \sqrt{1/K}, \Pi, \Pi'\Big) \log(\mathcal{N}_{\mathcal{F} \cup \mathcal{G}}(V_{\max}/K) K H |\Pi|/\delta)}\bigg).$$

Substitute the above bounds into (26) and we have:

$$\sum_{t=1}^{K} \overline{V}^t(\mu^t) - V_1^{\pi^t}(s_1)$$

$$\leq \mathcal{O}\bigg(H V_{\max} \sqrt{K \dim_{\mathrm{BEE}}\Big(\mathcal{F}, \sqrt{1/K}, \Pi, \Pi'\Big) \log(\mathcal{N}_{\mathcal{F} \cup \mathcal{G}}(V_{\max}/K) K H |\Pi|/\delta)}\bigg). \quad (29)$$

### I.3 STEP 3: BOUND THE REGRET

Now we only need to bound the online learning error. Notice that $p^t$ is updated using Hedge with reward $\overline{V}^t$. Since $0 \leq \overline{V}^t \leq V_{\max}$ and there are $|\Pi|$ policies, we have from the online learning literature [20] that

$$\sum_{t=1}^{K} \overline{V}^t(\mu^*) - \sum_{t=1}^{K} \langle \overline{V}^t, p^t \rangle \leq V_{\max} \sqrt{K \log |\Pi|}. \quad (30)$$

In addition, suppose $\mathfrak{F}_k$ denotes the filtration induced by $\{\nu^1\} \cup (\cup_{i=1}^k \{\mu^i, \mathcal{D}_i, \nu^{i+1}\})$. Then we can observe that $\langle \overline{V}^t, p^t \rangle - \overline{V}^t(\mu^t) \in \mathfrak{F}_t$. In addition, we have $\overline{V}^t \in \mathfrak{F}_{t-1}$ since the estimation of $\overline{V}^t$ only utilizes $\mathcal{D}_{1:t-1}$, which implies

$$\mathbb{E}[\langle \overline{V}^t, p^t \rangle - \overline{V}^t(\mu^t)|\mathfrak{F}_{t-1}] = 0.$$

Therefore (3) is a martingale difference sequence and by Azuma-Hoeffding's inequality we have with probability at least $1 - \delta/4$,

$$\sum_{t=1}^K \langle \overline{V}^t, p^t \rangle - \sum_{t=1}^K \overline{V}^t(\mu^t) \leq \mathcal{O}(V_{\max}\sqrt{K\log(1/\delta)}) \tag{31}$$

Substituting (23), (29), (30), and (31) into (22) concludes our proof for Theorem 1. For the adaptive setting, we can simply repeat the above arguments. The only difference is that now $\nu^t$ can depend on $\mathcal{D}_{1:t-1}$ and thus we need to introduce a union bound over $\Pi'$ when proving Lemma 2 and Lemma 3. This will incur an additional $\log|\Pi'|$ in $\beta$ and thus also in the regret bound. This concludes our proof.

## J  PROOFS OF LEMMAS IN APPENDIX I

### J.1  PROOF OF LEMMA 2

Let $\mathcal{V}_\rho$ be a $\rho$-cover of $\mathcal{G}$ with respect to $\|\cdot\|_\infty$. Consider an arbitrary fixed tuple $(\mu, t, h, g) \in \Pi \times [K] \times [H] \times \mathcal{G}$. Define $W_{t,k}(h, g, \mu)$ as follows:

$$W_{t,k}(h, g, \mu) := (g_h(s_h^k, a_h^k, b_h^k) - r_h^k - Q_{h+1}^{\mu,\nu^t}(s_{h+1}^k, \mu, \nu^t))^2$$
$$- (Q_h^{\mu,\nu^t}(s_h^k, a_h^k, b_h^k) - r_h^k - Q_{h+1}^{\mu,\nu^t}(s_{h+1}^k, \mu, \nu^t))^2,$$

and $\mathfrak{F}_{k,h}$ be the filtration induced by $\{\nu^1, \cdots, \nu^K\} \cup \{s_1^i, a_1^i, b_1^i, r_1^i, \cdots, s_{H+1}^i\}_{i=1}^{k-1} \cup \{s_1^k, a_1^k, b_1^k, r_1^k, \cdots, s_h^k, a_h^k, b_h^k\}$. Then we have for all $k \leq t-1$,

$$\mathbb{E}[W_{t,k}(h, g, \mu)|\mathfrak{F}_{k,h}] = [(g_h - Q_h^{\mu,\nu^t})(s_h^k, a_h^k, b_h^k)]^2,$$

and

$$\text{Var}[W_{t,k}(h, g, \mu)|\mathfrak{F}_{k,h}] \leq 4V_{\max}^2 \mathbb{E}[W_{t,k}(h, g, \mu)|\mathfrak{F}_{k,h}].$$

By Freedman's inequality, with probability at least $1 - \delta/4$, we have

$$\left| \sum_{k=1}^{t-1} W_{t,k}(h, g, \mu) - \sum_{k=1}^{t-1} [(g_h - Q_h^{\mu,\nu^t})(s_h^k, a_h^k, b_h^k)]^2 \right|$$
$$\leq \mathcal{O}\left( V_{\max}\sqrt{\log\frac{1}{\delta} \cdot \sum_{k=1}^{t-1} [(g_h - Q_h^{\mu,\nu^t})(s_h^k, a_h^k, b_h^k)]^2} + V_{\max}^2 \log\frac{1}{\delta} \right).$$

By taking union bound over $\Pi \times [K] \times [H] \times \mathcal{V}_\rho$ and the non-negativity of $\sum_{k=1}^{t-1} [(g_h - Q_h^{\mu,\nu^t})(s_h^k, a_h^k, b_h^k)]^2$, we have with probability at least $1 - \delta/4$, for all $(\mu, k, h, g) \in \Pi \times [K] \times [H] \times \mathcal{V}_\rho$,

$$-\sum_{k=1}^{t-1} W_{t,k}(h, g, \mu) \leq \mathcal{O}(V_{\max}^2 \iota),$$

where $\iota = \log(HK|\mathcal{V}_\rho||\Pi|/\delta)$. This implies for all $(\mu, t, h, g) \in \Pi \times [K] \times [H] \times \mathcal{G}$,

$$\sum_{k=1}^{t-1} (Q_h^{\mu,\nu^t}(s_h^k, a_h^k, b_h^k) - r_h^k - Q_{h+1}^{\mu,\nu^t}(s_{h+1}^k, \mu, \nu^t))^2$$
$$\leq \sum_{k=1}^{t-1} (g_h(s_h^k, a_h^k, b_h^k) - r_h^k - Q_{h+1}^{\mu,\nu^t}(s_{h+1}^k, \mu, \nu^t))^2 + \mathcal{O}(V_{\max}^2 \iota + V_{\max} t\rho).$$

Choose $\rho = V_{\max}/K$ and we know that with probability at least $1 - \delta$ for all $\mu \in \Pi$ and $t \in [K]$, $Q^{\mu,\nu^t} \in \mathcal{B}_{\mathcal{D}_{1:t-1}}(\mu, \nu^t)$. This concludes our proof.

### J.2 PROOF OF LEMMA 3

Let $\mathcal{Z}_\rho$ be a $\rho$-cover of $\mathcal{F}$ with respect to $\|\cdot\|_\infty$. Consider an arbitrary fixed tuple $(\mu, t, h, f) \in \Pi \times [K] \times [H] \times \mathcal{F}$. Let

$$X_{t,k}(h, f, \mu) := (f_h(s_h^k, a_h^k, b_h^k) - r_h^k - f_{h+1}(s_{h+1}^k, \mu, \nu^t))^2$$
$$- ((\mathcal{T}_h^{\mu,\nu^t} f_{h+1})(s_h^k, a_h^k, b_h^k) - r_h^k - f_{h+1}(s_{h+1}^k, \mu, \nu^t))^2,$$

and $\mathfrak{F}_{k,h}$ be the filtration induced by $\{\nu^1, \cdots, \nu^K\} \cup \{s_1^i, a_1^i, b_1^i, r_1^i, \cdots, s_{H+1}^i\}_{i=1}^{k-1} \cup \{s_1^k, a_1^k, b_1^k, r_1^k, \cdots, s_h^k, a_h^k, b_h^k\}$. Then we have for all $k \le t - 1$,

$$\mathbb{E}[X_{t,k}(h, f, \mu) | \mathfrak{F}_{k,h}] = [(f_h - \mathcal{T}_h^{\mu,\nu^t} f_{h+1})(s_h^k, a_h^k, b_h^k)]^2,$$

and

$$\mathrm{Var}[X_{t,k}(h, f, \mu) | \mathfrak{F}_{k,h}] \le 4 V_{\max}^2 \mathbb{E}[X_{t,k}(h, f, \mu) | \mathfrak{F}_{k,h}].$$

By Freedman's inequality, with probability at least $1 - \delta$,

$$\left| \sum_{k=1}^{t-1} X_{t,k}(h, f, \mu) - \sum_{k=1}^{t-1} [(f_h - \mathcal{T}_h^{\mu,\nu^t} f_{h+1})(s_h^k, a_h^k, b_h^k)]^2 \right|$$
$$\le \mathcal{O}\left( V_{\max} \sqrt{\log \frac{1}{\delta} \cdot \sum_{k=1}^{t-1} [(f_h - \mathcal{T}_h^{\mu,\nu^t} f_{h+1})(s_h^k, a_h^k, b_h^k)]^2} + V_{\max}^2 \log \frac{1}{\delta} \right).$$

By taking union bound over $\Pi \times [K] \times [H] \times \mathcal{Z}_\rho$, we have with probability at least $1 - \delta$, for all $(\mu, t, h, f) \in \Pi \times [K] \times [H] \times \mathcal{Z}_\rho$,

$$\left| \sum_{k=1}^{t-1} X_{t,k}(h, f, \mu) - \sum_{k=1}^{t-1} [(f_h - \mathcal{T}_h^{\mu,\nu^t} f_{h+1})(s_h^k, a_h^k, b_h^k)]^2 \right|$$
$$\le \mathcal{O}\left( V_{\max} \sqrt{\iota \cdot \sum_{k=1}^{t-1} [(f_h - \mathcal{T}_h^{\mu,\nu^t} f_{h+1})(s_h^k, a_h^k, b_h^k)]^2} + V_{\max}^2 \iota \right). \tag{32}$$

where $\iota = \log(HK|\mathcal{Z}_\rho||\Pi|/\delta)$.

Conditioned on the above event being true, we consider an arbitrary pair $(h, t, \mu) \in [H] \times [K] \times \Pi$. By the definition of $\mathcal{B}_{\mathcal{D}_{1:t-1}}(\mu, \nu^t)$ and Assumption 2, we have:

$$\sum_{k=1}^{t-1} X_{t,k}(h, f^{t,\mu}, \mu) = \sum_{k=1}^{t-1} (f_h(s_h^k, a_h^k, b_h^k) - r_h^k - f_{h+1}(s_{h+1}^k, \mu, \nu^t))^2$$
$$- ((\mathcal{T}_h^{\mu,\nu^t} f_{h+1})(s_h^k, a_h^k, b_h^k) - r_h^k - f_{h+1}(s_{h+1}^k, \mu, \nu^t))^2$$
$$\le \sum_{k=1}^{t-1} (f_h(s_h^k, a_h^k, b_h^k) - r_h^k - f_{h+1}(s_{h+1}^k, \mu, \nu^t))^2$$
$$- \inf_{g \in \mathcal{G}} (g_h(s_h^k, a_h^k, b_h^k) - r_h^k - f_{h+1}(s_{h+1}^k, \mu, \nu^t))^2$$
$$\le \beta.$$

Let $l^{t,\mu} = \arg\min_{l \in \mathcal{Z}_\rho} \max_{h \in [H]} \|f_h^{t,\mu} - l_h^{t,\mu}\|_\infty$. By the definition of $\mathcal{Z}_\rho$, we have

$$\sum_{k=1}^{t-1} X_{t,k}(h, l^{t,\mu}, \mu) \le \mathcal{O}(V_{\max} t \rho + \beta). \tag{33}$$

By (32), we know:

$$\left| \sum_{k=1}^{t-1} X_{t,k}(h, l^{t,\mu}, \mu) - \sum_{k=1}^{t-1} [(l_h^{t,\mu} - \mathcal{T}_h^{\mu,\nu^t} l_{h+1}^{t,\mu})(s_h^k, a_h^k, b_h^k)]^2 \right|$$

$$\leq \mathcal{O}\left(V_{\max}\sqrt{\iota \cdot \sum_{k=1}^{t-1}[(l_h^{t,\mu} - \mathcal{T}_h^{\mu,\nu^t}l_{h+1}^{t,\mu})(s_h^k, a_h^k, b_h^k)]^2 + V_{\max}^2\iota}\right). \tag{34}$$

Combining (33) and (34), we obtain

$$\sum_{k=1}^{t-1}[(l_h^{t,\mu} - \mathcal{T}_h^{\mu,\nu^t}l_{h+1}^{t,\mu})(s_h^k, a_h^k, b_h^k)]^2 \leq \mathcal{O}(V_{\max}^2\iota + V_{\max}t\rho + \beta).$$

This implies that

$$\sum_{k=1}^{t-1}[(f_h^{t,\mu} - \mathcal{T}_h^{\mu,\nu^t}f_{h+1}^{t,\mu})(s_h^k, a_h^k, b_h^k)]^2 \leq \mathcal{O}(V_{\max}^2\iota + V_{\max}t\rho + \beta).$$

Choose $\rho = V_{\max}/K$ and we can obtain (b). For (a), simply let $\mathfrak{F}_{k,h}$ be the filtration induced by $\{\nu^1, \cdots, \nu^K\} \cup \{\mu^i, s_1^i, a_1^i, b_1^i, r_1^i, \cdots, s_{H+1}^i\}_{i=1}^{k-1} \cup \mu^k$ and repeat the above arguments, which concludes our proof.

### J.3 Proof of Lemma 4

First notice that $\overline{V}^t(\mu^t) = f_1^{t,\mu^t}(s_1, \mu^t, \nu^t)$. Therefore, we have

$$\overline{V}^t(\mu^t) - V_1^{\pi^t}(s_1) = \mathbb{E}_{a_1\sim\mu^t(\cdot|s_1),b_1\sim\nu^t(\cdot|s_1)}[f_1^{t,\mu^t}(s_1, a_1, b_1) - Q_1^{\pi^t}(s_1, a_1, b_1)]$$

$$= \mathbb{E}_{a_1\sim\mu^t(\cdot|s_1),b_1\sim\nu^t(\cdot|s_1)}\big[\mathbb{E}_{s_2\sim P_1(\cdot|s_1,a_1,b_1)}[f_2^{t,\mu^t}(s_2, \mu^t, \nu^t)] - \mathbb{E}_{s_2\sim P_1(\cdot|s_1,a_1,b_1)}[V_2^{\pi^t}(s_2)]\big]$$

$$+ \mathbb{E}_{a_1\sim\mu^t(\cdot|s_1),b_1\sim\nu^t(\cdot|s_1)}[(f_1^{t,\mu^t} - \mathcal{T}_1^{\mu^t,\nu^t}f_2^{t,\mu^t})(s_1, a_1, b_1)]$$

$$= \mathbb{E}_{s_2\sim\pi^t}[f_2^{t,\mu^t}(s_2, \mu^t, \nu^t) - V_2^{\pi^t}(s_2)] + \mathbb{E}_{\pi^t}[(f_1^{t,\mu^t} - \mathcal{T}_1^{\mu^t,\nu^t}f_2^{t,\mu^t})(s_1, a_1, b_1)].$$

Repeat the above procedures and we can obtain Lemma 4. This concludes our proof.

## K Proof of Corollary 1

From Theorem 2, we have with probability at least $1 - \delta$, for all $i \in [n]$

$$\max_{\mu_i\in\Pi_i} \frac{1}{K}\sum_{t=1}^K V_{1,i}^{\mu_i\times\mu_{-i}^t}(s_1) \leq \frac{1}{K}\sum_{t=1}^K V_{1,i}^{\mu_i^t\times\mu_{-i}^t}(s_1) + \epsilon.$$

By the definition of $\widehat{\pi}$, this is equivalent to

$$\max_{\mu_i\in\Pi_i} V_{1,i}^{\mu_i\times\widehat{\mu}_{-i}}(s_1) \leq V_{1,i}^{\widehat{\pi}}(s_1) + \epsilon,$$

where $\widehat{\mu}_{-i}$ is uniformly sampled from $\{\mu_{-i}^t\}_{t=1}^K$ and thus is the marginal distribution of $\widehat{\pi}$ over the agents other than $i$. Therefore, by the definition of CCE in (2), $\widehat{\pi}$ is $\epsilon$-approximate CCE with probability at least $1 - \delta$, which concludes our proof.

## L Proof of Theorem 3

In this section we present the proof for Theorem 3. Our proof mainly consists of four steps:

- Prove $\overline{V}_r^t(\mu)$ and $\overline{V}_g^t(\mu)$ are optimistic estimations of $V_{r,1}^\mu(s_1)$ and $V_{g,1}^\mu(s_1)$ for all $t \in [K]$ and $\mu \in \Pi$.
- Bound the total estimation error $\sum_{t=1}^K \overline{V}_r^t(\mu^t) - V_{r,1}^{\mu^t}(s_1)$ and $\sum_{t=1}^K \overline{V}_g^t(\mu^t) - V_{g,1}^{\mu^t}(s_1)$.
- Bound the regret by decomposing it into estimation error and online learning error induced by Hedge.
- Bound the constraint violation by strong duality.

**Step 1: Prove optimism.** First we can show that the constructed set $\mathcal{B}_{\mathcal{D}^r_{1:t-1}}(\mu)$ ($\mathcal{B}_{\mathcal{D}^g_{1:t-1}}(\mu)$) is not vacuous in the sense that the true action-value function $Q^\mu_r$ ($Q^\mu_g$) belongs to it with high probability:

**Lemma 6.** *With probability at least $1 - \delta/4$, we have for all $t \in [K]$ and $\mu \in \Pi$,*

$$Q^\mu_r \in \mathcal{B}_{\mathcal{D}^r_{1:t-1}}(\mu), Q^\mu_g \in \mathcal{B}_{\mathcal{D}^g_{1:t-1}}(\mu).$$

*Proof.* The proof is almost the same as Lemma 2 and thus is omitted here. $\qquad\square$

Then since $\overline{V}^t_r(\mu) = \max_{f \in \mathcal{B}_{\mathcal{D}^g_{1:t-1}}(\mu)} f(s_1, \mu)$, we know for all $t \in [K]$ and $\mu \in \Pi$,

$$\overline{V}^t_r(\mu) \geq Q^\mu_r(s_1, \mu) = V^\mu_{r,1}(s_1).$$

Similarly, we know $\overline{V}^t_g(\mu) \geq V^\mu_{g,1}(s_1)$.

**Step 2: Bound estimation error.** Next we need to show the estimation error $\sum_{t=1}^K \overline{V}^t_r(\mu^t) - V^{\mu^t}_{r,1}(s_1)$ and $\sum_{t=1}^K \overline{V}^t_g(\mu^t) - V^{\mu^t}_{g,1}(s_1)$ are small. Let $f^{t,\mu,r} = \arg\max_{f \in \mathcal{B}_{\mathcal{D}^r_{1:t-1}}(\mu)} f(s_1, \mu)$ and $f^{t,\mu,g} = \arg\max_{f \in \mathcal{B}_{\mathcal{D}^g_{1:t-1}}(\mu)} f(s_1, \mu)$. Then we have

**Lemma 7.** *With probability at least $1 - \delta/4$, we have for all $t \in [K]$, $h \in [H]$ and $\mu \in \Pi$,*

$$(a) \quad \sum_{k=1}^{t-1} \mathbb{E}_{\mu^k} \left[ \left( f^{t,\mu,r}_h(s_h, a_h) - (\mathcal{T}^{\mu,r}_h f^{t,\mu,r}_{h+1})(s_h, a_h) \right)^2 \right] \leq \mathcal{O}(\beta_r),$$

$$\sum_{k=1}^{t-1} \mathbb{E}_{\mu^k} \left[ \left( f^{t,\mu,g}_h(s_h, a_h) - (\mathcal{T}^{\mu,g}_h f^{t,\mu,g}_{h+1})(s_h, a_h) \right)^2 \right] \leq \mathcal{O}(\beta_g),$$

$$(b) \quad \sum_{k=1}^{t-1} \left( f^{t,\mu,r}_h(s^k_h, a^k_h) - (\mathcal{T}^{\mu,r}_h f^{t,\mu,r}_{h+1})(s^k_h, a^k_h) \right)^2 \leq \mathcal{O}(\beta_r),$$

$$\sum_{k=1}^{t-1} \left( f^{t,\mu,g}_h(s^k_h, a^k_h) - (\mathcal{T}^{\mu,g}_h f^{t,\mu,g}_{h+1})(s^k_h, a^k_h) \right)^2 \leq \mathcal{O}(\beta_g).$$

*Proof.* The proof is almost the same as Lemma 3 and thus is omitted here. $\qquad\square$

Besides, using performance difference lemma we can easily bridge $\overline{V}^t_r(\mu^t) - V^{\mu^t}_{r,1}(s_1)$ and $\overline{V}^t_g(\mu^t) - V^{\mu^t}_{g,1}(s_1)$ with Bellman residuals, whose proof is also omitted:

**Lemma 8.** *For any $t \in [K]$, we have*

$$\overline{V}^t_r(\mu^t) - V^{\mu^t}_{r,1}(s_1) = \sum_{h=1}^H \mathbb{E}_{\mu^t}[(f^{t,\mu^t,r}_h - \mathcal{T}^{\mu^t,r} f^{t,\mu^t,r}_{h+1})(s_h, a_h)],$$

$$\overline{V}^t_g(\mu^t) - V^{\mu^t}_{g,1}(s_1) = \sum_{h=1}^H \mathbb{E}_{\mu^t}[(f^{t,\mu^t,g}_h - \mathcal{T}^{\mu^t,g} f^{t,\mu^t,g}_{h+1})(s_h, a_h)].$$

Therefore, from Lemma 8 we can obtain for any $t \in [K]$,

$$\overline{V}^t_r(\mu^t) - V^{\mu^t}_{r,1}(s_1) = \sum_{h=1}^H \mathbb{E}_{\mu^t}[(f^{t,\mu^t,r}_h - \mathcal{T}^{\mu^t,r} f^{t,\mu^t,r}_{h+1})(s_h, a_h)],$$

which implies

$$\sum_{t=1}^K \overline{V}^t_r(\mu^t) - V^{\mu^t}_{r,1}(s_1) = \sum_{h=1}^H \sum_{t=1}^K \mathbb{E}_{\mu^t}[(f^{t,\mu^t,r}_h - \mathcal{T}^{\mu^t,r} f^{t,\mu^t,r}_{h+1})(s_h, a_h)]. \tag{35}$$

Similar to Section I, from Lemma 5, conditioning on the event in Lemma 7 holds true, we have with probability at least $1 - \delta/4$

$$\sum_{t=1}^{K} \mathbb{E}_{\mu^t}[(f_h^{t,\mu^t,r} - \mathcal{T}_h^{\mu^t,r} f_{h+1}^{t,\mu^t,r})(s_h, a_h)]$$
$$\leq \mathcal{O}\left(\sqrt{H^2 K \dim_{\text{BEE}}\left(\mathcal{F}, \sqrt{1/K}, \Pi, r\right) \log(\mathcal{N}_{\mathcal{F}^r \cup \mathcal{G}^r}(H/K)KH|\Pi|/\delta)}\right).$$

Substitute the above bounds into (35) and we have:

$$\sum_{t=1}^{K} \overline{V}_r^t(\mu^t) - V_{r,1}^{\mu^t}(s_1)$$
$$\leq \mathcal{O}\left(H^2 \sqrt{K \dim_{\text{BEE}}\left(\mathcal{F}^r, \sqrt{1/K}, \Pi, r\right) \log(\mathcal{N}_{\mathcal{F}^r \cup \mathcal{G}^r}(H/K)KH|\Pi|/\delta)}\right). \tag{36}$$

Similarly, we have

$$\sum_{t=1}^{K} \overline{V}_g^t(\mu^t) - V_{g,1}^{\mu^t}(s_1)$$
$$\leq \mathcal{O}\left(H^2 \sqrt{K \dim_{\text{BEE}}\left(\mathcal{F}^g, \sqrt{1/K}, \Pi, g\right) \log(\mathcal{N}_{\mathcal{F}^g \cup \mathcal{G}^g}(H/K)KH|\Pi|/\delta)}\right). \tag{37}$$

**Step 3: Bound the regret.** Now we can bound the regret. We first decompose the fictitious total regret $\sum_{t=1}^{K}(V_{r,1}^{\mu^*_{\text{CMDP}}}(s_1) + Y_t \overline{V}_g^t(\mu^*_{\text{CMDP}})) - \sum_{t=1}^{K}(V_{r,1}^{\mu^t}(s_1) + Y_t \overline{V}_g^t(\mu^t))$ to the following terms:

$$\sum_{t=1}^{K}(V_{r,1}^{\mu^*_{\text{CMDP}}}(s_1) + Y_t \overline{V}_g^t(\mu^*_{\text{CMDP}})) - \sum_{t=1}^{K}(V_{r,1}^{\mu^t}(s_1) + Y_t \overline{V}_g^t(\mu^t))$$
$$= \underbrace{\left(\sum_{t=1}^{K} V_{r,1}^{\mu^*_{\text{CMDP}}}(s_1) - \sum_{t=1}^{K} \overline{V}_r^t(\mu^*_{\text{CMDP}})\right)}_{(1)}$$
$$+ \underbrace{\left(\sum_{t=1}^{K}(\overline{V}_r^t(\mu^*_{\text{CMDP}}) + Y_t \overline{V}_g^t(\mu^*_{\text{CMDP}})) - \sum_{t=1}^{K}\langle \overline{V}_r^t + Y_t \overline{V}_g^t, p^t\rangle\right)}_{(2)}$$
$$+ \underbrace{\left(\sum_{t=1}^{K}\langle \overline{V}_r^t + Y_t \overline{V}_g^t, p^t\rangle - \sum_{t=1}^{K}(\overline{V}_r^t(\mu^t) + Y_t \overline{V}_g^t(\mu^t))\right)}_{(3)}$$
$$+ \underbrace{\left(\sum_{t=1}^{K} \overline{V}_r^t(\mu^t) - \sum_{t=1}^{K} V_{r,1}^{\mu^t}(s_1)\right)}_{(4)}.$$

From Lemma 6, we know $(1) \leq 0$. Since $p^t$ is updated using Hedge with loss function $\overline{V}^t$, we have $(2) \leq H(1 + \chi)\sqrt{K \log |\Pi|}$. $(3)$ is a martingale difference sequence, which implies $(3) \leq \mathcal{O}(H(1 + \chi)\sqrt{K \log(1/\delta)})$ with probability at least $1 - \delta/4$. Finally, Step 2 has bounded term $(4)$ in (36), which implies

$$\sum_{t=1}^{K}(V_{r,1}^{\mu^*_{\text{CMDP}}}(s_1) + Y_t \overline{V}_g^t(\mu^*_{\text{CMDP}})) - \sum_{t=1}^{K}(V_{r,1}^{\mu^t}(s_1) + Y_t \overline{V}_g^t(\mu^t))$$

$$\leq \mathcal{O}\left(\left(H^2 + \frac{H^2}{\lambda_{\text{sla}}}\right)\sqrt{Kd_{\text{BEE},r}\log\left(\mathcal{N}_{\text{cov},r}|\Pi|/\delta\right)}\right). \tag{38}$$

Now we only need to bound $-\sum_{t=1}^{K}Y_t(\overline{V}_g^t(\mu_{\text{CMDP}}^*) - \overline{V}_g^t(\mu^t))$ if we want to bound the regret $\sum_{t=1}^{K}(V_{r,1}^{\mu_{\text{CMDP}}^*}(s_1) - V_{r,1}^{\mu^t}(s_1))$. In fact, updating the dual variable $Y^t$ with projected gradient descent guarantees us the following lemma:

**Lemma 9.** *Suppose the events in Lemma 6 hold true, we have*

$$-\sum_{t=1}^{K}Y_t(\overline{V}_g^t(\mu_{CMDP}^*) - \overline{V}_g^t(\mu^t)) \leq \frac{\alpha H^2 K}{2} = \frac{H^2\sqrt{K}}{2}.$$

*Proof.* See Appendix L.2. $\qquad\square$

Substituting Lemma 9 into (38), we can obtain the bound on Regret$(K)$:

$$\sum_{t=1}^{K}(V_{r,1}^{\mu_{\text{CMDP}}^*}(s_1) - V_{r,1}^{\mu^t}(s_1)) \leq \mathcal{O}\left(\left(H^2 + \frac{H^2}{\lambda_{\text{sla}}}\right)\sqrt{Kd_{\text{BEE},r}\log\left(\mathcal{N}_{\text{cov},r}|\Pi|/\delta\right)}\right).$$

**Step 4: Constraint Violation Analysis.** Next we need to bound the constraint violation. First notice that $\sum_{t=1}^{K}Y_t(b - \overline{V}_g^t(\mu^t))$ is indeed not far from $\sum_{t=1}^{K}Y(b - \overline{V}_g^t(\mu^t))$ for any $Y \in [0, \chi]$, as shown in the following lemma whose proof is deferred to Appendix L.3:

**Lemma 10.** *For any $Y \in [0, \chi]$, we have*

$$\sum_{t=1}^{K}(Y - Y_t)(b - \overline{V}_g^t(\mu^t)) \leq \frac{(H^2 + \chi^2)\sqrt{K}}{2}.$$

Substituting Lemma 10 into (38) and notice that $b \leq V_{g,1}^{\mu_{\text{CMDP}}^*}(s_1) \leq \overline{V}_g^t(\mu_{\text{CMDP}}^*)$, we have for any $Y \in [0, \chi]$,

$$\sum_{t=1}^{K}(V_{r,1}^{\mu_{\text{CMDP}}^*}(s_1) - V_{r,1}^{\mu^t}(s_1)) + Y\sum_{t=1}^{K}(b - \overline{V}_g^t(\mu^t))$$

$$\leq \mathcal{O}\left(\left(H^2 + \frac{H^2}{\lambda_{\text{sla}}^2}\right)\sqrt{Kd_{\text{BEE},r}\log\left(\mathcal{N}_{\text{cov},r}|\Pi|/\delta\right)}\right).$$

Combining the above inequality with (37), we have

$$\sum_{t=1}^{K}(V_{r,1}^{\mu_{\text{CMDP}}^*}(s_1) - V_{r,1}^{\mu^t}(s_1)) + Y\sum_{t=1}^{K}(b - V_{g,1}^{\mu^t}(s_1)) \leq \mathcal{O}\left(\left(\frac{H^2}{\lambda_{\text{sla}}^2} + \frac{H^3}{\lambda_{\text{sla}}}\right)\sqrt{K\epsilon_{\text{BEE}}}\right),$$

where

$$\epsilon_{\text{BEE}} = \max\left\{\dim_{\text{BEE}}\left(\mathcal{F}^r, \sqrt{1/K}, \Pi, r\right)\log(\mathcal{N}_{\mathcal{F}^r \cup \mathcal{G}^r}(H/K)KH|\Pi|/\delta),\right.$$

$$\left.\dim_{\text{BEE}}\left(\mathcal{F}^g, \sqrt{1/K}, \Pi, g\right)\log(\mathcal{N}_{\mathcal{F}^g \cup \mathcal{G}^g}(H/K)KH|\Pi|/\delta)\right\}.$$

Choose $Y$ as

$$Y = \begin{cases} 0 & \text{if } \sum_{t=1}^{K}(b - V_{g,1}^{\mu^t}(s_1)) < 0, \\ \chi & \text{otherwise.} \end{cases}$$

then we can bound the summation of regret and constraint violation as follows:

$$\left(V_{r,1}^{\mu_{\text{CMDP}}^*}(s_1) - \frac{1}{K}\sum_{t=1}^{K}V_{r,1}^{\mu^t}(s_1)\right) + \chi\left[b - \frac{1}{K}\sum_{t=1}^{K}V_{g,1}^{\mu^t}(s_1)\right]_+ \leq \mathcal{O}\left(\left(\frac{H^2}{\lambda_{\text{sla}}^2} + \frac{H^3}{\lambda_{\text{sla}}}\right)\sqrt{\epsilon_{\text{BEE}}/K}\right). \tag{39}$$

Further, when Assumption 6 and Assumption 7 hold, we have the following lemma showing that an upper bound on $(V_{r,1}^{\mu_{\text{CMDP}}^*}(s_1) - \frac{1}{K}\sum_{t=1}^{K} V_{r,1}^{\mu^t}(s_1)) + \chi[b - \frac{1}{K}\sum_{t=1}^{K} V_{g,1}^{\mu^t}(s_1)]_+$ implies an upper bound on $[b - \frac{1}{K}\sum_{t=1}^{K} V_{g,1}^{\mu^t}(s_1)]_+$:

**Lemma 11.** *Suppose Assumption 6 and Assumption 7 hold and $2Y^* \leq C^*$. If $\{\mu^t\}_{t=1}^{K} \subseteq \Pi$ satisfies*

$$\left(V_{r,1}^{\mu_{\text{CMDP}}^*}(s_1) - \frac{1}{K}\sum_{t=1}^{K} V_{r,1}^{\mu^t}(s_1)\right) + C^*\left[b - \frac{1}{K}\sum_{t=1}^{K} V_{r,1}^{\mu^t}(s_1)\right]_+ \leq \delta,$$

*Then*

$$\left[b - \frac{1}{K}\sum_{t=1}^{K} V_{r,1}^{\mu^t}(s_1)\right]_+ \leq \frac{2\delta}{C^*}.$$

See Appendix L.4 for the proof. Combining Lemma 11, Lemma 1 and (39), we have

$$\left[\sum_{t=1}^{K}(b - V_{g,1}^{\mu^t}(s_1))\right]_+ \leq \mathcal{O}\left(\left(H^2 + \frac{H}{\lambda_{\text{sla}}}\right)\sqrt{K\epsilon_{\text{BEE}}}\right).$$

This concludes our proof.

## L.1 PROOF OF LEMMA 1

Notice that $D(Y^*) = V_{r,1}^{\mu_{\text{CMDP}}^*}(s_1)$, which suggests:

$$V_{r,1}^{\mu_{\text{CMDP}}^*}(s_1) = D(Y^*) \geq \mathcal{L}_{\text{CMDP}}(\widetilde{\mu}, Y^*)$$
$$= V_{r,1}^{\widetilde{\mu}}(s_1) + Y^*(V_{g,1}^{\widetilde{\mu}}(s_1) - b) \geq V_{r,1}^{\widetilde{\mu}}(s_1) + Y^*\lambda_{\text{sla}}.$$

This implies that

$$Y^* \leq \frac{V_{r,1}^{\mu_{\text{CMDP}}^*}(s_1) - V_{r,1}^{\widetilde{\mu}}(s_1)}{\lambda_{\text{sla}}} \leq \frac{H}{\lambda_{\text{sla}}},$$

which concludes our proof.

## L.2 PROOF OF LEMMA 9

Notice that we have:

$$0 \leq Y_{K+1}^2 = \sum_{t=1}^{K}\left(Y_{t+1}^2 - Y_t^2\right)$$
$$= \sum_{t=1}^{K}\left(\left(\text{Proj}_{[0,\chi]}(Y_t + \alpha(b - \overline{V}_g^t(\mu^t)))\right)^2 - Y_t^2\right)$$
$$\leq \sum_{t=1}^{K}\left((Y_t + \alpha(b - \overline{V}_g^t(\mu^t)))^2 - Y_t^2\right)$$
$$= \sum_{t=1}^{K} 2\alpha Y_t(b - \overline{V}_g^t(\mu^t)) + \sum_{t=1}^{K}\alpha^2(b - \overline{V}_g^t(\mu^t))^2$$
$$\leq \sum_{t=1}^{K} 2\alpha Y_t(\overline{V}_g^t(\mu_{\text{CMDP}}^*) - \overline{V}_g^t(\mu^t)) + \alpha^2 KH^2,$$

where the last step is due to optimism and $V_{g,1}^{\mu_{\text{CMDP}}^*}(s_1) \geq b$. This implies that

$$-\sum_{t=1}^{K} Y_t(\overline{V}_g^t(\mu_{\text{CMDP}}^*) - \overline{V}_g^t(\mu^t)) \leq \frac{\alpha H^2 K}{2} = \frac{H^2\sqrt{K}}{2}.$$

This concludes our proof.

### L.3  PROOF OF LEMMA 10

Notice that we have for any $t \in [K]$ and $Y \in [0, \chi]$:

$$
\begin{aligned}
|Y_{t+1} - Y|^2 &\le |Y_t + \alpha(b - \overline{V}_g^t(\mu^t)) - Y|^2 \\
&= (Y_t - Y)^2 + 2\alpha(b - \overline{V}_g^t(\mu^t))(Y_t - Y) + \alpha^2 H^2.
\end{aligned}
$$

Repeating the above expansion procedures, we have

$$
0 \le |Y_{K+1} - Y|^2 \le (Y_1 - Y)^2 + 2\alpha \sum_{t=1}^{K}(b - \overline{V}_g^t(\mu^t))(Y_t - Y) + \alpha^2 H^2 K,
$$

which is equivalent to

$$
\sum_{t=1}^{K}(b - \overline{V}_g^t(\mu^t))(Y - Y_t) \le \frac{1}{2\alpha}(Y_1 - Y)^2 + \frac{\alpha}{2}H^2 K \le \frac{(H^2 + \chi^2)\sqrt{K}}{2}.
$$

This concludes our proof.

### L.4  PROOF OF LEMMA 11

First we extend $\Pi$ in a reasonable way to make the policy class more structured while not changing its optimal policy. Define the set of state-action visitation distributions induced by the policy $\Pi$ as follows:

$$
\mathcal{P}_{\Pi} = \{(d_h^\mu(s,a))_{h \in [H], s \in \mathcal{S}, a \in \mathcal{A}} \in (\Delta_{|\mathcal{S}| \times |\mathcal{A}|})^H : \mu \in \Pi\}.
$$

Let $\mathrm{conv}(\mathcal{P}_{\Pi})$ denote the convex hull of $\mathcal{P}_{\Pi}$, i.e., for any $d \in \mathrm{conv}(\mathcal{P}_{\Pi})$, there exists $\{w_\mu\}_{\mu \in \Pi} \ge 0$ such that for any $h \in [H], s \in \mathcal{S}. a \in \mathcal{A}$, we have

$$
d_h(s,a) = \sum_{\mu \in \Pi} w_\mu d_h^\mu(s,a), \sum_{\mu \in \Pi} w_\mu = 1.
$$

As a special case, there exists $d_h'(s,a) \in \mathrm{conv}(\mathcal{P}_{\Pi})$ such that for any $h \in [H], s \in \mathcal{S}. a \in \mathcal{A}$,

$$
d_h'(s,a) = \frac{1}{K}\sum_{t=1}^{K} d_h^{\mu^t}(s,a).
$$

Notice that there exists a one-to-one mapping from state-action visitation distributions to policies [41]. Let $\mathrm{conv}(\Pi)$ denote the policy class that induces $\mathrm{conv}(\mathcal{P}_{\Pi})$, and then there exists $\mu'$ such that $d' = d^{\mu'}$, which implies

$$
V_{r,1}^{\mu'}(s_1) = \frac{1}{K}\sum_{t=1}^{K} V_{r,1}^{\mu^t}(s_1), V_{g,1}^{\mu'}(s_1) = \frac{1}{K}\sum_{t=1}^{K} V_{g,1}^{\mu^t}(s_1).
$$

Therefore, the condition of this lemma says

$$
(V_{r,1}^{\mu_{\mathrm{CMDP}}^*}(s_1) - V_{r,1}^{\mu'}(s_1)) + C^*[b - V_{g,1}^{\mu'}(s_1)]_+ \le \delta. \tag{40}
$$

Next we show that $\mu_{\mathrm{CMDP}}^*$ is still the optimal policy in $\mathrm{conv}(\Pi)$ when Assumption 6, i.e., strong duality, holds. First notice that

$$
\max_{\mu \in \mathrm{conv}(\Pi)} \min_{Y \ge 0} \mathcal{L}_{\mathrm{CMDP}}(\mu, Y) \le \min_{Y \ge 0} \max_{\mu \in \mathrm{conv}(\Pi)} \mathcal{L}_{\mathrm{CMDP}}(\mu, Y) = \min_{Y \ge 0} \max_{d \in \mathrm{conv}(\mathcal{P}_{\Pi})} \mathcal{L}_{\mathrm{CMDP}}(d, Y). \tag{41}
$$

However, given $Y \ge 0, \mathcal{L}_{\mathrm{CMDP}}(d, Y)$ is linear in $d$, which means the maximum is always attained at the vertices of $\mathrm{conv}(\mathcal{P}_{\Pi})$, i.e., $\mathcal{P}_{\Pi}$. Therefore we know

$$
\max_{\mu \in \mathrm{conv}(\Pi)} \mathcal{L}_{\mathrm{CMDP}}(\mu, Y) = D(Y),
$$

which suggests

$$\min_{Y \geq 0} \max_{d \in \text{conv}(\mathcal{P}_\Pi)} \mathcal{L}_{\text{CMDP}}(d, Y) = \min_{Y \geq 0} \max_{d \in \mathcal{P}_\Pi} \mathcal{L}_{\text{CMDP}}(d, Y) = \min_{Y \geq 0} \max_{\mu \in \Pi} \mathcal{L}_{\text{CMDP}}(\mu, Y). \quad (42)$$

By strong duality, we have

$$\min_{Y \geq 0} \max_{\mu \in \Pi} \mathcal{L}_{\text{CMDP}}(\mu, Y) = \max_{\mu \in \Pi} \min_{Y \geq 0} \mathcal{L}_{\text{CMDP}}(\mu, Y) \leq \max_{\mu \in \text{conv}(\Pi)} \min_{Y \geq 0} \mathcal{L}_{\text{CMDP}}(\mu, Y). \quad (43)$$

Combining (41),(42) and (43), we know all the inequalities have to take equality, which implies

$$\mu^*_{\text{CMDP}} = \arg \max_{\mu \in \text{conv}(\Pi)} \min_{Y \geq 0} \mathcal{L}_{\text{CMDP}}(\mu, Y), Y^* = \arg \min_{Y \geq 0} \max_{\mu \in \text{conv}(\Pi)} \mathcal{L}_{\text{CMDP}}(\mu, Y).$$

Besides, strong duality also holds for $\max_{\mu \in \text{conv}(\Pi)} \min_{Y \geq 0} \mathcal{L}_{\text{CMDP}}(\mu, Y)$.

Now let $v(\tau) := \max_{\mu \in \text{conv}(\Pi)} \{V^\mu_{r,1}(s_1) | V^\mu_{g,1}(s_1) \geq b + \tau\}$, then we have for any $\mu \in \text{conv}(\Pi)$,

$$\mathcal{L}_{\text{CMDP}}(\mu, Y^*) \leq \max_{\mu \in \text{conv}(\Pi)} \mathcal{L}_{\text{CMDP}}(\mu, Y^*) = D(Y^*) = V^{\mu^*_{\text{CMDP}}}_{r,1}(s_1),$$

where the third step comes from strong duality. Therefore, for any $\mu \in \text{conv}(\Pi)$ and $\tau \in \mathbb{R}$ which satisfies $V^\mu_{g,1}(s_1) \geq b + \tau$, we have

$$V^{\mu^*_{\text{CMDP}}}_{r,1}(s_1) - \tau Y^* \geq \mathcal{L}_{\text{CMDP}}(\mu, Y^*) - \tau Y^*$$
$$= V^\mu_{r,1}(s_1) + Y^*(V^\mu_{g,1}(s_1) - b - \tau) \geq V^\mu_{r,1}(s_1).$$

This implies that for any $\tau \in \mathbb{R}$, $V^{\mu^*_{\text{CMDP}}}_{r,1}(s_1) - \tau Y^* \geq v(\tau)$. Pick $\tau = \widetilde{\tau} := -[b - V^{\mu'}_{g,1}(s_1)]_+$, then we have

$$V^{\mu'}_{r,1}(s_1) - V^{\mu^*_{\text{CMDP}}}_{r,1}(s_1) \leq -\widetilde{\tau} Y^*.$$

On the other hand, (40) is equivalent to

$$V^{\mu^*_{\text{CMDP}}}_{r,1}(s_1) - V^{\mu'}_{r,1}(s_1) - C^* \widetilde{\tau} \leq \delta.$$

Thus we have $(C^* - Y^*)|\widetilde{\tau}| \leq \delta$, which means that

$$[b - V^{\mu'}_{g,1}(s_1)]_+ \leq \frac{\delta}{C^* - Y^*} \leq \frac{2\delta}{C^*}.$$

Recall that $V^{\mu'}_{g,1}(s_1) = \frac{1}{K} \sum_{t=1}^K V^{\mu^t}_{g,1}(s_1)$, which concludes our proof.

## M    PROOF OF THEOREM 4

In this section we present the proof for Theorem 4. Our proof mainly consists of four steps:

- Prove $\langle \underline{\boldsymbol{V}}^t(\mu), \boldsymbol{\theta}_t \rangle$ is a pessimistic estimations of $\langle \boldsymbol{V}^\mu_1(s_1), \boldsymbol{\theta}_t \rangle$ for all $t \in [K]$ and $\mu \in \Pi$.

- Bound the total estimation error $\| \frac{1}{K} \sum_{t=1}^K \underline{\boldsymbol{V}}^t(\mu^t) - \boldsymbol{V}^{\mu^t}_1(s_1) \|$.

- Bound $\text{dist}(\boldsymbol{V}^{\widehat{\mu}}_1(s_1), \mathcal{C})$.

**Step 1: Prove pessimism.**    First we can show that the true action-value function $\boldsymbol{Q}^\mu$ belongs to the constructed set $\mathcal{B}_{\mathcal{D}_{1:t-1}}(\mu)$ with high probability:

**Lemma 12.** *With probability at least $1 - \delta/4$, we have for all $t \in [K]$ and $\mu \in \Pi$, $\boldsymbol{Q}^\mu \in \mathcal{B}_{\mathcal{D}_{1:t-1}}(\mu)$.*

*Proof.* Repeat the arguments in the proof of Lemma 2 for each dimension $j \in [d]$ and the lemma follows directly. □

Then since $\underline{\boldsymbol{V}}^t(\mu) = f_1(s_1, \mu)$ where $f = \arg\min_{f' \in \mathcal{B}_{\mathcal{D}_{1:t-1}}(\mu)} \langle f'_1(s_1, \mu), \boldsymbol{\theta}_t \rangle$, we know for all $t \in [K]$ and $\mu \in \Pi$,

$$\langle \underline{\boldsymbol{V}}^t(\mu), \boldsymbol{\theta}_t \rangle \leq \langle \boldsymbol{Q}^\mu_1(s_1, \mu), \boldsymbol{\theta}_t \rangle = \langle \boldsymbol{V}^\mu_1(s_1), \boldsymbol{\theta}_t \rangle.$$

**Step 2: Bound estimation error.** Next we need to show the estimation error $\|\frac{1}{K}\sum_{t=1}^{K}\underline{\boldsymbol{V}}^{t}(\mu^{t}) - \boldsymbol{V}_{1}^{\mu^{t}}(s_1)\|$ is small. Let $f^{t,\mu} = \arg\min_{f\in\mathcal{B}_{\mathcal{D}_{1:t-1}}(\mu)}\langle f_1(s_1,\mu),\boldsymbol{\theta}_t\rangle$. Let $f^{t,\mu,j}$ denotes the $j$-the dimension of $f^{t,\mu}$. Then we have

**Lemma 13.** *With probability at least* $1 - \delta/4$*, we have for all* $t \in [K]$*,* $h \in [H]$*,* $j \in [d]$ *and* $\mu \in \Pi$*,*

$$(a) \quad \sum_{k=1}^{t-1}\mathbb{E}_{\mu^k}\left[\left(f_h^{t,\mu,j}(s_h,a_h) - (\mathcal{T}_h^{\mu,j}f_{h+1}^{t,\mu,j})(s_h,a_h)\right)^2\right] \le \mathcal{O}(\beta),$$

$$(b) \quad \sum_{k=1}^{t-1}\left(f_h^{t,\mu,j}(s_h^k,a_h^k) - (\mathcal{T}_h^{\mu,j}f_{h+1}^{t,\mu,j})(s_h^k,a_h^k)\right)^2 \le \mathcal{O}(\beta),$$

*Proof.* Repeat the arguments in the proof of Lemma 3 for each dimension $j \in [d]$ and the lemma follows directly. $\qquad\square$

Besides, using performance difference lemma we have:

**Lemma 14.** *For any* $t \in [K]$ *and* $j \in [d]$*, we have*

$$\underline{V}^{t,j}(\mu^t) - V_1^{\mu^t,j}(s_1) = \sum_{h=1}^{H}\mathbb{E}_{\mu^t}[(f_h^{t,\mu^t,j} - \mathcal{T}^{\mu^t,j}f_{h+1}^{t,\mu^t,j})(s_h,a_h)],$$

*where* $\underline{V}^{t,j}(\mu^t)$ *is the* $j$*-th dimension of* $\underline{\boldsymbol{V}}^t(\mu^t)$*.*

Therefore, from Lemma 14 we can obtain for any $t \in [K]$ and $j \in [d]$

$$\sum_{t=1}^{K}\underline{V}^{t,j}(\mu^t) - V_1^{\mu^t,j}(s_1) = \sum_{h=1}^{H}\sum_{t=1}^{K}\mathbb{E}_{\mu^t}[(f_h^{t,\mu^t,j} - \mathcal{T}_h^{\mu^t,j}f_{h+1}^{t,\mu^t,j})(s_h,a_h)]. \tag{44}$$

Similar to Section I, from Lemma 5, conditioning on the event in Lemma 13 holds true, with probability at least $1 - \delta/4$, we have for any $j \in [d]$ and $h \in [H]$,

$$\left|\sum_{t=1}^{K}\mathbb{E}_{\mu^t}\left[(f_h^{t,\mu^t,j} - \mathcal{T}_h^{\mu^t,j}f_{h+1}^{t,\mu^t,j})(s_h,a_h)\right]\right| \le \mathcal{O}\left(\sqrt{H^2 K d_{\mathrm{BEE,V}}\log\left(\mathcal{N}_{\mathrm{cov,V}}|\Pi|d/\delta\right)}\right).$$

Substitute the above bounds into (44) and we have for any $j \in [d]$:

$$\left|\sum_{t=1}^{K}\underline{V}^{t,j}(\mu^t) - V_1^{\mu^t,j}(s_1)\right| \le \mathcal{O}\left(H^2\sqrt{K d_{\mathrm{BEE,V}}\log(\mathcal{N}_{\mathrm{cov,V}}|\Pi|d/\delta)}\right),$$

which implies if the event in Lemma 13 is true,

$$\left\|\frac{1}{K}\sum_{t=1}^{K}\underline{\boldsymbol{V}}^{t}(\mu^t) - \boldsymbol{V}_{1}^{\mu^t}(s_1)\right\| \le \mathcal{O}\left(H^2\sqrt{d}\cdot\sqrt{d_{\mathrm{BEE,V}}\log(\mathcal{N}_{\mathrm{cov,V}}|\Pi|d/\delta)/K}\right).$$

**Step 3: Bound the distance.** Now we can bound the distance $\mathrm{dist}(\boldsymbol{V}^{\widehat{\mu}}(s_1),\mathcal{C})$. First since $\widehat{\mu}$ is sampled uniformly from $\{\mu^t\}_{t=1}^{K}$, we know

$$\mathrm{dist}(\boldsymbol{V}_1^{\widehat{\mu}}(s_1),\mathcal{C}) = \mathrm{dist}\left(\frac{1}{K}\sum_{t=1}^{K}\boldsymbol{V}_1^{\mu^t}(s_1),\mathcal{C}\right).$$

By Fenchel's duality, we know

$$\mathrm{dist}\left(\frac{1}{K}\sum_{t=1}^{K}\boldsymbol{V}_1^{\mu^t}(s_1),\mathcal{C}\right) = \max_{\boldsymbol{\theta}\in\mathbb{B}(1)}\left[\left\langle\boldsymbol{\theta},\frac{1}{K}\sum_{t=1}^{K}\boldsymbol{V}_1^{\mu^t}(s_1)\right\rangle - \max_{\boldsymbol{x}\in\mathcal{C}}\langle\boldsymbol{\theta},\boldsymbol{x}\rangle\right]$$

$$\leq \max_{\boldsymbol{\theta}\in\mathbb{B}(1)} \left[ \left\langle \boldsymbol{\theta}, \frac{1}{K}\sum_{t=1}^{K}\underline{\boldsymbol{V}}^t(\mu^t) \right\rangle - \max_{\boldsymbol{x}\in\mathcal{C}}\langle\boldsymbol{\theta},\boldsymbol{x}\rangle \right] + \max_{\boldsymbol{\theta}\in\mathbb{B}(1)} \left\langle \boldsymbol{\theta}, \frac{1}{K}\sum_{t=1}^{K}\boldsymbol{V}_1^{\mu^t}(s_1) - \underline{\boldsymbol{V}}^t(\mu^t) \right\rangle,$$

where the second step is due to $\max[f_1+f_2]\leq \max f_1 + \max f_2$.

Notice by Cauchy-Schwartz inequality and Step 2, we have

$$\max_{\boldsymbol{\theta}\in\mathbb{B}(1)} \left\langle \boldsymbol{\theta}, \frac{1}{K}\sum_{t=1}^{K}\boldsymbol{V}_1^{\mu^t}(s_1) - \underline{\boldsymbol{V}}^t(\mu^t) \right\rangle \leq \left\| \frac{1}{K}\sum_{t=1}^{K}\underline{\boldsymbol{V}}^t(\mu^t) - \boldsymbol{V}_1^{\mu^t}(s_1) \right\|$$

$$\leq \mathcal{O}\big(H^2\sqrt{d}\cdot\sqrt{d_{\mathrm{BEE,V}}\log(\mathcal{N}_{\mathrm{cov,V}}|\Pi|d/\delta/K)}\big).$$

Now we only need to bound $\max_{\boldsymbol{\theta}\in\mathbb{B}(1)} \left[ \langle\boldsymbol{\theta}, \frac{1}{K}\sum_{t=1}^{K}\underline{\boldsymbol{V}}^t(\mu^t)\rangle - \max_{\boldsymbol{x}\in\mathcal{C}}\langle\boldsymbol{\theta},\boldsymbol{x}\rangle \right]$. Recall that we update $\boldsymbol{\theta}_t$ using online gradient descent. Using the conclusions from the online learning literature [20], we know

$$\max_{\boldsymbol{\theta}\in\mathbb{B}(1)} \left[ \left\langle \boldsymbol{\theta}, \frac{1}{K}\sum_{t=1}^{K}\underline{\boldsymbol{V}}^t(\mu^t) \right\rangle - \max_{\boldsymbol{x}\in\mathcal{C}}\langle\boldsymbol{\theta},\boldsymbol{x}\rangle \right]$$

$$\leq \frac{1}{K}\sum_{t=1}^{K} \left( \langle\boldsymbol{\theta}_t, \underline{\boldsymbol{V}}^t(\mu^t)\rangle - \max_{x\in\mathcal{C}}\langle\boldsymbol{\theta}_t, x\rangle \right) + \mathcal{O}(H\sqrt{d}/\sqrt{K}).$$

Further, notice that $p^t$ is updated via Hedge with loss function being $\langle\boldsymbol{\theta}_t, \underline{\boldsymbol{V}}^t(\mu)\rangle$, similarly to the analysis in Section I, we have with probability at least $1-\delta$,

$$\frac{1}{K}\sum_{t=1}^{K}\langle\boldsymbol{\theta}_t, \underline{\boldsymbol{V}}^t(\mu^t)\rangle \leq \frac{1}{K}\sum_{t=1}^{K}\langle\boldsymbol{\theta}_t, \underline{\boldsymbol{V}}^t(\mu_{\mathrm{VMDP}}^*)\rangle + \mathcal{O}(H\sqrt{d}\cdot\sqrt{\log(|\Pi|/\delta)/K}),$$

where $\mu_{\mathrm{VMDP}}^* = \arg\min_{\mu\in\Pi}\mathrm{dist}(\boldsymbol{V}_1^\mu(s_1),\mathcal{C})$. Let $P(\boldsymbol{V}_1^{\mu_{\mathrm{VMDP}}^*}(s_1))$ denote the projection of $\boldsymbol{V}_1^{\mu_{\mathrm{VMDP}}^*}(s_1)$ onto $\mathcal{C}$.

Conditioning on the event of Lemma 12 holds, we have

$$\sum_{t=1}^{K}\langle\boldsymbol{\theta}_t, \underline{\boldsymbol{V}}^t(\mu_{\mathrm{VMDP}}^*)\rangle \leq \sum_{t=1}^{K}\langle\boldsymbol{\theta}_t, \boldsymbol{V}_1^{\mu_{\mathrm{VMDP}}^*}(s_1)\rangle.$$

Therefore we have

$$\frac{1}{K}\sum_{t=1}^{K} \left( \langle\boldsymbol{\theta}_t, \underline{\boldsymbol{V}}^t(\mu^t)\rangle - \max_{x\in\mathcal{C}}\langle\boldsymbol{\theta}_t, x\rangle \right)$$

$$\leq \frac{1}{K}\sum_{t=1}^{K} \left( \langle\boldsymbol{\theta}_t, \boldsymbol{V}_1^{\mu_{\mathrm{VMDP}}^*}(s_1)\rangle - \max_{x\in\mathcal{C}}\langle\boldsymbol{\theta}_t, x\rangle \right) + \mathcal{O}\big(H\sqrt{d}\cdot\sqrt{\log(|\Pi|/\delta)/K}\big)$$

$$\leq \frac{1}{K}\sum_{t=1}^{K} \left( \langle\boldsymbol{\theta}_t, \boldsymbol{V}_1^{\mu_{\mathrm{VMDP}}^*}(s_1)\rangle - \langle\boldsymbol{\theta}_t, P(\boldsymbol{V}_1^{\mu_{\mathrm{VMDP}}^*}(s_1))\rangle \right) + \mathcal{O}\big(H\sqrt{d}\cdot\sqrt{\log(|\Pi|/\delta)/K}\big)$$

$$\leq \left\| \boldsymbol{V}_1^{\mu_{\mathrm{VMDP}}^*}(s_1) - P(\boldsymbol{V}_1^{\mu_{\mathrm{VMDP}}^*}(s_1)) \right\| + \mathcal{O}\big(H\sqrt{d}\cdot\sqrt{\log(|\Pi|/\delta)/K}\big)$$

$$= \min_{\mu\in\Pi}\mathrm{dist}(\boldsymbol{V}_1^\mu(s_1),\mathcal{C}) + \mathcal{O}\big(H\sqrt{d}\cdot\sqrt{\log(|\Pi|/\delta)/K}\big),$$

where the second step is due to $P(\boldsymbol{V}_1^{\mu_{\mathrm{VMDP}}^*}(s_1))\in\mathcal{C}$, the third step is from Cauchy-Schwartz inequality, and the last step is from the definition of $\mu_{\mathrm{VMDP}}^*$.

In conclusion, we have with probability at least $1-\delta$,

$$\mathrm{dist}(\boldsymbol{V}_1^{\widehat{\mu}}(s_1),\mathcal{C}) \leq \min_{\mu\in\Pi}\mathrm{dist}(\boldsymbol{V}_1^\mu(s_1),\mathcal{C}) + \mathcal{O}\big(H^2\sqrt{d}\cdot\sqrt{d_{\mathrm{BEE,V}}\log(\mathcal{N}_{\mathrm{cov,V}}|\Pi|d/\delta)/K)}\big).$$

This concludes our proof.

