# OpenReview forum: "Decentralized Optimistic Hyperpolicy Mirror Descent: Provably No-Regret Learning in Markov Games"
_ICLR.cc/2023/Conference — ICLR 2023 poster_

### Official Review · Reviewer_RXcH · 2022-10-21

**Confidence:** 3
**Correctness:** 4
**Technical Novelty And Significance:** 3
**Empirical Novelty And Significance:** Not applicable
**Recommendation:** 8

**Clarity, Quality, Novelty And Reproducibility:**

## Clarity

The paper is extremely well written and easy to follow.  I listed some small comments on writing earlier up in the review.  The two most useful clarifications which should be included in the revision would be:
1. Comment on computational tractability of the algorithm
2. Discuss the lower bound listed in [33]
3. Move some of discussion in Appendix on the relationship between [33] and this paper to the main paper

## Quality + Novelty

The paper provides novel theoretical contributions for developing an algorithm for no-regret learning in decentralized Markov Games.  This allows the algorithm to only manage a single player, than a centralized algorithm that is normally studied in the prior literature.  The theoretical results are strong (although no matching lower bounds are included), but seem to be correct based on lower bounds established in the simpler single agent settings.

**Strength And Weaknesses:**

## Strengths
1. The authors study a novel model for learning in Markov Games - providing the first study of no-regret learning (i.e. benchmarking against best fixed policy in hindsight) with general function approximation.
2. The theoretical contributions include developing a novel problem complexity measure similar to the Eluder dimension extended to the multi-player setting


## Weaknesses
1. The authors include no discussion on the computational complexity of the algorithm.
2. There are no theoretical matching lower bounds (or discussion along this point)
3. The authors provide no empirical results of their algorithm's performance
4. The theoretical contributions and algorithm design seems like a straightforward extension of the algorithm presented in [33] with prior mechanisms for estimating the value function with general function approximation techniques.


**Summary Of The Paper:**

## Paper Summary

This paper considers decentralized (i.e. control over a single agent) learning in Markov games with adversarial multiple opponents.  At a high level, the fundamental goal is to design a no-regret learning policy, i.e. sublinear regret to converge towards the best fixed policy in hindsight.  This is complicated in this model for two reasons (1) the system is non-stationary since the opponents are adversarial and can change over time (2) the algorithm still needs to efficiently generalize through the use of a function class.  The authors tackle this problem by designing a decentralized multi-agent reinforcement learning algorithm with function approximation which has provably sublinear no-regret learning guarantees.  The algorithm, coined DORIS, uses a technical combination of Hedge (for adapting over the adversarial nature of the problem) alongside a novel value function estimation step within the multi-agent environment.  The authors show sublinear regret guarantees with respect to a new complexity measure, the Bellman Evaluation Eluder dimension, which generalizes to multi-agent problems.  The complement these results with special guarantees for the constrained MDP and vector-valued MDP setup (although these results are not discussed in the main paper).

More concretely the authors consider an n-agent MDP $(S, A_i, r_i, P, H)$ where $S$ is the state space, $A_i$ the actions for player $i$, $r_i$ the individual reward function, $P$ the global transition distribution, and $H$ is the horizon.  Policies are decentralized (i.e. agent $i$ observes state and picks only action in $A_i$.  The authors consider the decentralized setting which means that the player only controls (let's say the first action sets $A_1$) and the rest are chosen by potential adversary.  However, the authors consider the policy revealing setting, meaning that at the end of an episode the algorithm observes the policies that were played by the other agents (i.e. distribution over actions instead of just the observed action selected).  The goal is to achieve sublinear regret guarantees, where regret is measured as:

$R(K) - \max_{\pi} \sum_k V_1^{\pi} - V_1^{\pi^k}$

where I have omitted the dependence on the policy for the other agents.  Note that this is fundamentally different from prior settings which just hope to converge to a Nash policy, this is instead attempting to compete against the best fixed policy in hindsight.

The algorithm works as follows.  First we maintain a distribution $p_t$ over the space of all possible policies $\Pi$ which is updated with exponential weights via Hedge.  The estimates or "losses" which are used in the Hedge updates are collected via an optimistic policy estimation technique, which is novel to this work over prior literature, which uses advances in optimistic policy estimation over general function classes.

The authors complement the algorithm formulation with several regret bounds under different adversarial assumptions on the problem.

## Questions
- Can you comment on the assumption that the function class for other agents is required for the algorithm? It seems a bit strange - especially since the reward functions could be modelled with different features for different agents?
- Can you comment more on the computational feasibility of the algorithm? Or are some of these issues avoided with the assumption of finite policy class?

## Minor Comments
- The hardness results in [33] are cited frequently with no formal description.  It is relatively easy to understand form context, but it would be helpful to describe that result more thoroughly
- Space before citation [44] in introduction
- On page 2 the discussion before the bolded question makes it seem as though your paper does not consider policy revealing - but that is exactly the model which is studied
- On page 8 before Adaptive Opponent maybe add some signaling for $\Pi$ being policy for controlled agent and $\Pi'$ being other agents - found the assumptions a bit hard to keep track of

EDIT: Thanks to the authors for addressing all of my comments. I have no further concerns, but still suggest that the hardness results be expanded in the revisions.

**Summary Of The Review:**

The paper provides strong theoretical contributions for understanding no-regret learning in decentralized Markov Games.  The paper is extremely well written, highlighting the main differences and contributions between the paper and the related work.  However, the paper offers no empirical results (or discussion on computational tractability of their algorithm).

---

> ### Author Response · Authors · 2022-11-11
> **Response (Part 1)**
>
> Thanks for the valuable feedback! Here are our responses:
>
> - **Function class for other agents**:
>
> First we would like to clarify that in the **decentralized setting** with possibly adversarial opponents, our algorithm DORIS is a decentralized algorithm for the player to achieve no-regret learning and **does not make assumptions on the opponent’s strategy**, which includes the function class they use (if they really use one). In the **self-play setting**, as characterized in Assumption 4, we assume each agent has **its own function class** and the function class satisfies the reliability and completeness **for this agent**. That is, suppose the function class of the $i$-th agent is $\mathcal{F} _ i = \prod _ {h=1} ^ {H} \mathcal{F} _ {h,i}$ and $\mathcal{G} _ i = \prod _ {h=1} ^ {H} \mathcal{G} _ {h,i}$, then we suppose $Q _ {h, i} ^ {\mu,\nu}\in\mathcal{F} _ {h,i}$ and $\mathcal{T} _ {h,i} ^{\mu,\nu} f _ {h+1} \in \mathcal{G} _ {h,i}$ for any $h\in[H], \mu\in\Pi _ i, \nu \in \Pi _ {-i}, f _ {h+1} \in \mathcal{F} _ {h+1,i}$ where $\mathcal{T} _ {h,i} ^{\mu,\nu}$ is the Bellman evaluation operator of agent $i$, i.e., $(\mathcal{T} _ {h,i} ^{\mu,\nu} f _ {h+1})(s,a,b)=r _ {h,i}(s,a,b) + \mathbb{E} _ {s’\sim P _ {h}(\cdot|s,a,b)}[f _ {h+1} (s’, \mu, \nu)]$. Therefore, even if the reward for each agent is different, Assumption 4 can still hold since each agent’s function class only needs to satisfy reliability and completeness **with respect to its own reward function**.
>
>
> - **Computational efficiency**:
>
> There are mainly two steps in DORIS that require computation, **optimistic policy evaluation via OptLSPE** and **hyperpolicy update via Hedge**. Assuming the policy class is finite facilitates the second step, but even with finite policy class, OptLSPE is still computationally inefficient. This is due to the **global optimism** step in OptLSPE, i.e., constructing the confidence set (Equation (3)) and finding the most optimistic estimation. This is a common issue of algorithms with general function approximation even in single-agent MDPs. For example, the global optimism step of the algorithms in [1-6] are all computationally inefficient and hard to implement.
>
> However, if we only consider **linear MGs**, computationally efficient algorithms are possible since we can use local optimism and implement Algorithm 2 by an analog of LSVI-UCB [8], which is computationally efficient. In addition, if there is a computationally efficient solver for optimistic policy evaluation with general function approximation in **single-agent MDPs**, we believe that we can utilize it here since the confidence set update rule (Equation (3)) is similar to single-agent MDPs. That said, in this work we mainly focus on the statistical complexity of learning the Markov game and thus computationally efficient algorithms are left as future works. We have clarified this in the revised paper.

---

> > ### Author Response · Authors · 2022-11-11
> > **Response (Part 2)**
> >
> > - **Lower bound in [7]**:
> >
> > We present the lower bound from [7] here:
> >
> > (Theorem 4, [7]) *There exists a Markov game with $S,A=\mathcal{O}(H)$ and an opponent who chooses policy uniformly
> > at random from an unknown set of H Markov policies in each episode, such that when the opponent’s policy is not revealed, the regret for competing with the best fixed Markov policy in hindsight is $\Omega (\min\{K, 2 ^ H\}/H)$.*
> >
> > The above lower bound shows that if the opponent's policy is not revealed, even when the opponent only plays a finite number of Markov policies, the exponential regret lower bound for competing with the best Markov policy in hindsight is inevitable, which validates the necessity of policy revealing condition. We have added this in the revised paper.
> >
> >
> >
> > - **Minor comments**:
> >
> > Thank you for pointing them out! We have revised them in the revised version.
> >
> >
> > [1] Jin, C., Liu, Q., and Miryoosefi, S. (2021a). Bellman eluder dimension: New rich classes of RL problems, and sample-efficient algorithms.
> >
> > [2] Huang, B., Lee, J. D., Wang, Z., and Yang, Z. (2021). Towards general function approximation in zero-sum Markov games.
> >
> > [3] Jin, C., Liu, Q., and Yu, T. (2021c). The power of exploiter: Provable multi-agent RL in large state spaces.
> >
> > [4] Zanette, A., Lazaric, A., Kochenderfer, M., & Brunskill, E. (2020, November). Learning near optimal policies with low inherent bellman error. In International Conference on Machine Learning (pp. 10978-10989). PMLR.
> >
> > [5] Jiang, N., Krishnamurthy, A., Agarwal, A., Langford, J., & Schapire, R. E. (2017, July). Contextual decision processes with low bellman rank are pac-learnable. In International Conference on Machine Learning (pp. 1704-1713). PMLR.
> >
> > [6] Du, S., Kakade, S., Lee, J., Lovett, S., Mahajan, G., Sun, W., & Wang, R. (2021, July). Bilinear classes: A structural framework for provable generalization in rl. In International Conference on Machine Learning (pp. 2826-2836). PMLR.
> >
> > [7] Liu, Q., Wang, Y., & Jin, C. (2022). Learning markov games with adversarial opponents: Efficient algorithms and fundamental limits. arXiv preprint arXiv:2203.06803.
> >
> > [8] Jin, C., Yang, Z., Wang, Z., & Jordan, M. I. (2020, July). Provably efficient reinforcement learning with linear function approximation. In Conference on Learning Theory (pp. 2137-2143). PMLR.

---

> > > ### Comment · Reviewer_RXcH · 2022-11-19
> > > **Response**
> > >
> > > Thanks to the authors for addressing all of my comments. I have no further concerns, but still suggest that the hardness results be expanded in the revisions.

---

### Official Review · Reviewer_HFvo · 2022-10-24

**Confidence:** 2
**Correctness:** 4
**Technical Novelty And Significance:** 4
**Empirical Novelty And Significance:** Not applicable
**Recommendation:** 8

**Clarity, Quality, Novelty And Reproducibility:**

The paper is well-written and highly novel as discussed before. As a side note, it may be better to add a conclusion section to summarize and discuss some potential future work.

### Questions
- What is the motivation to define DE dimension and corresponding BEE dimension?
- Intuitively, what information does BEE dimension captures more compared to BE dimension (or distributional Eluder dimension compared to Eluder dimension)?
- Is that possible to modify $\mathsf{DORIS}$ (or add more assumptions to the Markov games) such that when all players adopt it, their average policy converge to a Nash equilibrium?
- The lower bound in [Liu et al. (2022)] indicates an impossibility result only when both $\Pi$ and $\Pi'$ are large. Therefore, will it be possible to have infinite $\Pi$ if we assume $\Pi'$ to be small?

```
[Liu et al. (2022)] Liu, Q., Wang, Y., and Jin, C. (2022). Learning markov games with adversarial opponents: Efficient algorithms and fundamental limits. arXiv preprint arXiv:2203.06803.
```

**Strength And Weaknesses:**

### Strengths
This paper extends algorithms for Markov games under adversarial opponents from tabular MDPs to thos with general function approximation, which is considered to be significant. Meanwhile, it proposes the novel Distributional Eluder dimension and corresponding BEE dimension to aid its regret analysis, which may be of independent intereest. Meanwhile, the proposed algorithm $\mathsf{DORIS}$ also has wide application including self-play scenario, CMDPs and VMDPs.

### Weaknesses
It would be idea to have infinite policy class by imposing more assumptions. However, from my perspective, it is okay to leave it as future work for now.

**Summary Of The Paper:**

This paper investigates Markov games with adversarial opponents with general function approximation. It proposes provably efficient algorithm $\mathsf{DORIS}$ and a new complexity measure called Bellman Evaluation Eluder (BEE) dimension to aid its analysis. Finally, this algorithm is also extended to Constrained MDPs and Vector-valued MDPs.

**Summary Of The Review:**

This paper proposes the first provably efficient algorithm for Makrov games under adversarial opponents with general function approximation. This algorithm has wide application and its analysis also contains novel tools called BEE dimension.

---

> ### Author Response · Authors · 2022-11-11
> **Response (Part 1)**
>
> Thanks for the valuable feedback! Here are our responses:
>
> - **Motivation of BEE dimension**:
>
> The motivation of the BEE dimension is that with **policy-pair specific optimism**, we need to bound the **policy evaluation error** $\sum _ {t=1} ^ K (\bar{V} ^ t ( \mu ^ t) - V _ 1 ^ {\pi ^ t}(s _ 1))$ to obtain the regret bound as shown in Equation (22). From **performance difference lemma**, this is equivalent to bound the **residuals of the Bellman evaluation operator** $\sum_{t=1}^K\mathbb{E} _ {\pi ^ t}\big[(f ^ {t,\mu ^ t} _ h- \mathcal{T} ^ {\mu ^ t,\nu ^ t} _ hf ^ {t,\mu ^ t} _ {h+1})(s _ h,a _ h,b _ h)\big]$ as shown in Equation (26). However, since $f ^ {t, \mu ^ t}$ is calculated based **on the data collected by $\pi ^ {1:t-1}$**, i.e., $\mathcal{D} _ {1:t-1}$, we can only bound  $\sum_{k=1}^{t-1}\mathbb{E} _ {\pi ^ k}\big[(f ^ {t,\mu ^ t} _ h- \mathcal{T} ^ {\mu ^ t,\nu ^ t} _ hf ^ {t,\mu ^ t} _ {h+1})(s _ h,a _ h,b _ h)\big]$ for each $t\in[K]$. This is indeed the **distribution mismatch problem** in machine learning where the **training data** is $\mathcal{D} _ {1:t-1}$ while **the testing distribution** $\mathbb{E} _ {\pi ^ t}$ is different from the training data. To bound such testing error, we need to introduce a **complexity measure** on the function class $\{f _ h - \mathcal{T} ^ {\mu,\nu} _ hf _ {h+1}: f\in \mathcal{F}, \mu\in\Pi, \nu\in\Pi’\}$, which is a standard method in the machine learning literature. In our setting this complexity measure is exactly the **BEE dimension**.
>
>
> - **BEE dimension vs BE dimension**:
>
> Different from the BE dimension [1] in the literature, the BEE dimension captures **the information of policy classes** too and measures the residuals of $f-\mathcal{T} _ h ^ {\mu,\nu}f$ for all policy pair $(\mu,\nu)$ and $f\in\mathcal{F}$. In comparison, BE dimension is determined **solely by the function class** $\mathcal{F}$. The information of policy classes enables BEE dimension to bound the testing error  $\sum_{t=1}^K\mathbb{E} _ {\pi ^ t}\big[(f ^ {t,\mu ^ t} _ h- \mathcal{T} ^ {\mu ^ t,\nu ^ t} _ hf ^ {t,\mu ^ t} _ {h+1})(s _ h,a _ h,b _ h)\big]$ from **training error** $\sum_{k=1}^{t-1}\mathbb{E} _ {\pi ^ k}\big[(f ^ {t,\mu ^ t} _ h- \mathcal{T} ^ {\mu ^ t,\nu ^ t} _ hf ^ {t,\mu ^ t} _ {h+1})(s _ h,a _ h,b _ h)\big]$, which is a quantity **related to the executed policy**, while BE dimension cannot achieve this.
>
> - **Finding Nash equilibrium**:
>
> In general, when the number of players is greater than 2, the uniform mixture of the played policies constitutes an **approximate coarse correlated equilibrium**, as we have shown in Corollary 1. However, in the special setting of **two-player zero-sum Markov games**, the average policy $(\hat{\mu} _ 1, \hat{\mu} _ 2)$ is an **approximate Nash equilibrium** where $\hat{\mu} _ 1$ and $\hat{\mu} _ 2$ are the average policies of the max player and min player. This can be proved using the well-known conclusion in game theory that no-regret learning in two-player zero-sum games leads to Nash equilibrium. More specifically, assume there are two players and $r_{h,1}=1-r_{h,2}$, then we have $\max _ {\mu _ 1} V _ {1,1} ^ {\mu _ 1, \hat{\mu} _ 2} (s _ 1)- \min _ {\mu _ 2 } V _ {1,1} ^ {\hat{\mu} _ 1, \mu _ 2} (s _ 1)=\max _ {\mu _ 1} V _ {1,1} ^ {\mu _ 1, \hat{\mu} _ 2} (s _ 1)- \big ( H - \max _ {\mu _ 2 } V _ {1,2} ^ {\hat{\mu} _ 1, \mu _ 2} (s _ 1)\big)$, which is further equivalent to $\big(\max _ {\mu _ 1} V _ {1,1} ^ {\mu _ 1, \hat{\mu} _ 2} (s _ 1) - V _ {1,1} ^ {\hat{\pi}} (s _ 1)\big) +  \big ( \max _ {\mu _ 2 } V _ {1,2} ^ {\hat{\mu} _ 1, \mu _ 2} (s _ 1) - V _ {1,2} ^ {\hat{\pi}} (s _ 1)\big)$ where $\hat{\pi} \sim \text{Unif}(\{ \mu _ 1 ^ 1 \times \mu _ 2 ^ 1, \cdots, \mu _ 1 ^ K \times \mu _ 2 ^ K \} )$ is the output of Algorithm 3. Then applying Theorem 2, we know $\max _ {\mu _ 1} V _ {1,1} ^ {\mu _ 1, \hat{\mu} _ 2} (s _ 1) - V _ {1,1} ^ {\hat{\pi}} (s _ 1)\leq \epsilon$ and  $\max _ {\mu _ 2 } V _ {1,2} ^ {\hat{\mu} _ 1, \mu _ 2} (s _ 1) - V _ {1,2} ^ {\hat{\pi}} (s _ 1)\leq\epsilon$, which implies that the average policy $(\hat{\mu} _ 1, \hat{\mu} _ 2)$ is an approximate Nash equilibrium.

---

> > ### Author Response · Authors · 2022-11-11
> > **Response (Part 2)**
> >
> > - **Infinite player’s policy class**:
> >
> > It is possible to achieve no-regret learning with infinite player’s policy class and finite opponent’s policy class. The reason why we do not consider such cases here is that [2] implies that if the player’s policy class is infinite and the opponent's policy class is exponentially large, no-regret learning in the MG is still hard. Considering that the policy space can be **exponentially large** easily, we do not focus on this case in this paper (recall that when the player’s policy class is finite, our results indeed don’t scale with $|\Pi’|$ or only scale with $\log |\Pi’|$ as shown in Theorem 1 and 2 and thus we can deal with exponentially large policy classes).
> >
> > [1] Jin, C., Liu, Q., and Miryoosefi, S. (2021a). Bellman eluder dimension: New rich classes of RL problems, and sample-efficient algorithms.
> >
> > [2] Liu, Q., Wang, Y., & Jin, C. (2022). Learning markov games with adversarial opponents: Efficient algorithms and fundamental limits. arXiv preprint arXiv:2203.06803.

---

> > ### Comment · Reviewer_HFvo · 2022-11-26
> > **Response**
> >
> > Thank you very much for your detailed explanation and my concern has been well-addressed!

---

### Official Review · Reviewer_8Rj7 · 2022-10-24

**Confidence:** 4
**Correctness:** 4
**Technical Novelty And Significance:** 3
**Empirical Novelty And Significance:** Not applicable
**Recommendation:** 6

**Clarity, Quality, Novelty And Reproducibility:**

The quality is good. The paper is clear to follow overall. The technique seems to be a combination of previous decentralized Markov games in the tabular case and the general function approximation with low BE dimension but is tailored to the multi-agent case.

**Strength And Weaknesses:**

Strength:

The paper first studies the general function approximation problem in the decentralized Markov games, which generalizes previous works [33] for the tabular setting. The work provides complete results including regret bounds for oblivious and adaptive adversaries,
guarantee for CCE results, and also discusses other generalizations to CMDP and VMDP.

Weakness:

1. More comparisons on the regret order with previous works for decentralized Markov games in tabular case (when reducing the current setting to the tabular case) and single-agent MDP with bounded BE dimension.

2. Given previous works for decentralized Markov games in tabular settings and MDP with the bounded BE dimension, the theoretical contribution seems to be weak. The authors discussed in Appendix E that the difference between these works lies in the construction of optimism. Though the learner should build confidence sets for each \mu when faced with each \nu_k, the global optimism seems to hold by modifying the complexity with respect to the two policies of the agent itself and adversarial opponents.

3. Minor: The notation is not easy to follow. The paper uses \mu_i and \mu_{-i} to represent the policy of agent i and the joint policy of other agents while using \pi to represent the joint policy of all agents. Why not use \mu to represent the joint policy? I think it will be more consistent. The paper uses t to represent the index of both epoch and step in Section 2. Perhaps it would be better only use it to represent epoch but use h to represent step. Algorithm 1 shows “run \pi_t =\mu_t \times \nu_t” to collect data. In fact \nu_t is not run by
the controlled agent.

Question:
Since the paper regards all of other agents as adversarial opponents whose actions cause the non-stationarity of the transition, I am wondering whether such a technique can deal with single-agent MDP with adversarial transitions. Could the authors give some intuitions?

**Summary Of The Paper:**

This paper studies the decentralized Markov games with possibly adversarial opponents. Specifically, only one agent in the game can be controlled and others are regarded as adversarial opponents. The paper first considers the general function approximation problem in this setting with the bounded Bellman Evaluation Eluder dimension. To deal with the problem, it proposes an algorithm that is a variant of Online Mirror Descent with optimistic estimated V values. The regret bounds for both oblivious and adaptive adversarial settings are provided. When all agents run this algorithm simultaneously, the mixed policy is also an epsilon-CCE with sample complexity 1/epsilon^2. The paper finally discusses the application of the algorithm in the case of constrained MDP (CMDP) and vector-valued MDP (VMDP).

**Summary Of The Review:**

In general, this paper generalizes previous works for decentralized Markov games in tabular case to general function approximation with bounded Bellman Evaluation Eluder dimension. Though the technique is not novel, the results are good compliments of existing literature.

---

> ### Author Response · Authors · 2022-11-11
> **Response (Part 1)**
>
> Thanks for the valuable feedback! Here are our responses:
>
> - **Comparison of regret with tabular cases**:
>
> In the appendix we provide the bound of covering number and BEE dimension for Kernel Markov games (including tabular cases) and generalized linear complete models. Substituting the results into Theorem 1 and 2, we can obtain that the regret upper bounds for the oblivious setting and adaptive setting in tabular cases are $\tilde{O}(H^{1.5}V_{\max}|\mathcal{S}||\mathcal{A}||\mathcal{B}|\sqrt{K\log|\Pi|})$ and $\tilde{O}(H^{1.5}V_{\max}|\mathcal{S}||\mathcal{A}||\mathcal{B}|\sqrt{K\log(|\Pi||\Pi'|)})$. The corresponding tabular case result in the literature [1] is $\tilde{O}(H\sqrt{K\log|\Pi|}+H^2|\mathcal{S}|\sqrt{|\mathcal{A}||\mathcal{B}|}K)$.
>
> Our results are slightly worse, which is the cost for **achieving generality in the function approximation setting**. If we know that we are in the tabular setting, then we can design tighter confidence regions that leverage such knowledge. But such confidence regions cannot be extended to the general case. Moreover, even for MDPs, it is common that results in the case of function approximation, when restricted to the tabular case, yield a slightly bad regret. For example, in single-agent MDPs, when reducing the results in linear MDPs [2] to tabular MDPs, the regret is also not minimax optimal because linear MDPs also cover linear bandits and for linear bandits the regret lower bound is $\tilde{O}(d\sqrt{K})$ where $d$ is the dimension. Thus, the regret for linear MDPs is also at least $\tilde{O}(d\sqrt{K})$, which is larger than the minimax regret in tabular MDPs $\tilde{O}(\sqrt{SAK})$ [3].
>
> In addition, as mentioned in Remark 4, the regret for the adaptive setting depends on the size of $\Pi’$, which is not the case in the tabular setting [1]. This dependency originates from our **model-free** type of policy evaluation algorithm (Algorithm 2) and is inevitable for DORIS in general. That said, when the Markov game has **special structures** (e.g., the Markov games in Appendix C and D), we can avoid this dependency.
>
>
> - **Comparison of regret with single-agent MDPs**:
>
> When reducing to single-agent MDPs, the regret of DORIS is $\tilde{O}(HV_{\max}\sqrt{Kd _ {BEE} \log(\mathcal{N} _ {cov}|\Pi|}))$. In comparison, the regret bound for GOLF [4] is $\tilde{O}(H\sqrt{Kd _ {BE}\log(\mathcal{N} _ {cov}|\Pi|}))$ where they assume $V _ {\max}=1$. Note that when reducing to single-agent MDPs, the BEE dimension is still different from the BE dimension in [4] because the BEE dimension measures the residuals of the Bellman evaluation operator while the BE dimension measures the residuals of the Bellman operator (i.e., $(\mathcal{T}f)(s,a)=r(s,a)+\mathbb{E} _ {s’\sim P(\cdot|s,a)} \max _ {a}[f (s’,a)]$). In addition, our result depends on $\log|\Pi|$ while [4] doesn’t. As discussed in Appendix E, this is due to the **different optimism and confidence construction mechanism** of DORIS, which is required in our setting but not necessary in single-agent MDPs.
>
>
> - **Optimism Construction**:
>
> In fact in Appendix E we have mentioned that our optimism mechanism is **global optimism** (compared with the local optimism in [1]). The difference between DORIS and other existing works with general function approximation [5-6] is that our optimism is with respect to $V ^ {\mu,\nu^k} _ 1 (s _ 1)$ for **each policy pair** $(\mu, \nu ^ k)$ while [5-6] are optimistic with respect to the **Nash value function** $V ^ * _1 (s_1)$. Consequently, DORIS is able to compete against the **best policy in hindsight** while [5-6] can only compete against the Nash value of the game, which is a much weaker baseline.
>
> - **Notations**:
>
> Thank you so much for pointing them out! The reason why we use $\pi$ for the joint policy is that $\mu$ already represents the player’s policy in the decentralized setting. We have followed the other suggestions in the revised paper.

---

> > ### Author Response · Authors · 2022-11-11
> > **Response (Part 2)**
> >
> > - **Extension to adversarial MDPs**:
> >
> > The extension is possible but not straightforward since DORIS requires the policy revealing condition and it is not clear what structure this corresponds to in adversarial MDPs. An intuitive idea would be to directly model the adversarial MDP into a Markov game where there is a “latent” opponent that the player cannot observe. Then if the components that are adversely changed (such as the transition and the reward function) are revealed after each episode, we can directly apply DORIS.
> >
> >
> > [1] Liu, Q., Wang, Y., & Jin, C. (2022). Learning markov games with adversarial opponents: Efficient algorithms and fundamental limits. arXiv preprint arXiv:2203.06803.
> >
> > [2] Hu, P., Chen, Y., & Huang, L. (2022, June). Nearly minimax optimal reinforcement learning with linear function approximation. In International Conference on Machine Learning (pp. 8971-9019). PMLR.
> >
> > [3] Azar, M. G., Osband, I., & Munos, R. (2017, July). Minimax regret bounds for reinforcement learning. In International Conference on Machine Learning (pp. 263-272). PMLR.
> >
> > [4] Jin, C., Liu, Q., and Miryoosefi, S. (2021a). Bellman eluder dimension: New rich classes of RL problems, and sample-efficient algorithms.
> >
> > [5]Huang, B., Lee, J. D., Wang, Z., and Yang, Z. (2021). Towards general function approximation in zero-sum Markov games.
> >
> > [6] Jin, C., Liu, Q., and Yu, T. (2021c). The power of exploiter: Provable multi-agent RL in large state spaces.

---

> > > ### Comment · Reviewer_8Rj7 · 2022-11-30
> > > **Further questions**
> > >
> > > Thanks for your reply to all of my questions. I still have some concerns.
> > > 1. Is there any typo in your presented regret $\tilde{O}(H\sqrt{K\log \Pi}+H^2S\sqrt{AB}K)$ for [1] since the presented regret is linear in $K$?
> > > 2. When comparing your result with GOLF, you say they have no dependence on $\Pi$ but the presented result does depend on it. Is there any definition confusion or is it just a typo? I also suggest the authors provide these comparisons in the revised version.
> > > 3. When deriving the result in the 'reduced' single-agent case, can you give a more detailed explanation on how to ‘reduce’ to a single-agent setting?

---

> > > > ### Author Response · Authors · 2022-11-30
> > > > **Further Response**
> > > >
> > > > Thanks for your feedback!
> > > >
> > > > First, we apologize for the two typos in the original response. The regret in [1] should be $\tilde{O}(H\sqrt{K\log|\Pi|}+H^2|\mathcal{S}|\sqrt{|\mathcal{A}||\mathcal{B}| K})$ and the regret of GOLF should be $\tilde{O}(H\sqrt{Kd _ {BE}\log\mathcal{N} _ {cov}})$.
> > > >
> > > >
> > > > In addition, for the single -agent case, intuitively you can just consider the case where the opponent’s action does not affect the state transition and rewards. More specifically, for the algorithm, we no longer require policy-revealing condition and in each iteration we replace the confidence set construction (Eq (3) in Algorithm 2) with its counterpart in the single-agent setting, i.e., let $\mathcal{B} _ {\mathcal{D}} (\mu)$ be all functions $f\in\mathcal{F}$ such that:
> > > > $$ \sum _ {(s _ h ,a _ h ,r _ h ,s _ {h+1})\in\mathcal{D}} [f _ h (s _ h, a _ h)- r _ h - f _ {h+1} (s _ {h+1},\mu)]^2 \leq \inf _ {g\in\mathcal{F}} \sum _ {(s _ h ,a _ h ,r _ h ,s _ {h+1})\in\mathcal{D}} [g _ h (s _ h,a _ h)-r _ h - f _ {h+1} (s _ {h+1},\mu)]^2 + \beta.$$
> > > > Notice that now the function class $\mathcal{F}$ approximates the single-agent  Q function $Q^ {\mu} _ h (s,a)$.
> > > >
> > > > For the analysis, the definition of BEE dimension (Definition 6) will become $\max _ {h} \mathrm{dim} _ {\mathrm{DE}} ( (\mathcal{I} - \mathcal{T} _ h ^ {\Pi}) \mathcal{F}, \mathcal{Q} _ h, \epsilon )$. Here $(\mathcal{I} - \mathcal{T} _ h ^ {\Pi}) \mathcal{F}$ contains all $f _ {h} - \mathcal {T} _ h ^ {\mu} f _ {h+1}$ where $f\in\mathcal{F}, \mu\in\Pi$ and $ \mathcal {T} _ h ^ {\mu}$ is the single-agent Bellman evaluation operator, i.e., $\mathcal {T} _ h ^ {\mu} f _ {h+1}(s,a) = r _ {h} (s,a) + \mathbb{E} _ {s'\sim P _ h(\cdot|s,a)}[f _ {h+1} (s', \mu)] $. With this single-agent BEE dimension, our original analysis still holds and leads to a regret bound of $\tilde{O}(HV_{\max}\sqrt{Kd _ {BEE} \log(\mathcal{N} _ {cov}|\Pi|}))$ for DORIS where $ d _ {BEE}$ is now the single-agent BEE dimension.

---

> > > > > ### Comment · Reviewer_8Rj7 · 2022-12-01
> > > > > **Reply to authors**
> > > > >
> > > > > Thanks for your detailed reply. I have no further concerns and would maintain my score.

---

### Official Review · Reviewer_pavp · 2022-10-25

**Confidence:** 3
**Correctness:** 4
**Technical Novelty And Significance:** 3
**Empirical Novelty And Significance:** Not applicable
**Recommendation:** 6

**Clarity, Quality, Novelty And Reproducibility:**

**Clarity**: The paper is very clear. Problem settings, notations and definitions, and results are clearly presented. The organization and writing are also easy to follow.

**Quality**: The contribution is clear and important to my understanding. The results are technically sound.

**Originality**: The originality is OK. The policy revealing setting has been introduced before, and the BEE dimension is adapted from existing measures. Combining those results together seems non-trivial.

**Strength And Weaknesses:**

**Strength**:

1. DORIS achieves $\sqrt{K}$ regret in terms of Definition 1, using function approximations with low BEE dimensions, which is a stronger result than NEs.
2. The proposed BEE dimension seems a reasonable complexity measure in this setting, and the authors provided calculations to show the BEE dimensions for kernel MGs and generalized linear complete models.
3. Comparison with existing works are also discussed clearly.

**Weaknesses**:

1. The current results are for episodic settings. How do you generalize the techniques to other scenarios like discounted settings?
2. As noted in the paper, computational efficiency could be a problem for Algorithm 2. Is it possible that for some specific models (such as the generalized linear complete models mentioned), Algorithm 2 can be implemented in practice?
3. I found the policy revealing assumption kind of strong. Revealing the opponents' policies seems weaken the results for adversarial settings studied in the paper. I feel this part needs better argument, such as using examples to show that policy revealing does appear in practice.

**Summary Of The Paper:**

This paper proposes an algorithm Decentralized Optimistic hypeRpolicy mIrror deScent (DORIS) for decentralized policy learning in Markov games with function approximations. The setting is "policy revealing", where the opponents (could be adversarial) would reveal its previous policies to the agent for making decisions.

To combat non-stationarity, DORIS (Algorithm 1) mains a "hyperpolicy", which is a probability distribution over policies of the agent, and uses Hedge / mirror descent (MD) to update the hyperpolicy, with estimated value functions. The value function is estimated using optimism in LSPE, as shown in Algorithm 2.

To upper bound the value function estimation errors, the authors proposed a new complexity measure called Bellman Evaluation Eluder (BEE) dimension of function classes, with discussions of its relation to existing measures like Eluder dimension.

Under finite policy class Assumptions 1 and 5, and realizability and completeness Assumptions 2, 3 and 4, the authors proved that the proposed DORIS algorithm achieves $\sqrt{K}$ regret comparing to best policy in hindsight in both oblivious and adversarial opponent settings, as shown in Theorems 1 and 2, where $K$ is the number of episodes.

Finally, Corollary shows that if every agent is playing DORIS algorithm simultaneously, then with high probability the uniformly mixed policy can find a $\epsilon$-approximate coarse correlated equilibrium (CCE) in $\mathcal{O}(1/\epsilon^2)$ iterations.

In the appendix, the authors also studied using DORIS in constrained MDPs and vector-valued MDPs, where similar regret bounds can be obtained.

**Summary Of The Review:**

Overall, I found both the results and the technical contributions interesting and sound.

---

> ### Author Response · Authors · 2022-11-11
> **Response (Part 1)**
>
> Thanks for the valuable feedback! Here are our responses:
>
> - **Extension to the discounted setting**:
>
> A simple way to extend our algorithm to the discounted setting is to replace the **confidence set construction** (Eq (3) in Algorithm 2) with its counterpart in the discounted setting, i.e., let $\mathcal{B} _ {\mathcal{D}} (\mu,\nu)$ be all functions $f\in\mathcal{F}$ such that:
> $$ \sum _ {(s,a,b,r,s’)\in\mathcal{D}} [f (s,a,b)-r-\gamma f(s’,\mu,\nu)]^2 \leq \sum _ {(s,a,b,r,s’)\in\mathcal{D}} \inf _ {g\in\mathcal{F}} [g (s,a,b)-r-\gamma f(s’,\mu,\nu)]^2 + \beta.$$ In addition, the **new sample** collected at $k$-th iteration in the dataset $\mathcal{D}$ should follow the discounted visitation distribution $d ^ {\pi ^ k}$. To achieve this, we can let the player and opponent execute their policies and end with probability $1-\gamma$ at each time step. Suppose the last timestep is $t$, then we add $(s _ t, a _ t, b _ t, r _ t, s _ {t+1})$ to the dataset. This guarantees that $(s _ t, a _ t, b _ t)\sim d ^ {\pi ^ k}$.
>
>
> For the analysis, we only need to replace the Bellman evaluation operator in the **completeness assumption** with the discounted version, i.e., $(\mathcal{T} ^ {\mu,\nu} f)(s,a,b) = r(s,a,b) + \gamma\mathbb{E} _ {s’\sim P(\cdot|s,a,b)}[f(s’,\mu,\nu)]$ and define the **BEE dimension** with respect to the **discounted Bellman residuals**. Then we can show optimism and the regret decomposition / performance difference lemma are all similar to the episodic setting, which enables us to extend the analysis naturally to the discounted setting. Note that the regret in the discounted setting can also be defined as $\sum_{k=1}^HV^*-V^{\pi^k}$ and an upper bound on regret still implies a PAC learning bound.
>
> - **Computational Efficiency**:
>
> Implementing our algorithm in practice involves approximately achieving **global optimism** in the context of **general function approximation**, which is difficult in practice even for generalized linear complete models. This is indeed a common issue of algorithms with general function approximation, even in single-agent MDPs. For example, the global optimism step of the algorithms in [1-6] are all computationally inefficient and hard to implement. Note that even though [4] only considers **linear bellman complete MDPs**, it is still not computationally efficient.
>
> However, if we only consider **linear MGs**, computationally efficient algorithms are possible since we can use local optimism and implement Algorithm 2 by an analog of LSVI-UCB [9], which is computationally efficient. In addition, if there is a computationally efficient solver for optimistic policy evaluation with general function approximation in **single-agent MDPs**, we believe that we can utilize it here since the confidence set update rule (Equation (3)) is similar to single-agent MDPs. We have clarified this in the revised paper.

---

> > ### Author Response · Authors · 2022-11-11
> > **Response (Part 2)**
> >
> > - **Policy Revealing Assumption**:
> >
> > We admit that the policy revealing condition is not very common in the literature. However, the information structure defined in our paper is the best we can hope for to achieve no-regret learning. As shown in Liu et al. 2022, if the policy is not revealed, the no-regret learning problem can be reduced to POMDPs or Latent MDPs, which are statistically hard to solve in general.
> >
> > When the opponent’s policy is not revealed but changes slowly, we indeed can **infer** the opponent’s policy approximately via similar procedures in [7-8] and this can be viewed as an **approximate policy revealing condition** in practice. In this case, we can simply run DORIS with the inferred opponent’s policy to achieve a good performance. We have clarified this in the revised paper.
> >
> >
> >
> >
> >
> > [1] Jin, C., Liu, Q., and Miryoosefi, S. (2021a). Bellman eluder dimension: New rich classes of RL problems, and sample-efficient algorithms.
> >
> > [2] Huang, B., Lee, J. D., Wang, Z., and Yang, Z. (2021). Towards general function approximation in zero-sum Markov games.
> >
> > [3] Jin, C., Liu, Q., and Yu, T. (2021c). The power of exploiter: Provable multi-agent RL in large state spaces.
> >
> > [4] Zanette, A., Lazaric, A., Kochenderfer, M., & Brunskill, E. (2020, November). Learning near optimal policies with low inherent bellman error. In International Conference on Machine Learning (pp. 10978-10989). PMLR.
> >
> > [5] Jiang, N., Krishnamurthy, A., Agarwal, A., Langford, J., & Schapire, R. E. (2017, July). Contextual decision processes with low bellman rank are pac-learnable. In International Conference on Machine Learning (pp. 1704-1713). PMLR.
> >
> > [6] Du, S., Kakade, S., Lee, J., Lovett, S., Mahajan, G., Sun, W., & Wang, R. (2021, July). Bilinear classes: A structural framework for provable generalization in rl. In International Conference on Machine Learning (pp. 2826-2836). PMLR.
> >
> > [7] Nakamura, M., & Ohtsuki, H. (2016). Optimal decision rules in repeated games where players infer an opponent’s mind via simplified belief calculation. Games, 7(3), 19.
> >
> > [8] Shen, M., & How, J. P. (2021, May). Robust opponent modeling via adversarial ensemble reinforcement learning. In Proceedings of the International Conference on Automated Planning and Scheduling (Vol. 31, pp. 578-587).
> >
> > [9] Jin, C., Yang, Z., Wang, Z., & Jordan, M. I. (2020, July). Provably efficient reinforcement learning with linear function approximation. In Conference on Learning Theory (pp. 2137-2143). PMLR.

---

> > > ### Comment · Reviewer_pavp · 2022-12-13
> > > **Thanks for the comments**
> > >
> > > Thank you for the comments and references. I would like to keep my current score.

---

### Decision · Program_Chairs · 2023-01-20

**Decision:**

Accept: poster

**Justification For Why Not Higher Score:**

The weaknesses listed above, as well as the strength of the support from the reviewers (weak accept on average), suggest that a poster presentation is the most appropriate for the paper, although depending on the quality of other papers, a spotlight presentation may also be considered.

**Justification For Why Not Lower Score:**

The paper presents novel results which are worthy of publication.

**Metareview: Summary, Strengths And Weaknesses:**

The paper studies decentralized policy learning in Markov games with function approximations under the "policy revealing" setting. An algorithm with regret bounds is provided for both oblivious and non-oblivious opponents. The paper is well written and the analysis is sound, with a new complexity measure being introduced. The weaknesses of the paper include the assumption that the policy of the other agents is revealed, the extension from the tabular case to the function approximation case is not very novel (but definitely sufficiently novel for publication), and the algorithm is not computationally efficient (although it is for the linear case).

**Note From Pc:**

if the above contains the word "oral" or "spotlight" please see: "oral" presentation means -> notable-top-5% and "spotlight" means -> notable-top-25%. As stated in our emails, we are disassociating presentation type from AC recommendations